# Epitope editing enables targeted immunotherapy of acute myeloid leukaemia

Gabriele Casirati[1,2,3], Andrea Cosentino[1,3,4], Adele Mucci[1,3], Mohammed Salah Mahmoud[1,3,5], Iratxe Ugarte Zabala[1,3,6], Jing Zeng[1,3], Scott B. Ficarro[7,8,9,10], Denise Klatt[1,3], Christian Brendel[1,3,11], Alessandro Rambaldi[4], Jerome Ritz[1,11,12], Jarrod A. Marto[7,8,9,10,11], Danilo Pellin[1,3,11], Daniel E. Bauer[1,3,11], Scott A. Armstrong[1,3,11] & Pietro Genovese[1,3,11 ✉]

Despite the considerable efficacy observed when targeting a dispensable lineage antigen, such as CD19 in B cell acute lymphoblastic leukaemia[1,2], the broader applicability of adoptive immunotherapies is hampered by the absence of tumour-restricted antigens[3–5]. Acute myeloid leukaemia immunotherapies target genes expressed by haematopoietic stem/progenitor cells (HSPCs) or differentiated myeloid cells, resulting in intolerable on-target/off-tumour toxicity. Here we show that epitope engineering of donor HSPCs used for bone marrow transplantation endows haematopoietic lineages with selective resistance to chimeric antigen receptor (CAR) T cells or monoclonal antibodies, without affecting protein function or regulation. This strategy enables the targeting of genes that are essential for leukaemia survival regardless of shared expression on HSPCs, reducing the risk of tumour immune escape. By performing epitope mapping and library screenings, we identified amino acid changes that abrogate the binding of therapeutic monoclonal antibodies targeting FLT3, CD123 and KIT, and optimized a base-editing approach to introduce them into CD34+ HSPCs, which retain long-term engraftment and multilineage differentiation ability. After CAR T cell treatment, we confirmed resistance of epitope-edited haematopoiesis and concomitant eradication of patient-derived acute myeloid leukaemia xenografts. Furthermore, we show that multiplex epitope engineering of HSPCs is feasible and enables more effective immunotherapies against multiple targets without incurring overlapping off-tumour toxicities. We envision that this approach will provide opportunities to treat relapsed/refractory acute myeloid leukaemia and enable safer non-genotoxic conditioning.

CAR T cells, bispecific antibodies and antibody–drug conjugates are promising adoptive immunotherapies that can overcome the limitations of conventional cancer treatments and have demonstrated considerable efficacy when targeting dispensable haematopoietic antigens[1,2]. Nonetheless, the absence of safely actionable tumour-restricted markers hampers their application to other haematological malignancies, such as acute myeloid leukaemia (AML)[6,7]. As AML shares most surface markers with normal HSPCs or differentiated myeloid cells, on-target/off-tumour toxicities would result in myeloid aplasia and impairment of haematopoietic reconstitution[3,4,8]. Furthermore, owing to AML intratumoural heterogeneity and plasticity[9], targeting more than one surface antigen may be required, therefore exacerbating the risk of overlapping toxicity[5,10]. Despite this, a range of AML immunotherapies is currently under development[11–16], but their role will probably be time restricted to bridge treatment before allogeneic HSPC

transplantation (HSCT), decreasing the chances of AML eradication. Removal of the targeted antigen through CRISPR–Cas knockout or exon skipping from donor HSPCs used in HSCT has recently been proposed[17–19] to reduce the adverse effects associated with anti-CD33 treatments and this approach is currently undergoing clinical testing (ClinicalTrials.gov: NCT04849910). However, although these studies provided evidence for the dispensable role of CD33 for engraftment and myeloid differentiation in non-human primates, the long-term effects of *CD33* knockout on myeloid cell functionality in humans remain unclear[20,21]. Furthermore, targeting non-essential genes would facilitate tumour escape through antigen loss or downregulation, as observed in CD19-negative relapses after CD19 CAR T cell therapy[22–24] or HLA-loss after haplo-HSCT[25]. Here we show that precise editing of the targeted epitope on FMS-like tyrosine kinase 3 (FLT3; also known as CD135), KIT (also known as CD117) and the α subunit of the IL-3 receptor

[1]Division of Hematology/Oncology, Boston Children's Hospital, Boston, MA, USA. [2]Milano-Bicocca University, Milan, Italy. [3]Department of Pediatric Oncology, Dana-Farber Cancer Institute, Boston, USA. [4]Department of Oncology and Hematology, University of Milan and Azienda Socio-Sanitaria Territoriale Papa Giovanni XXIII, Bergamo, Italy. [5]Zoology Department, Faculty of Science, Fayoum University, Fayoum, Egypt. [6]German Cancer Research Center (DKFZ), Heidelberg, Germany. [7]Department of Cancer Biology, Dana-Farber Cancer Institute, Boston, MA, USA. [8]Department of Pathology, Brigham and Women's Hospital, Boston, MA, USA. [9]Blais Proteomics Center, Dana-Farber Cancer Institute, Boston, MA, USA. [10]Center for Emergent Drug Targets, Dana-Farber Cancer Institute, Boston, MA, USA. [11]Harvard Medical School, Boston, MA, USA. [12]Connell and O'Reilly Families Cell Manipulation Core Facility, Dana-Farber Cancer Institute, Boston, USA. ✉e-mail: Pietro_Genovese@dfci.harvard.edu

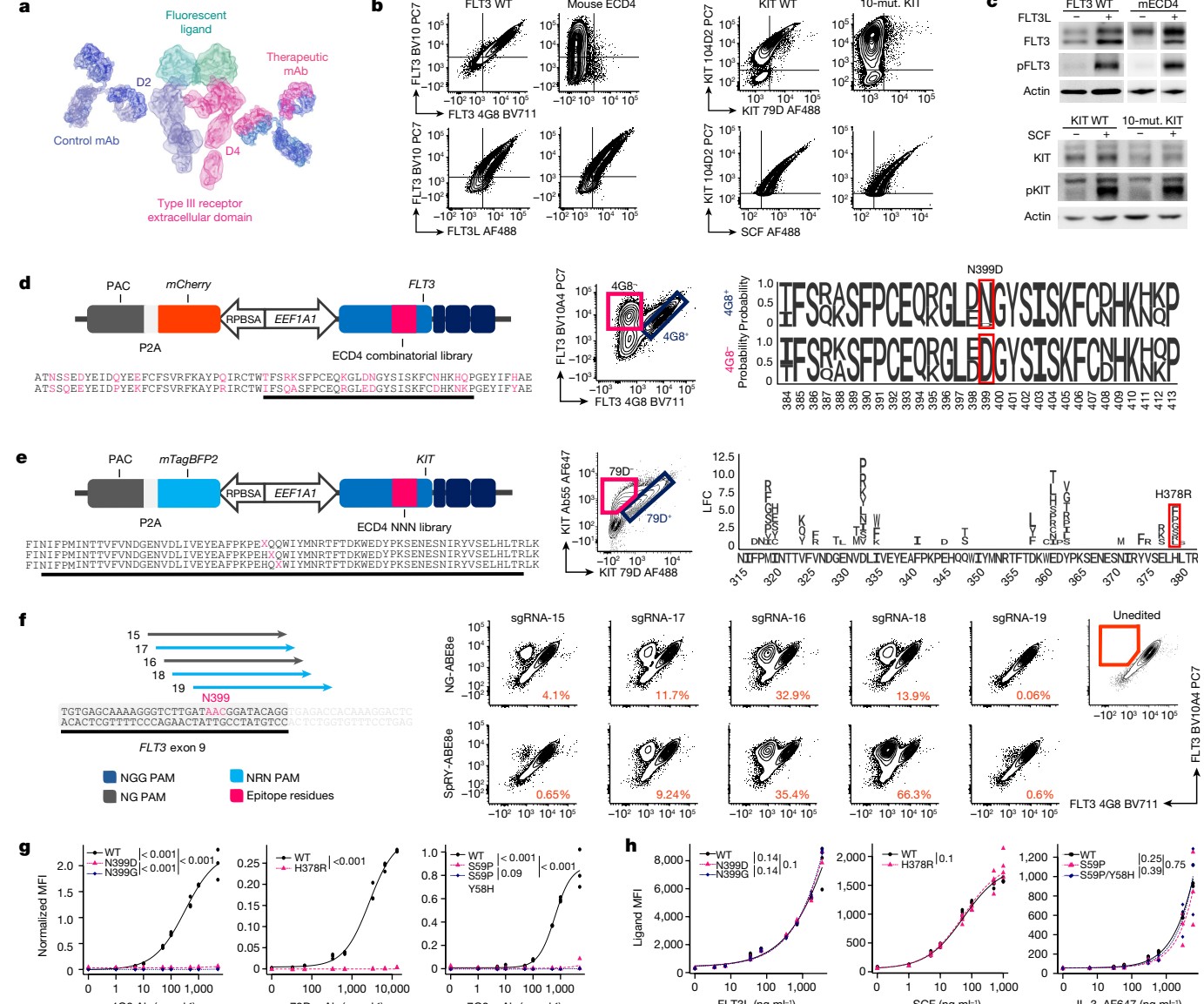

**Fig. 1 | Epitope engineering can be achieved by base editing. a**, Type III receptor tyrosine kinases bound to control and therapeutic antibodies. AlexaFluor 488 (AF488)- or AF647-conjugated ligands were used to assess binding affinity. Protein models are based on Protein Data Bank 3QS9 (FLT3) and 1IGT (Ig). mAb, monoclonal antibody. **b**, Loss of 4G8 and Fab-79D binding to FLT3 with 16 amino acid substitutions or KIT with 10 amino acid substitutions, respectively (top). Bottom, ligand assay for WT and mutated receptor variants. Mut., mutation. **c**, Western blot analysis of phosphorylated FLT3 at Tyr589–591 and KIT at Tyr719 on receptor variants. **d**, FLT3 combinatorial library. Left, the Sleeping Beauty plasmid expressing *FLT3* with human or mouse codons at 16 positions (red) within ECD4. Middle, K562 cells were transduced with the FLT3 library. 4G8⁻ and 4G8⁺ fractions were sorted using fluorescence-activated cell sorting (FACS) and sequenced by next-generation sequencing (NGS). Right, the relative amino acid frequency at positions 384–413. **e**, KIT epitope mapping. Left, the Sleeping Beauty plasmid containing degenerated codons (NNN) at each position (red) of ECD4. Middle, K562 cells were transduced with the KIT

library. Fab-79D⁻ and Fab-79D⁺ fractions were FACS-sorted and sequenced. Right, the log-transformed fold change in amino acid substitutions enriched in Fab-79D⁻ cells (positions 314–381). **f**, gRNAs targeting FLT3 codon N399 (left). Dark blue, NGG-PAM; grey, NGN-PAM; light blue, NRN-PAM. The PAM is indicated by the arrowhead. Right, plots of K562 reporter cells electroporated with base-editor plasmids (NG-ABE8e or SpRY-ABE8e). The percentage of 4G8⁻ cells is reported (gating is shown on the unedited sample). **g**, Affinity of therapeutic antibodies to receptor variants measured on K562 (for FLT3 (left) and CD123 (right)) or NIH-3T3 (for KIT (middle)) cells. Affinity curves fitted to the MFI of therapeutic monoclonal antibodies normalized to control monoclonal antibodies, after background subtraction. $n = 3$. Statistical analysis was performed using likelihood ratio tests. **h**, FLT3, SCF and IL-3 affinity. Cell lines expressing FLT3 (left), CD123 (right) and KIT (middle) receptor variants were incubated with fluorescent ligands and evaluated using flow cytometry. An affinity curve was fitted to the ligand's MFI. Statistical analysis was performed using likelihood ratio tests. $n = 3$ (FLT3 and CD123) and $n = 4$ (KIT).

(IL-3RA; also known as CD123) in HSPCs results in loss of antibody binding without gene knockout, therefore preserving physiological protein expression, regulation and intracellular signalling. Critically, this strategy enables targeting one or more genes that are fundamental for leukaemia survival, resulting in potent anti-leukaemia efficacy with minimal on-target/off-tumour toxicity.

## Base editing generates stealth receptors

FLT3 and proto-oncogene KIT are class III receptor tyrosine kinase expressed, either in the wild type or a mutated form, in 93% and 85% of AML cases, respectively[26–28]. CD123 is a type I cytokine receptor that is found in more than 75% of AML cases, including leukaema stem cells[5,29].

These genes are expressed at various stages of normal haematopoietic development and their overexpression on AML cells is associated with a higher incidence of relapse after HSCT and lower overall survival[28,30,31]. To develop our approach, we selected monoclonal antibodies currently studied for the development of AML immunotherapies: clone 4G8 (FLT3)[16,32], Fab-79D (KIT)[15,33] and 7G3 (CD123)[34,35]. 4G8 recognizes the FLT3 extracellular domain 4 (ECD4), whereas BV10A4—which binds to an unrelated epitope within ECD2—was used as control for FLT3 surface expression[32]. As 4G8 was generated by immunizing BALB/c mice with human FLT3-transfected cell lines, we reasoned that 4G8 recognizes a human-specific epitope, despite the high degree of homology and FLT3 ligand (FLT3L) cross-reactivity between human and mouse FLT3. We confirmed that the substitution of FLT3 ECD4 with its mouse orthologue (16 codon changes) results in a loss of 4G8 binding with preservation of FLT3L binding and kinase phosphorylation (Fig. 1a,b (left) and 1c (top)). To identify the minimal number of residues involved in 4G8 binding, we designed a Sleeping Beauty combinatorial library with human or mouse codons at the 16 mismatched positions (Fig. 1d (left)). Flow cytometry analysis of library-transduced cells revealed a 4G8⁻BV10A4⁺ population (Fig. 1d (centre)). Comparison of the relative codon abundance, by targeted deep sequencing of sorted 4G8⁻ and 4G8⁺ cells, revealed enrichment for a single amino acid substitution (N399D) (Fig. 1d (right) and Extended Data Fig. 1a). To validate this result, we transduced K562 cells with FLT3$^{N399D}$ and confirmed the loss of 4G8 binding despite FLT3 expression comparable to the wild type (Extended Data Fig. 1b). We next evaluated gene-editing strategies to introduce the N399D mutation. To easily evaluate the outcomes of genome engineering on cells that do not depend on FLT3 signalling, we generated K562 reporters expressing *FLT3* from the endogenous locus by targeted integration of a *EEF1A1* promoter upstream of the transcriptional start site (Extended Data Fig. 1c). We next confirmed that N399D can be inserted by homology-directed repair (HDR) using either *Streptococcus pyogenes* Cas9 (*Sp*Cas9) or *Acidaminococcus* Cas12a (*As*Cas12a) nucleases and 200 bp single-stranded oligodeoxynucleotide templates. Notably, the use of Cas nucleases resulted in *FLT3* knockout in a large proportion of non-edited cells (Extended Data Fig. 1d). As epitope engineering can be achieved by point mutations, we reasoned that base editing could be a suitable and safer option for epitope editing by avoiding double-stranded breaks (DSBs). The FLT3 asparagine residue at position 399, encoded by an AAC codon, can be converted to aspartate (GAC) or glycine (GGC) by adenine base editing (ABE). We tested this hypothesis by electroporating FLT3 reporters with sgRNAs in a 1 bp staggered manner (target A in position 3–9 of the spacer) in combination with advanced-generation TadA-8e deaminase, linked with *Sp*Cas9 nickase (NGG PAM) or variants with relaxed PAM specificity (NG-SpCas9n and SpRY-Cas9n) (Fig. 1f). Flow cytometry showed successful epitope editing with the loss of 4G8 recognition, with the highest efficiency achieved by SpRY-ABE8e + *FLT3*-sgRNA-18 (66.3%) (Fig. 1f). In contrast to the HDR-based strategy, non-edited cells retained FLT3 expression without significant knockout (Fig. 1f). As both N399D and N399G are potential outcomes of our base editing, we included these mutations in all further validation analysis.

A similar strategy was applied to the epitope mapping of Fab-79D, which binds to KIT ECD4 and blocks its ligand-induced dimerization[33]. We first confirmed the loss of Fab-79D binding by introducing ten orthologue amino acid changes at predicted contact points[33] (F316S, M318V, I319K, V323I, I334V, E360K, P363V, E366D, E376Q, H378R) and verified the preservation of stem cell factor (SCF) binding and ligand-induced kinase phosphorylation (Fig. 1b,c). To comprehensively screen the interaction between ECD4 and Fab-79D, we used a degenerated codon library in which each position of KIT ECD4 was substituted by a random amino acid (Fig. 1e (left)). We transduced HEK293T cells with the library and sorted KIT-expressing cells with either reduced or preserved Fab-79D staining (Fig. 1e (centre)). We found several mutations enriched in the Fab-79D⁻ fraction (Fig. 1e (right)) and selected those that could be introduced by base editing at ten positions identified by the library

(Met318, Ile319, Val323, Asp332, Ile334, Asp357, Glu360, Glu376 and His378). We selected H378R for further development, as it showed the best reduction in Fab-79D binding while preserving SCF affinity (Extended Data Fig. 1e). Furthermore, H378R can be inserted by ABE, similarly to FLT3$^{N399D}$, potentially enabling dual-epitope engineering. We next screened three sgRNAs aimed at His378 (A in position 5 to 7) in combination with SpRY-ABE8e and identified sgRNA KIT-Y as the best-performing candidate (Extended Data Fig. 1c,f).

For CD123, the epitope and amino acid substitutions affecting the binding of clone 7G3 (or its humanized counterpart, CSL362, talacotuzumab) had been previously reported[35]. We therefore designed a targeted base-editing screening on K562 reporter cells (Extended Data Fig. 1e) by testing sgRNAs aimed at positions Glu51, Tyr58, Ser59, Arg84, Pro88 and Pro89 of the CD123 N-terminal domain and rationally combined them with CBE (evo-APOBEC1-BE4 with NGG-, NG- and SpRY-Cas9) and ABE (NG- and SpRY-ABE8e) (Extended Data Fig. 1h). While several base-editing combinations reduced the affinity of 7G3 to CD123, only gRNAs CD123-N and -R with SpRY-ABE8e and gRNA CD123-L with SpRY-BE4 completely abrogated 7G3 binding (Extended Data Fig. 1h). Sequencing of edited cells revealed that ABE and CBE resulted in S59P and S59F, respectively. A bystander mutation (Y58H) was also introduced at higher efficiency with CD123-R than with -N by SpRY-ABE8e, and was therefore included in further validations. Owing to the possibility of editing multiple targets with ABE, we selected CD123-gRNA-R (S59P) for further development. Finally, by testing base-edited CD123 reporter cells with two additional CD123 monoclonal antibodies, we found that cytidine base editing of Pro88/Pro89 residues (giving rise to combinations of P88S/P89S or P88L/P89L) resulted in a loss of recognition by clones 6H6 and S18016E, therefore widening the pool of monoclonal antibodies with potential epitope-engineering applications (Extended Data Fig. 1i). To precisely quantify the monoclonal antibody affinity to epitope-edited receptors, we transduced K562 or NIH-3T3 cells with receptor variants and stained them with therapeutic monoclonal antibodies. We observed a near complete loss of binding of all of the tested variants up to saturating concentrations (>5,000 ng ml⁻¹) (Fig. 1g). Notably, FLT3$^{N399G}$ showed a reduction in 4G8 binding comparable to N399D. Similarly, addition of the Y58H bystander to CD123(S59P) had similar effects as S59P alone, without affecting CD123 surface expression.

We next assessed whether our engineering procedure could alter receptor functionality. Using fluorescent FLT3L, SCF and IL-3, we confirmed comparable ligand binding to WT or engineered receptors across all of the tested concentrations (1 to >1,000 ng ml⁻¹) (Fig. 1h). CD123 variants were co-expressed with the common β subunit (CSF2RB, also known as CD131) to form the heterodimeric IL-3 receptor and allow signal transduction (Extended Data Fig. 1j). Activation of intracellular signalling by FLT3 and KIT was confirmed by western blotting, which showed ligand-dependent kinase phosphorylation (Extended Data Fig. 2a,b). For CD123, we confirmed ligand-induced activation by measuring downstream phosphorylation of STAT5, which was equally activated in WT and engineered variants at all of the tested concentrations (Extended Data Fig. 2c). Finally, to confirm the preservation of the ligand-induced proliferative response, we performed a kinase complementation assay on BaF3 cells, which require mouse IL-3-mediated signalling for survival. We confirmed comparable and dose-dependent rescue of cell proliferation by WT and epitope-engineered receptors after exposure to human FLT3L, SCF and IL-3 during mouse IL-3 starvation (Extended Data Fig. 2d). Overall, we concluded that epitope engineering of functional FLT3, KIT and CD123 receptors is feasible and can be achieved with high efficiencies by selecting appropriate combinations of gRNAs and base-editing enzymes.

## Stealth receptors are resistant to CAR T cells

Recent preclinical studies have shown that CAR T cells generated from clones 4G8[16], Fab-79D[15] or CSL362[12,35] have considerable efficacy against

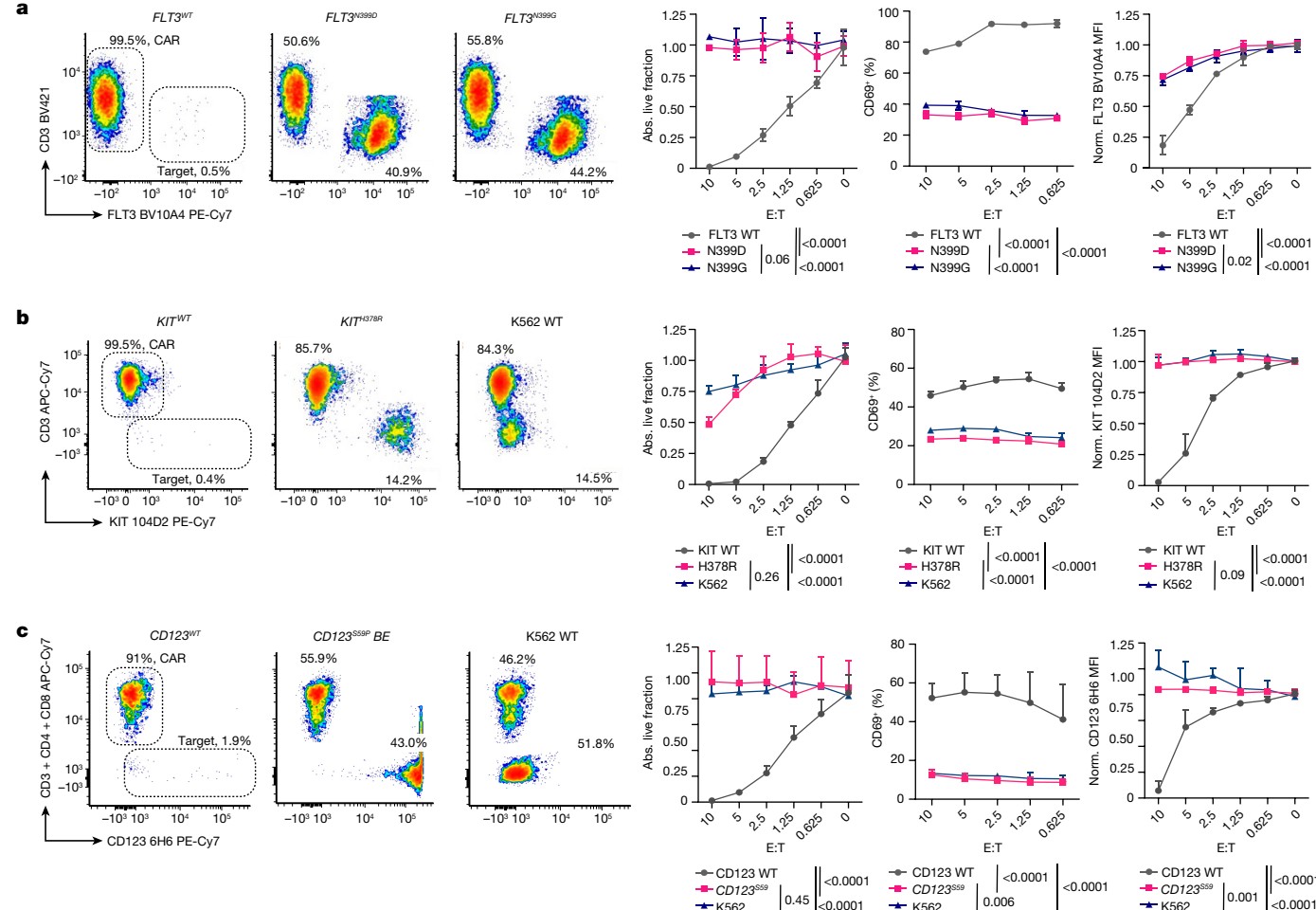

**Fig. 2 | Epitope-engineered variants are resistant to CAR T cells. a**, *FLT3^N399D* or *FLT3^N399G* cells avoid 4G8-CAR-mediated killing. K562 cells expressing FLT3 variants were cultured with 4G8-CAR at different effector:target ratios (E:T). Left, FACS analysis of K562 cells expressing either FLT3 WT, N399D or N399G, after 48 h of co-culture. T cells and targets are identified by CD3 and FLT3, respectively. From left to right, the fraction of live target cells (absolute (abs.) counts of annexin V⁻7-AAD⁻ cells) relative to E:T = 0; T cell activation by CD69⁺ (%) and surface expression of FLT3 by BV10A4 staining on residual live targets, normalized (norm.) to E:T = 0. The E:T ratio is reported on the *x* axis. Data are mean ± s.d. *n* = 4. Statistical analysis was performed using two-way analysis of variance (ANOVA); *P* values of the comparisons between each condition are reported. **b**, *KIT^H378R* cells avoid Fab-79D-CAR killing. Left, plots of K562 cells expressing wild-type (WT) KIT, H378R or unmodified (KIT⁻) (K562 WT) after 48 h of co-culture with Fab-79D CAR T cells. T cells and targets are identified by

CD3 and KIT, respectively. From left to right, the fraction of live target cells; T cell activation on the basis of CD69 staining; and surface expression of KIT by 104D2 on residual live targets, normalized to E:T = 0. The E:T ratio is reported on the *x* axis. Data are mean ± s.d. *n* = 4. Statistical analysis was performed using two-way ANOVA. **c**, *CD123^S59* base-edited cells are resistant to CSL362 CAR T cells. Left, representative plots of CD123-reporter cells, either unmodified (K562 WT) or base-edited, or CD123⁻ K562 cells after 48 h of co-culture with CSL362-CAR. T cells and targets are identified by CD3, CD4 and CD8, and CD123, respectively. From left to right, the fraction of live target cells; T cell activation on the basis of CD69 staining; and surface expression of CD123 on the basis of staining with the 6H6 control antibodies on residual target cells. The E:T ratio is reported on the *x* axis. Data are mean ± s.d. *n* = 6. Statistical analysis was performed using two-way ANOVA.

AML. To assess the resistance of epitope-engineered cells to CAR T cell therapy, we cloned the 4G8, Fab-79D and CSL362 single-chain variable fragments in second-generation CAR constructs with a CD28 costimulatory domain and used a bidirectional lentiviral vector to co-express an optimized[36] truncated *EGFR* selection/depletion marker (tEGFR) (Extended Data Fig. 2e). For CAR T cell production, we used stimulation with CD3–CD28 beads in the presence of IL-7 and IL-15 to impart a T stem memory phenotype[37] (Extended Data Fig. 3a,b). We obtained higher than 85% CAR transduction efficiency with greater than twentyfold in vitro expansion (Extended Data Fig. 2f). By performing in vitro killing assays with K562 reporters as targets, we found that, although the majority of cells expressing unmodified FLT3, KIT or CD123 were killed (<2% survival at an effector to target ratio (E:T) = 10 versus E:T = 0), cells expressing epitope-engineered variants were resistant to CAR-mediated killing (both in absolute counts and relative viability) and survived up

to experiment termination (Fig. 2a–c and Extended Data Fig. 3c,d). T cell activation (CD69) and degranulation (surface CD107a) were significantly higher in co-cultures with cells expressing WT genes, consistent with a lack of recognition of epitope-edited variants by the CAR (Fig. 2a–c and Extended Data Fig. 3c,d). Moreover, surviving target cells still expressed the receptors at levels comparable to the untreated controls (Fig. 2a–c (right)). Untransduced T cells did not show target killing or CD69 upregulation across all of the target conditions (Extended Data Fig. 3d). To further confirm epitope-specific killing by CAR T cells, we co-cultured mixed populations of dual-expressing cells (FLT3⁺CD123⁺ K562), either unmodified or base edited, and observed selective resistance of the epitope-edited populations when plated with the corresponding CAR T cells (Extended Data Fig. 3e). These data provide a stringent validation that cells overexpressing epitope-engineered FLT3, KIT and CD123 variants are resistant to CAR T cell recognition and killing.

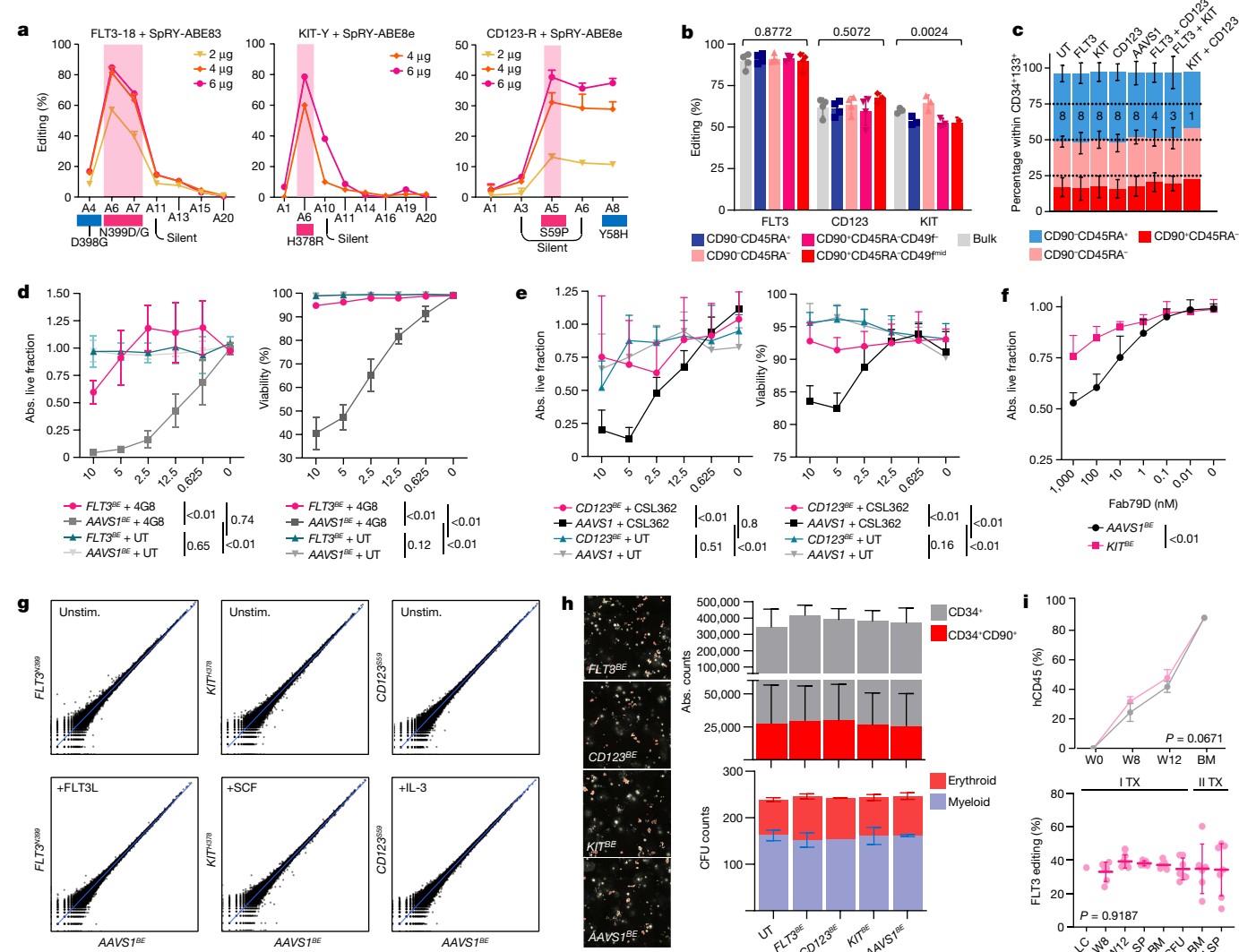

**Fig. 3 | Epitope editing does not affect stemness and differentiation of HSPCs. a**, The editing windows within FLT3, KIT and CD123 sgRNAs after electroporation with different doses of mRNA. Data are mean ± s.d. **b**, Editing efficiencies of bulk CD34+ or FACS-sorted subsets. The gating strategy is reported in Supplementary Fig. 2c. $n = 4$ biological replicates, $n = 3$ for KIT. Data are mean ± s.d. Statistical analysis was performed using one-way ANOVA. **c**, The CD90/CD45RA composition of epitope-edited HSPCs measured by FACS analysis within the CD34+CD133+ subset. Data are mean ± s.d. The sample size is reported within the bars. **d**, In vitro 4G8-CAR killing assay of $FLT3^{N399}$ edited HSPCs. The fraction of persisting live cells (left) and the viability on the basis of annexin V and 7-AAD staining (right) of CD34+CD45RA+ cells, 48 h after co-culture with 4G8-CAR or untransduced T cells at different effector:target (E:T) ratios. Data are mean ± s.d. $n = 4$. Statistical analysis was performed using two-way ANOVA. **e**, In vitro CSL362-CAR killing assay of $CD123^{S59}$ edited HSPCs. The fraction of persisting live CD90+ cells (left) and the viability of CD34+ cells on the basis of annexin V and 7-AAD staining (right), 48 h after co-culture with CSL362-CAR or untransduced T cells at different E:T ratios. $n = 8$ on 2 donors. Statistical analysis was performed using two-way ANOVA. **f**, The fraction of

absolute live CD34+ cells (relative to the no-antibody control) of $KIT^{H378}$ or $AAVS1^{BE}$ HSPCs after 48 h of culture with Fab-79D monoclonal antibodies. Data are mean ± s.d. $n = 6$ on 2 donors. Statistical analysis was performed using two-way ANOVA. **g**, RNA-seq analysis of epitope-edited CD34+ HSPCs with or without stimulation. log-scale scatter plot of the mean gene expression values of the $AAVS1$ control and $FLT3$-, $KIT$- and $CD123$-edited cells, either at the baseline (top) or after stimulation with FLT3L, SCF or IL-3 (bottom). $n = 3$ biological replicates. Unstim., unstimulated. **h**, The absolute counts of total CD34+ and CD90+CD45RA− cells ($n = 4$; top right) and of myeloid and erythroid colonies ($n = 2$; bottom right) of edited HSPCs. Left, representative colony-forming-unit (CFU) microphotographs. **i**, Primary and secondary xenotransplantation of $FLT3^{N399}$ or $AAVS1^{BE}$ HSPCs. Top, human engraftment (hCD45+) by flow cytometry analysis in primary recipients. Data are mean ± s.d. Statistical analysis was performed using two-way ANOVA. $n = 7$ ($FLT3^{N399}$) and $n = 4$ ($AAVS1^{BE}$). Bottom, FLT3 editing was measured on peripheral blood, haematopoietic organs (BM and spleen (SP)) or CFUs in primary and secondary transplants. Data are mean ± s.d. Statistical analysis was performed using one-way ANOVA. LC, liquid culture; SP, spleen; W, week.

## Efficient epitope editing of human HSPCs

To effectively introduce our variants into human HSPCs, we optimized a base-editing protocol on mobilized-peripheral-blood-derived CD34+ cells based on co-electroporation of sgRNAs and in vitro transcribed (IVT) SpRY-ABE8e mRNA (Fig. 3a and Extended Data Fig. 4a,b). After optimization of the mRNA IVT, culture, electroporation conditions and editing timepoint (Extended Data Fig. 4c–e), we achieved up to 86.6%,

78.6% and 78.9% editing efficiency of the target adenines within the windows of FLT3-18, KIT-Y and CD123-R sgRNAs (Fig. 3a). Contrary to previous observations with HDR-mediated editing[38], base-editing efficiencies were similar in bulk cells and in more primitive, HSC-enriched subsets (Fig. 3b). Analyses of edited cells showed no skewing of the composition of phenotypically identified progenitors (lymphoid-primed multipotent progenitor (LMPP)-like, CD90−CD45RA+; multipotent progenitor (MPP)-like, CD90−CD45RA−; haematopoietic stem cell

(HSC)-like, CD90⁺CD45RA⁻) compared with the control during in vitro culture (Fig. 3c and Extended Data Fig. 4f,h). To assess the functionality of the receptors, edited HSPCs were cultured with increasing concentrations of the respective ligand without other cytokines. We observed dose-dependent HSPC expansion for all three targets, with no difference between receptor-edited and controls except for $CD123^{S59}$ cells at IL-3 concentrations of 1–10 ng ml⁻¹ (Extended Data Fig. 4i). Nonetheless, we did not observe counter-selection of base-edited HSPCs, confirming uniform expansion of CD34⁺ cells regardless of CD123 editing (Extended Data Fig. 4j). To confirm the resistance of $FLT3^{N399}$ and $CD123^{S59}$ engineered HSPCs, we performed killing assays by plating edited HSPCs with 4G8 and CSL362 CARs. Specific killing by FLT3- and CD123-CAR was most pronounced within the CD45RA⁺ and the CD90⁺ subsets, respectively, which were therefore used to evaluate the outcome of these experiments. Whereas cells edited at a control site (AAVS1 safe genomic harbour) were eliminated by CAR T cell co-culture, epitope-edited cells showed higher viability and absolute counts (Fig. 3d,e). As KIT has known extrahaematopoietic expression in humans[39] we focused on the use of monoclonal antibodies instead of CAR T cells, which might result in less severe on-target toxicities. By plating edited HSPCs with increasing concentrations of the dimerization-blocking Fab-79D monoclonal antibody, we observed preserved expansion of $KIT^{H378}$ HSPCs in response to SCF, whereas cells expressing base-edited AAVS1 ($AAVS1^{BE}$) were inhibited in a dose-dependent manner (Fig. 3f). These data show that epitope-engineered HSPCs can efficiently be generated by ABE and become resistant to targeted immunotherapies.

## Epitope editing preserves HSPC function

To evaluate the transcriptional changes associated with epitope editing, we performed RNA-sequencing (RNA-seq) analysis of CD34⁺ HSPCs edited for *FLT3, CD123, KIT* or *AAVS1*, either stimulated or unstimulated with the respective ligands. We found 78, 2,667 and 7,944 differentially expressed genes associated with FLT3L, SCF and IL-3 stimulation, respectively (Extended Data Fig. 5a–e). By comparing receptor-edited conditions with $AAVS1^{BE}$, we confirmed the absence of transcriptional differences both at the baseline and after stimulation (Fig. 3g and Supplementary Tables 3–5). Phospho-proteomic profiling by mass spectrometry analysis of edited CD34⁺ HSPCs showed concordant changes of differentially phosphorylated sites after ligand stimulation between the receptor-edited cells and the AAVS1 control cells (Extended Data Fig. 5f), again confirming in an unbiased manner the activation of downstream signalling by epitope-modified receptors. To further corroborate the minimal impact on the differentiation ability of receptor-edited HSPCs, we performed expansion culture and a colony-forming assay and observed comparable absolute cell counts and numbers of myeloid and erythroid colonies (Fig. 3h and Extended Data Fig. 4g). In vitro differentiation of CD34⁺ HSPCs towards myeloid, macrophage, classical dendritic, granulocytic and megakaryocytic lineages was similar irrespective of the editing condition and did not result in counterselection of edited cells (Extended Data Fig. 6a). Functional assays of lineage-differentiated cells showed similar results across all of the conditions, including reactive oxygen species production by myeloid cells, *Escherichia coli* phagocytosis by macrophages, M1/M2-like macrophage polarization, phospho-flow profiling of IL-4-, PMA/ionomycin-, GM-CSF-, IFN type-I-, IL-6- and LPS-stimulated myeloid cells, HLA class II/CD86 upregulation by dendritic cells, induction of granulocyte NETosis and generation of hyperdiploid megakaryocytes (Extended Data Fig. 6b–i). Xenotransplantation of $FLT3^{N399}$ HSPCs into female immunodeficient mice (NBSGW) showed preserved engraftment, repopulation and multilineage differentiation capacity (Fig. 3i and Extended Data Fig. 4k), similar to the AAVS1 controls. FLT3 editing of engrafted cells was comparable to input cells (35.5%) and stable up to 13 weeks after transplant (Fig. 3l), confirming successful editing of the most primitive HSPC subset. Transplantation of bone marrow (BM)

cells into secondary recipients resulted in high human engraftment and no differences in lineage distribution up to 17 weeks (Extended Data Fig. 7a,b). Again, *FLT3* editing levels remained comparable to primary recipients (Fig. 3i). As mouse FLT3L is cross-reactive with human FLT3, these results further confirm the functionality of $FLT3^{N399}$ HSPCs. Similarly, in vivo repopulating ability and multilineage differentiation of $KIT^{H378}$ and $CD123^{S59}$ edited HSPCs was comparable to $AAVS1^{BE}$ controls in both primary and secondary recipients, with no skewing of lineage differentiation or counterselection of edited cells (Extended Data Fig. 7c–k). Overall, these data confirm that epitope editing in HSPCs does not affect receptor signalling, stem cell differentiation ability and the functionality of their lineage-specific progenies.

## Off-target effects of epitope editing

As the use of SpRY-Cas9 might lead to potential off-target effects, we performed a specificity analysis by combining genome-wide, unbiased identification of off-target sites (GUIDE-seq) and in silico off-target prediction. We performed genome-wide, unbiased identification of DSBs enabled by sequencing (GUIDE-seq) using SpRY-nuclease (Extended Data Fig. 8a) and found that, for gRNA mismatches + bulge < 6, all identified off-target sites were in non-coding genomic regions (12 intronic and 11 intergenic) (Extended Data Fig. 8a and Supplementary Table 16). As a complementary approach, we characterized the top exonic and intronic in silico predicted off-target sites for *FLT3* ($n = 12$), *CD123* ($n = 9$) and *KIT* ($n = 12$) sgRNAs and assessed the levels of undesired deamination on base-edited HSPCs using targeted deep sequencing (Extended Data Fig. 8b and Supplementary Tables 17–19). No significant deamination was observed at CD123 or KIT off-target sites (Extended Data Fig. 8b), whereas four FLT3-sgRNA-18 loci showed comparatively higher deamination. Only one of these off-target sites was in an exonic region, but the affected gene, *SNTG1* (syntrophin-γ1), encodes a brain-specific syntrophin-family protein with no expression (Supplementary Table 22) or known functional role in haematopoietic tissue[40,41]. Despite this generally safe profile, we found that an alternative gRNA (2 nucleotides upstream of FLT3-18, AGA PAM) in combination with a more PAM-restricted Cas variant (SpG) avoids deamination at predicted off-target sites while preserving around 90% of on-target activity compared with the FLT3-18 sgRNA (70.6% versus 81% editing in HSPCs) (Extended Data Fig. 8b and Supplementary Table 20). To assess the occurrence of non-gRNA-dependent deamination, we examined our RNA-seq dataset generated on CD34⁺ HSPCs and observed no significant A>G conversion on transcripts with high sequencing coverage (top 5%) compared with the controls (Extended Data Fig. 8c). Finally, the rate of on-target indels was estimated to be ≤1.2% for all target loci (0.6, 1.2 and 0.2% for *FLT3*, *CD123* and *KIT*, respectively) (Extended Data Fig. 8d), consistent with previously reported data for ABE[42,43]. Overall, these data support a generally safe genotoxicity profile of FLT3, CD123 and KIT epitope editing in CD34⁺ HSPCs.

## $FLT3^{BE}$ confers resistance to 4G8 CAR in vivo

To assess whether FLT3 CAR T cells can effectively eliminate AML while sparing FLT3-edited haematopoiesis, we sequentially engrafted NBSGW mice with CD34⁺ HSPCs (either $FLT3^{BE}$ or $AAVS1^{BE}$) and a human patient-derived AML xenograft (PDX-1), characterized by MLL-AF9 and FLT3-ITD, previously transduced with a reporter gene to facilitate its detection within the mixed haematopoiesis (Extended Data Fig. 7l,m and Supplementary Fig. 4c). Then, 10 days after the PDX challenge, the mice were treated with 4G8 CAR T cells and, after an additional 14 days, the composition of haematopoietic organs was analysed using flow cytometry (BM and spleen) (Fig. 4a). As observed in previous experiments, with no AML or CAR T cell administration, both editing groups showed similar peripheral blood composition (Extended Data Fig. 9a), editing levels comparable to input cells (around 85%), which

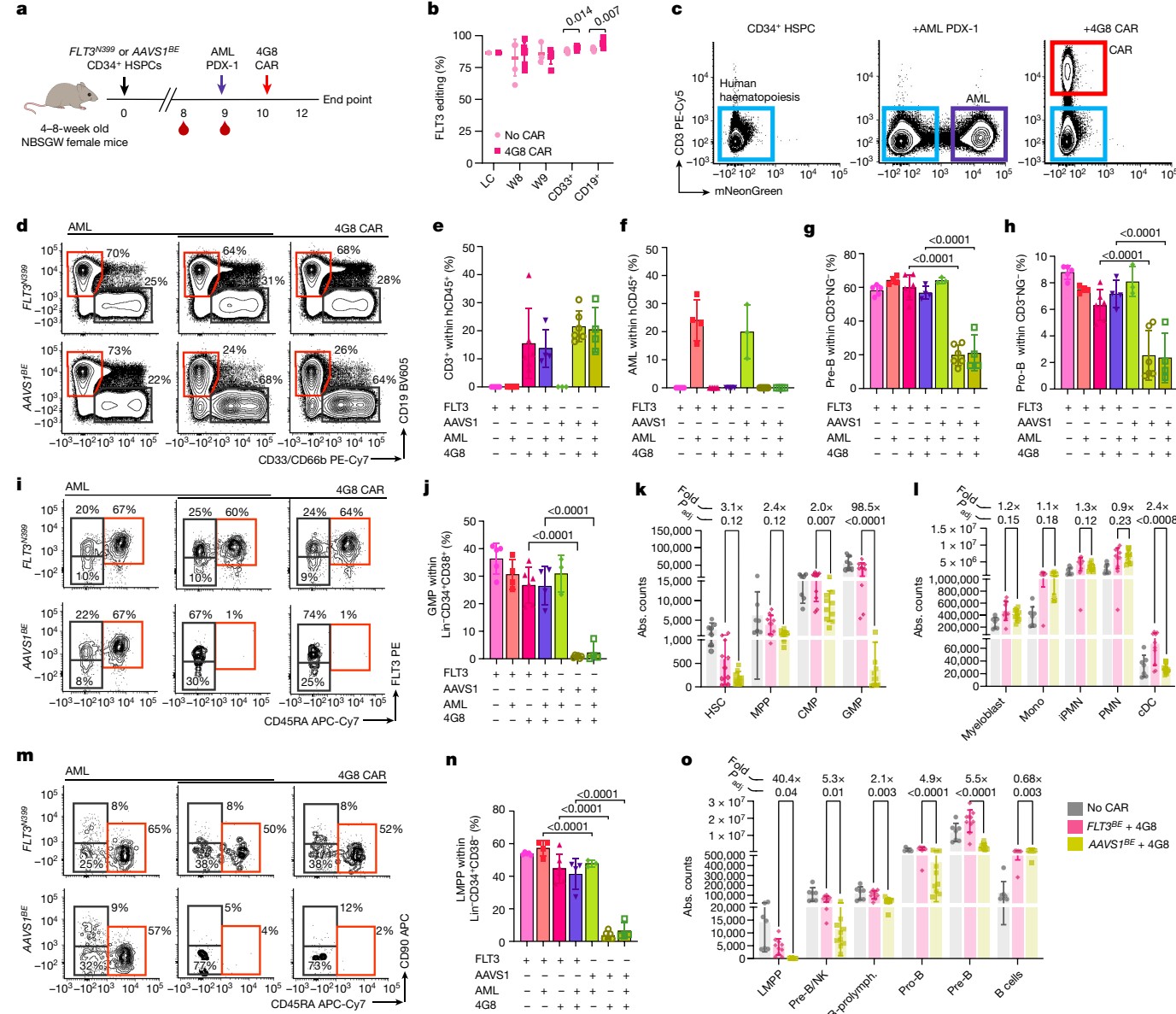

**Fig. 4 | 4G8-CAR eradicates AML PDX while sparing *FLT3^N399* HSPC progeny.**
**a**, Co-engraftment experiments of *FLT3^N399* or *AAVS1^BE* HSPCs with AML PDX-1, treated with 4G8-CAR. **b**, FLT3 editing was measured on liquid culture, peripheral blood (weeks 8–9) and sorted CD33⁺CD19⁺ BM cells. Statistical analysis was performed using multiple unpaired *t*-tests. Data are mean ± s.d. **c**, Representative FACS plots of haematochimeric mouse BM, pregated on hCD45⁺. CAR T cells and AML were identified by CD3⁺ and mNeonGreen⁺, respectively. Data are mean ± s.d. **d**, FACS plots, gated on hCD45⁺CD3⁻mNeonGreen⁻, showing the depletion of B cells by 4G8-CAR. NG, mNeonGreen. **e**, The percentage of CD3⁺ cells within hCD45⁺mNeonGreen⁻ BM cells. Data are mean ± s.d. Statistical analysis was performed using one-way ANOVA. **f**, The percentage of AML cells within hCD45⁺CD3⁻ BM cells. Data are mean ± s.d. **g,h**, The percentages of pre-B (CD19⁺CD10⁺CD34⁻) (**g**) and pro-B (CD19⁺CD10⁺CD34⁺) (**h**) cells among hCD45⁺CD3⁻mNeonGreen⁻ cells. Data are mean ± s.d. Statistical analysis was performed using one-way ANOVA, comparing *FLT3^N399* versus *AAVS1^BE*. **i**, The composition of lineage⁻CD34⁺CD38⁺CD10⁻ progenitors was analysed using flow cytometry. Cells were defined as follows: GMPs (CD45RA⁺FLT3⁺), CMPs (CD45RA⁻FLT3⁺) and mega-erythroid progenitors (MEPs, CD45RA⁻FLT3⁻). **j**, The GMP percentage within lineage⁻CD34⁺CD38⁺. Data are mean ± s.d. **k,l**, Absolute counts of progenitors (**k**) and myeloid subsets (**l**) in the BM. Untreated mice were pooled together (grey bars), and 4G8-treated *FLT3^N399* and *AAVS1^BE* mice are reported in pink and yellow, respectively. Data are mean ± s.d. The fold change in absolute counts (*FLT3^N399*/*AAVS1*) for CAR-treated groups is reported. Statistical analysis was performed using one-way ANOVA comparing the *FLT3^N399* versus *AAVS1^BE* conditions treated with 4G8-CARs. **m**, The composition of lineage⁻CD34⁺CD38⁻CD10⁻ progenitors was determined using flow cytometry. Cells were defined as follows: HSCs (CD45RA⁻CD90⁺), MPPs (CD45RA⁻CD90⁻) and lymphoid-primed MPPs (LMPPs, CD45RA⁺CD90⁻). **n**, The LMPP percentage within lineage⁻CD34⁺CD38⁻ cells. Data are mean ± s.d. Statistical analysis was performed using one-way ANOVA. **o**, The absolute counts of lymphoid subsets. The fold change in absolute counts (*FLT3^N399*/*AAVS1*) for CAR-treated groups is reported. Data are mean ± s.d. Statistical analysis was performed using one-way ANOVA comparing the *FLT3^N399* versus *AAVS1^BE* conditions treated with 4G8-CAR.

remained stable over time, and a lack of differences within the BM myeloid and lymphoid subsets (sorted CD33⁺ and CD19⁺ cells, respectively) (Fig. 4b). Mice treated with 4G8-CAR showed robust CAR T cell engraftment (13.7% and 20.4% CD3⁺ in the *FLT3^BE* and *AAVS1^BE* groups, respectively) and AML eradication in the BM and spleen (Fig. 4c,e,f

and Extended Data Fig. 9b). As expected, we observed a significant increase in the fraction of *FLT3^N399* cells in the BM of CAR-treated mice (88% versus 90% within myeloid and 89% versus 94% within lymphoid cells) (Fig. 4b). Multiparametric flow cytometry analysis of the BM revealed relative depletion of CD19⁺ subsets (precursor B (pre-B) cells

and progenitor B (pro-B) cells) only in the *AAVS1^BE* group treated with 4G8-CAR, whereas mice engrafted with *FLT3^N399* HPSCs were protected (Fig. 4d,g,h). Within differentiated myeloid cells (excluding AML), the proportion of immature granulocytes was reduced in *AAVS1^BE* compared with in *FLT3^N399* mice (Extended Data Fig. 9c). *FLT3^N399* cells conferred selective resistance to lineage-negative progenitor cells (Extended Data Fig. 9d,e) and in particular to granulo-mono progenitors (GMPs) and LMPPs, which were nearly completely eliminated in the *AAVS1^BE* group (1.4% versus 26.6% GMP and 4.8% versus 43.3% LMPP in *AAVS1^BE* versus FLT3^N399, respectively) (Fig. 4i,j,m,n). To more precisely identify haematopoietic subsets depleted by 4G8-CAR—and selectively protected by epitope engineering—we compared the absolute counts of BM lineages between treated groups. Absolute counts of common myeloid progenitors (CMPs), classical dendritic cells (cDCs) and GMPs were reduced in *AAVS1^BE* mice (CMPs (0.48×), GMPs (0.01×) and cDCs (0.41×), in *AAVS1^BE* versus *FLT3^N399* mice) (Fig. 4k) compared with in *FLT3^N399* mice. Within lymphoid subsets, LMPPs, pre-B/natural killer (NK) cells and downstream populations (B cell prolymphocytes (B-prolymphocytes), pro-B cells and pre-B cells) were protected by *FLT3^N399* (LMPPs (0.02×), pre-B/NK cells (0.19×), pro-B cells (0.2×) and pre-B cells (0.18×) in *AAVS1^BE* versus *FLT3^N399* mice) (Fig. 4o). A decrease in HSC number that was more pronounced in the *AAVS1^BE* group (0.32× compared with *FLT3^N399* mice), and an increase in the number of mature B cells were observed in CAR-treated versus untreated conditions. These differences probably reflect a non-CAR-specific effect, as humanized mice treated with untransduced T cells showed a similar reduction in HSC numbers, an increase in monocytes and a trend towards mature B cell expansion (Extended Data Fig. 7n). The FLT3 median fluorescence intensity (MFI) of persisting pre-B/NK cells, B-prolymphocytes, pro-B cells and pre-B cells, monocytes and myeloblasts in *AAVS1^BE* mice exposed to 4G8-CAR was lower than in the same populations of *FLT3^N399* edited mice (Extended Data Fig. 9f), providing additional evidence that *FLT3^N399* cells can retain FLT3 expression while avoiding CAR-mediated killing. Notably, while CAR T cells detected at the end of the experiment showed a similar phenotype (mostly effector and central memory) and CD69 expression in all groups, those exposed to *FLT3^N399*-cell haematopoiesis displayed a reduced expansion and a significant reduction in PD-1 expression compared with the *AAVS1^BE* group (Extended Data Fig. 9g), suggesting an overall decrease in activation/exhaustion associated with the lower antigen burden to which the CAR T cells were exposed. Importantly, *FLT3^N399* epitope editing provided the same protection against 4G8-CAR killing regardless of the presence of human PDXs, highlighting the possibility of selectively eliminating AML cells while preserving haematopoietic reconstitution. Overall, these data confirmed that, in the NBSGW model, FLT3⁺ CAR T cells preferentially deplete B cells and progenitor subsets (GMPs, LMPPs), while *FLT3^N399* epitope editing confers protection to these subpopulations.

## *CD123^BE* haematopoiesis is resistant to CD123 CAR T

As done for *FLT3* editing, we transplanted *CD123^S59* HSPCs into NBSGW mice and confirmed engraftment and multilineage repopulation comparable to *AAVS1^BE* HSPCs, with a high and stable fraction of edited cells (Extended Data Figs. 9i and 10a,c). Transplanted mice were then injected with PDX-1—which also expresses CD123 (Extended Data Fig. 7m)—and, after 10 days, were treated with CSL362 CAR T cells. Similar to 4G8-CAR therapy, CSL362 CAR T cells nearly completely eradicated AML cells and displayed higher expansion in mice that were engrafted with *AAVS1^BE* HSPCs compared with *CD123^S59* HSPCs (Extended Data Fig. 10b,d,e). Flow cytometry analysis of BM highlighted significantly lower absolute counts of human haematopoietic cells (after exclusion of AML and CAR T cells) (Extended Data Fig. 10f) and depletion of myeloid cells, including mature and immature granulocytes, in mice that were transplanted with *AAVS1^BE* HSPCs, while the progeny of *CD123^S59* HSPCs was protected (Extended Data Fig. 10h–k).

Differently from the killing pattern observed with 4G8-CARs, within the lymphoid lineage, only the percentage of pro-B cells showed a trend towards a reduction in CSL362 CAR-T-cell-treated *AAVS1^BE* mice (Extended Data Fig. 10g). Dendritic cells, including CD123^high plasmacytoid dendritic cells (pDCs), were depleted by CSL362 CAR T cell treatment, while they were preserved in the *CD123^S59* group (Extended Data Fig. 10l–n). Similar to *FLT3* editing, lineage⁻CD34⁺ progenitors were protected in the *CD123^S59* group (Extended Data Fig. 10o,p). Absolute counts of myeloid populations (CMPs, GMPs, myeloblasts, granulocytes and dendritic cells) were significantly reduced in the *AAVS1^BE* group versus the *CD123^S59* group (Extended Data Fig. 10q,r). Among lymphoid cells, when comparing *AAVS1^BE* versus CD123^S59, we observed a partial depletion of B-prolymphocytes to mature B cells. As observed for FLT3, the CD123 MFI of persisting GMPs, myeloblasts, monocytes, cDCs and pDCs was higher in *CD123^S59* versus *AAVS1^BE* mice treated with CAR T cells (Extended Data Fig. 11a). Overall, these data show a reduction in on-target toxicity induced by CD123 CAR T cells on the haematopoiesis of *CD123^S59* epitope-edited cells, which would otherwise result in depletion of myeloid subsets and dendritic cell and an overall reduction in absolute counts of haematopoietic cells.

## Multiplex editing enables AML eradication

We reasoned that editing two or more epitopes might enable more effective immunotherapies by enabling simultaneous targeting of multiple AML antigens. To obtain proof-of-concept of protection from multiple CAR T cell therapies, we co-edited two (FLT3 and CD123) or three (FLT3, KIT and CD123) targets on dual- or triple-reporter K562 cells and co-cultured sorted single-, double- or triple-edited cells with bi- or tri-specific CAR T cells, respectively. After 2 days, only the double- or triple-epitope-edited cells survived dual- or triple-specificity CAR-mediated killing without inducing T cell activation (CD69), degranulation (CD107a), proliferation (CellTrace dilution) or cytokine secretion (IFNγ and TNF), while still expressing FLT3, CD123 and KIT (Fig. 5b,c and Extended Data Figs. 11e and 12a–c). To assess the feasibility of multiplex HSPC base editing, we electroporated CD34⁺ cells with SpRY-ABE8e and two sgRNAs and observed editing efficiencies comparable to single-base editing, without an increase in cellular toxicity (Fig. 5a and Extended Data Fig. 4e,g). To model the increased efficacy conferred by dual-targeting therapies, we identified an AML PDX (PDX-2) (Extended Data Fig. 7l,m) that is not effectively eradicated by 4G8-CAR (Extended Data Fig. 11b,c). When PDX-2 was transplanted in haematochimeric mice, 4G8-CARs showed a trend towards higher anti-leukaemia efficacy in *FLT3^N399*-HSPC-engrafted mice compared with the control (Extended Data Fig. 11c), suggesting a detrimental role of off-tumour CAR activation for therapeutic success. We next tested whether combinations of CAR T cells targeting two antigens could provide superior elimination of PDX-2 and found that both 4G8-CAR plus Fab-79D or CSL362-CAR were more effective (Extended Data Fig. 11d). To obtain formal proof that dual-edited HSPCs are resistant to a combined CAR T cell therapy, we transplanted dual *FLT3^N399CD123^S59* epitope-edited HSPCs or *AAVS1^BE* controls into NBSGW mice. After confirmation of multilineage engraftment (Extended Data Fig. 11f) and injection of the AML PDX-2 cells, we treated the mice with 4G8 and CSL362 CAR T cells (1:1 co-infusion) (Fig. 5d). As suggested by our CAR-combination experiment, dual CAR T cell therapy was able to fully eradicate AML cells from the BM and spleen (Fig. 5d,e). *FLT3^N399CD123^S59* edited cells persisted until the end point and were equally detected within lymphoid and myeloid cells (Fig. 5f). The selective pressure imposed by CAR T cell treatment increased the fraction of *FLT3^BE* cells within the lymphoid and myeloid subsets and, to a lesser degree, of *CD123^BE* cells within myeloid cells (as expected by the expression pattern of their respective target). By analysing the non-malignant haematopoiesis, we observed widespread reduction in myeloid and lymphoid haematopoietic lineages in CAR-T-cell-treated *AAVS1^BE* mice compared

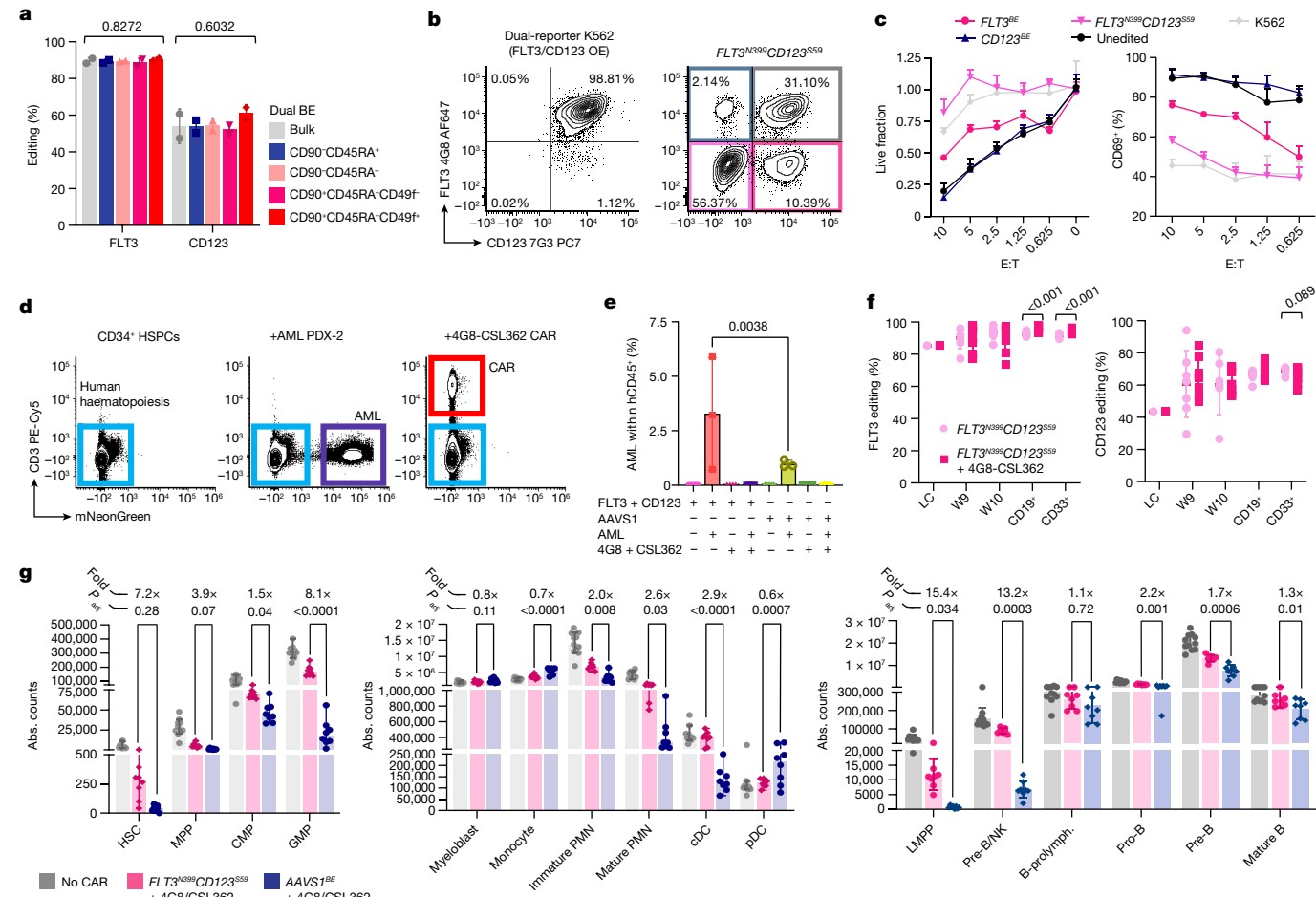

**Fig. 5 | Dual FLT3/CD123 epitope editing enables more effective AML immunotherapies. a**, The efficiency of multiplex $FLT3^{N399}CD123^{S59}$ editing measured on bulk HSPCs or FACS-sorted subsets. Data are mean ± s.d. Statistical analysis was performed using one-way ANOVA. **b**, Representative plots of dual FLT3/CD123 reporter cells showing loss of recognition by 4G8 and 7G3 monoclonal antibodies after multiplex editing. OE, overexpressed. **c**, Single- or dual-edited K562 reporters were FACS-sorted and co-cultured with dual-specificity FLT3/CD123 CAR T cells at different effector:target (E:T) ratios. The fraction of live target cells (left) and T cell activation (CD69⁺; right) are shown. Statistical analysis was performed using two-way ANOVA; the comparisons are reported in Supplementary Table 24. **d**, Representative plots of BM samples from mice engrafted with HSPCs only, HSPCs + PDX-2 or HSPCs + PDX-2 treated with 4G8/CSL362 CAR T cells. Plots were pregated on total hCD45⁺; CAR T cells were identified by CD3 and AML PDXs by mNeonGreen. **e**, The percentage of PDX cells within hCD45⁺CD3⁻ BM cells of mice transplanted and treated as

indicated. Data are mean ± s.d. Statistical analysis was performed using one-way ANOVA with correction for multiple comparisons. **f**, $FLT3$ (left) and $CD123$ (right) editing efficiencies measured on peripheral blood (week 9–10) and sorted CD33⁺ and CD19⁺ BM fractions at the end point. Data are mean ± s.d. Statistical analysis was performed using multiple unpaired $t$-tests. **g**, The absolute counts of progenitors (left) and myeloid (middle) and lymphoid (right) subsets in the BM. Untreated mice were pooled (grey bars), CAR-treated $FLT3^{N399}CD123^{S59}$ mice are shown in pink and CAR-treated $AAVS1^{BE}$ mice are shown in blue. The fold change in the absolute counts ($FLT3^{N399}CD123^{S59}/AAVS1$) for CAR-treated groups is reported above each population. Data are mean ± s.d. Statistical analysis was performed using one-way ANOVA. False-discovery rate (FDR)-adjusted $P$ values of the comparison between the $FLT3^{N399}CD123^{S59}$ and $AAVS1^{BE}$ conditions treated with CAR T cells are reported. Full counts are provided in Supplementary Table 23. PMN, polymorphonucleate granulocytes; prolymph., prolymphocytes.

within the $FLT3^{N399}CD123^{S59}$ group, with the strongest depletion of GMPs, granulocytes, cDCs, LMPPs and pre-B/NK, pro-B and pre-B cells and a less pronounced but significant reduction across nearly all of the other subpopulations, suggesting a depletion pattern in between what we had observed with 4G8 and CSL362 CARs alone (Fig. 5g). Overall, these data provide a proof of concept that multiplex epitope editing can be efficiently obtained in HSPCs and enables potent multitarget immunotherapies with reduced overlapping on-target/off-tumour haematopoietic toxicities.

## Discussion

Our studies show that tumour-associated antigens shared by normal tissue can be safely targeted by precisely modifying the epitope recognized by adoptive immunotherapies in healthy cells, endowing them

with selective resistance and generating artificial leukaemia-restricted antigens. Different from approaches aimed at the removal or truncation of the target molecule, which can be applied only to dispensable genes (such as $CD33$ and $CLEC12A$ (also known as $CLL-1$)), epitope engineering enables the targeting of genes that are essential for leukaemia survival regardless of their function in healthy haematopoiesis. Cytokine receptors are important signalling mediators in AML and their constitutive activation or overexpression is a major driver of leukaemia expansion and self-renewal. Tyrosine kinase inhibitors have demonstrated therapeutic efficacy in patients with AML (particularly for FLT3-ITD AML)[6], underscoring the relevance of these receptors for leukaemia persistence. Although several agents targeting FLT3, CD123 or KIT are under development, preclinical and clinical reports have highlighted the risk of myelosuppression due to on-target/off-tumour toxicity, and this issue is expected to be exacerbated when a prolonged anti-leukaemia

effect is desirable or when multiple antigens are targeted. Here we show that the introduction of a single amino acid substitution within FLT3, CD123 and KIT ECDs is sufficient to abrogate their recognition by monoclonal antibodies and CAR T cells without affecting protein function. Although we cannot exclude that these point mutations could be immunogenic, in the allo-HSCT setting, the immunosuppressive graft-versus-host disease (GvHD) prophylaxis is expected to confer tolerance towards the engineered proteins. The innovative tools developed in this study could increase the therapeutic index of AML immunotherapies and enable long-term anti-leukaemia maintenance, thanks to CAR T cell persistence or repeated administration of antibody–drug conjugates or bi-specific T cell engagers. Moreover, by restricting the on-target activity to leukaemia cells, epitope-editing can reduce the antigen burden to which CAR T cells are exposed. This has the additional benefit of decreasing undesired CAR T cell stimulation and possibly excessive cytokine release or exhaustion, as suggested by the reduced PD-1 levels observed in vivo. Finally, we show that epitope engineering by base editing can easily be multiplexed to enable combination therapies with synergistic on-tumour effects while avoiding overlapping toxicities. Targeting multiple molecules could eliminate AML with heterogeneous or low-level antigen expression and further reduce the risk of relapse. One possible caveat of this approach could be the reduction in haematopoietic clonality after immunotherapy-mediated elimination of non-edited cells. We have shown that mRNA electroporation enables highly efficient editing of HSCs, with no overt effect on stem cell functionality. On the basis of previous observations of clonality in gene therapy trials[44], we expect that these efficiencies will provide a safe margin for effective engraftment and multilineage reconstitution. Moreover, base editing, which does not require DSBs nor template delivery, is inherently safer compared with nuclease-mediated gene inactivation or HDR, which have been associated with the induction of translocations or other genomic abnormalities[45]. Similarly, exogenous DNA templates bear the risk of unspecific genomic insertion[46] and require strategies to overcome the detrimental activation of DNA damage responses[47,48]. Our data show minimal risk of introducing bystander mutations or converting a fraction of the DNA nick into DSBs. Although we cannot formally exclude that base editing could induce additional unpredicted on-target mutations, the regions affected by base editing are distant from mutational hotspots of cancer-associated variants (that is, the tyrosine kinase domain for KIT and FLT3 or the juxta-membrane domain for FLT3). Thus, the main concern would be a possible loss of function in a fraction of the treated cells, which are likely to be spontaneously counter-selected due to reduced fitness. Despite their overall safety, the potential for off-target deamination by base-editing tools warrants consideration. Although we observed a relatively safe profile for our sgRNAs coupled with PAM-relaxed base editors, a more comprehensive assessment of the specificity of gRNA- and non-gRNA-dependent off-target activity will be required before moving towards clinical translation. Nevertheless, the risk of single-base mutation in relevant coding regions is relatively minor, especially if compared with the genotoxic risk of conventional chemotherapy or the risk of disease progression itself. Risk–benefit evaluation will probably require case-by-case considerations, including the probability of relapse and the availability of alternative therapeutic options. Overall, we envisage a straightforward path to clinical translation, given the growing clinical experience with HSPCs genome editing strategies and the fact that immunotherapies based on our selected antibodies have already reached clinical testing (NCT02789254; NCT02642016; talacotuzumab). HSCT is widely used for the treatment of patients with AML, but their long-term survival hinges entirely on the remission status before HSCT, disease biology and the delicate balance between graft versus leukaemia and graft-versus-host disease. In the presence of post-transplant minimal residual disease or relapse, these patients are left with few treatment options. Our epitope-editing strategy could be rapidly implemented in current allo-HSCT protocols to enable minimal residual disease eradication or anti-leukaemia maintenance after HSCT. Moreover, using HSPCs from healthy donors will avoid the risk of inadvertently editing residual host AML. A paradigmatic setting for first-in-human testing could involve CAR T cell administration at the time of haematopoietic reconstitution after HSCT of relapsed/refractory or high-risk patients to prevent early relapse. Another option, also proposed by others[49], would take advantage of the CAR T cell therapy itself as myeloablative conditioning before HSCT to concomitantly kill leukaemic cells and free the BM niche. Eliminating chemotherapy or radiotherapy could also minimize the risk of graft-versus-host disease due to reduced tissue damage and release of pro-inflammatory mediators. While we focused on advancing AML treatment, we expect that epitope engineering can be applied to other haematological malignancies characterized by the paucity of safely actionable antigens, such as T lymphoblastic leukaemia or relapsed CD19⁻ B cell acute lymphoblastic leukaemia. Moreover, as KIT mutations have been implicated in small-cell lung carcinoma, melanoma, colorectal cancer and more than 80% of gastrointestinal stromal tumours[39], autologous transplant of engineered HSPCs could theoretically improve the therapeutic index of KIT-directed therapies also for non-haematological malignancies. Finally, immunotherapies aimed at HSPC-specific markers have recently been proposed as non-genotoxic conditioning for non-malignant diseases[50]. In this setting, epitope-edited HSPCs would remove the limitations imposed by drug pharmacokinetics and reduce the time of aplasia while increasing HSPC engraftment. Moreover, if epitope editing is coupled with other therapeutic base-editing approaches, progressive in vivo enrichment of genetically modified autologous cells can be achieved by infusion of monoclonal antibodies, antibody–drug conjugates or bi-specific T cell engagers. In conclusion, epitope editing of HSPCs can enable safer and more effective immunotherapies when on-target/off-tumour toxicities are the key limiting factor to successful clinical translation.

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

# Methods

## Sleeping Beauty transgene overexpression

Bidirectional Sleeping Beauty plasmids encoding the codon-optimized cDNA of the gene of interest (*FLT3*, *KIT*, *CD123*) or their mutated variants were cloned downstream of an *EEF1A1* promoter. The same construct included a reporter cassette comprising a fluorescent protein (mCherry or mTagBFP2 for *FLT3* and *KIT*, respectively) and puromycin-acetyl-transferase (PAC) separated by a P2A peptide downstream a RPBSA chimeric promoter. CD123 constructs included human *CSF2RB* (cytokine receptor common subunit β) cDNA in place of the reporter cassette. dsDNA inserts used for cloning were synthesized by Genewiz or IDT (IDT gBlocks). Human (K562, HEK293T) or mouse (NIH-3T3 or BaF3) cell lines were electroporated using the Lonza 4D-Nucleofector system according to the manufacturer instructions with 500 ng pSB100x transposase plasmid (Addgene, 34879) and 100 ng of the transfer plasmid, unless stated otherwise. Cells underwent puromycin selection (1–5 μg ml$^{-1}$) starting 3 days after electroporation and were analysed by flow cytometry 7–14 days after the start of the selection.

## FLT3 combinatorial library

We designed a combinatorial plasmid library in which the human or the mouse codons were randomly selected at each of 16 positions (Asn354, Ser356, Asp358, Gln363, Glu366, Gln378, Thr384, Arg387, Lys388, Lys395, Asp398, Asn399, Asn408, His411, Gln412, His419) within FLT3 ECD4 (GenScript). The library was cloned to allow a total theoretical complexity of 65,536 (covered at least 10 times). Single-colony sequencing showed a rate of $n = 4$ mutations in 68.75% of the colonies, and $n = 5$ mutations in 18.75% of colonies. NGS sequencing of K562 cells transduced with the library, among 9,166 filtered reads, detected 4,375 unique amino acid sequences with a frequency of >0.0001. K562 cells were electroporated with 5–15 ng of library plasmid as described above and selected with 1 μg ml$^{-1}$ puromycin. Cells were then stained with anti-FLT3 control antibodies (BV10A4, BioLegend, 313314) and FLT3 therapeutic antibodies (4G8, BD, 563908). The single-positive (BV10A4$^+$4G8$^-$) and double-positive (BV10A4$^+$4G8$^+$) populations were sorted using the BD FACS Aria sorter and expanded in culture. After gDNA extraction, the library region was PCR-amplified with primers bearing Illumina partial adapters for NGS sequencing (250 bp paired-end, Genewiz, Azenta Life Sciences). Amplification of the integrated transgene was ensured using exon-spanning primers specific for the codon-optimized sequence. Read alignment and trimming was performed by Genewiz (Azenta), after which the sequences were translated to protein sequence and indels were discarded. After aggregation of reads encoding the same protein variant, the sequences were filtered by minimum abundance >0.001 within single-positive and double-positive samples, and the relative representations of each amino acid at each position within the sorted fractions were plotted as sequence logos. Analysis was performed using R Studio 2022.07.0 (R v.4.1.2).

## KIT degenerated codon library

We cloned a bidirectional Sleeping Beauty plasmid encoding a codon-optimized human KIT cDNA bearing unique restriction sites flanking each ECD and a reporter cassette composed of mTag-BFP2-P2A-puromycin *N*-acetyl-transferase. To introduce random amino-acids, we generated a library insert by PCR amplification of 350-bp-long pooled ssDNA oligos (IDT oPools), each encoding for the KIT ECD4 bearing degenerated bases (NNN) at each amino-acid position, flanked by homology arms for the backbone plasmid. The gel-purified insert was then cloned into the Sleeping Beauty transfer by HiFi cloning (NEB, E2621L) and plated onto ten 15 cm agar dishes to estimate the library complexity. Electroporation of HEK293T cells with low plasmid doses (50–250 ng) enabled us to select cells transduced with (~5–10%) efficiency to obtain approximately 1 variant integration

per cell. Sorting of cells positive for KIT control antibodies (104D2 or Ab55 clones) and negative for the therapeutic Fab-79D clone, and cells positive for both antibodies was performed using the BD FACS Melody sorter. gDNA of the sorted fractions was extracted and the library region was amplified by PCR using primers bearing Illumina partial adapters for NGS sequencing (Genewiz, Azenta Life Sciences). Read alignment and trimming was performed by automated analysis (Genewiz), after which the read sequences were translated into protein and indels were discarded. After aggregation of reads encoding the same protein variant, the relative abundance of each protein sequence within each sample and the enrichment in single-positive versus double-positive samples was calculated as the log[fold change]. Sequences were filtered by log[fold change] > 0.5, and the minimum-abundance fraction within single-positive samples >0.0015 of each amino acid substitution was plotted as a sequence logo. Analysis was performed using R Studio 2022.07.0 (R v.4.1.2).

## Gene overexpression through promoter integration

Reporter K562 cells overexpressing *FLT3*, *KIT* and *CD123* from the endogenous genomic locus were generated using CRISPR–Cas nuclease and HDR integration of a full *EEF1A1* promoter upstream of the gene reading frame. A dsDNA donor template encoding the *EEF1A1* promoter flanked by 50 bp homology arms (HA) was prepared by PCR amplification (Promega GoTaq G2, M7845) of plasmid DNA templates using primers with 5′-tails encoding the sHA. K562 cells were electroporated with 100 pmol *Sp*Cas9 gRNAs (for *FLT3* and *KIT*) or *As*Cas12a sgRNAs (*CD123*) complexed with 12 μg of corresponding Cas protein (IDT; SpCas9 V2, 1081058; or AsCas12a, 10001272) to form RNPs, along with 0.5 to 10 μg donor template using the Lonza 4D-Nucleofector system. Edited cells were evaluated by flow cytometry 3–5 days after electroporation and FACS-sorted (BD FACS Melody) to select cells expressing the gene of interest. Single-cell cloning was performed by limiting dilution for FLT3 and CD123 to isolate clones with the highest expression for subsequent editing experiments.

## Fluorescent ligand-binding assay

Human FLT3L (Peprotech, 300-19), hSCF (Peprotech, 300-07) or hIL-3 (200-03) were resuspended at 1 mg ml$^{-1}$ and conjugated with AF488 or AF647 (Invitrogen AF-488 or AF-647 Antibody Labeling Kit, A20181 and A20186) for 1 h at room temperature. The reaction was quenched with 1% 1 M Tris-HCl pH 8. Cell lines transduced with the respective cytokine receptor by Sleeping Beauty transposase were incubated in the presence of AF488-conjugated FLT3L, SCF or IL-3 for 15 min at room temperature, washed with PBS + 2% FBS and analysed by flow cytometry.

## Modelling of antibody–ligand affinity curve

The data obtained from the antibody and ligand affinity assays at different concentrations were analysed using the drc R package (v.3.0). The LL.4 model was used to fit the affinity curves and estimate the parameters of interest, including the maximum response (ED$_{50}$), the slope of the curve at the inflection point (Hill slope), the lower asymptote (lower limit) and the upper asymptote (upper limit). The difference among affinity curves was tested using a likelihood ratio test in which the parameters were fixed (model0) or antibody-dependent (model1).

## Western blotting

K562 (for FLT3) or NIH-3T3 (for KIT) cells overexpressing the receptor of interest by Sleeping Beauty transposase-mediated integration were starved overnight in IMDM without FBS. The next day, cells were collected, washed, counted and stimulated with the respective ligand at different concentrations (0, 0.1, 1, 10, 100 ng ml$^{-1}$) for 15 min at 37 °C. Cells were then collected, washed twice in ice-cold PBS and lysed in cell extraction buffer (Thermo Fisher Scientific, FNN0011) + protease inhibitor (1 mM PMSF) for 30 min on ice. Protein-containing supernatants were then collected, and the concentration was measured using

the BCA assay. A total of 35 μg of protein lysate was then mixed with Laemmli loading buffer (Bio-Rad, 161-0747) supplemented with 0.1× volumes of 2-mercaptoethanol and incubated for 5 min at 95 °C. Denatured protein samples and a prestained protein ladder (Bio-Rad Precision Plus Kaleidoscope, 1610375) were loaded onto 4–12% Tris-Glycine gel (Invitrogen Novex WedgeWell, XP04125BOX), run at 150 V for 90 min and transferred onto PDVF membranes by blotting at 100 mA for 60 min. Washed membranes were then blocked with TBST + 5% BSA, washed three times with TBST and cut at the 75 kDa mark to separate FLT3 and KIT (110–140 kDa) and actin (42 kDa) bands and incubated with primary antibodies at 1:1,000 concentration (pKIT Tyr568/570, Cell Signalling, 48347S; pFLT3 Y589-591, Cell Signalling, 3464S) for 1 h at room temperature. The membrane was then washed three times and incubated with the appropriate secondary antibodies (anti-rabbit IgG, HRP-linked Antibody, Thermo Fisher Scientific, 31460), washed and developed with Thermo Scientific SuperSignal West Femto substrate (34095). The membranes were imaged using the ImageQuant LAS 4000 system. To reprobe the same samples with anti-FLT3 or anti-KIT antibodies, the membranes were incubated for 30 min with Thermo Scientific Restore Stripping Buffer (Thermo Fisher Scientific, 21059), blocked, washed and incubated with primary antibodies (anti-FLT3, Origene, TA808157; anti-KIT, Invitrogenc MA5-15894). The membrane was then washed and incubated with secondary antibodies (anti-mouse IgG, HRP-linked antibodies, Thermo Fisher Scientific, 62-6520; or anti-rat IgG, HRP-linked antibodies, Thermo Fisher Scientific, 31470) and reimaged as described above.

## Base editing of cell lines
Human K562 cells were cultured in IMDM supplemented with 10% FBS, 1% penicillin–streptomycin (10,000 U ml⁻¹), 2% L-glutamine (200 mM). For base-editing experiments, K562 cells were collected and resuspended in electroporation solution[51] supplemented with 500 ng base editor plasmid and 150–360 pmol of sgRNA (IDT) in 20 μl. Cells were electroporated using the Lonza 4D-Nucleofector system and cultured for 72 h before evaluation by flow cytometry or gDNA collection.

## Base-editing evaluation by Sanger sequencing
Genomic DNA was extracted using the DNeasy Blood & Tissue Kit (Qiagen, 69506) or QuickExtract solution (Lucigen, QE0905T). PCR amplification of the 450–600 bp region surrounding the target residue was performed using GoTaq G2 polymerase (Promega, M7848) and purified using the SV-Wizard PCR Clean-Up kit (Promega, A9282) or sparQ PureMag magnetic beads (MagBio, 95196). Sanger sequencing was performed by Genewiz (Azenta Life Sciences) and base-editing outcomes were estimated using the editR v.1.08 R package[52].

## CAR T cell generation
Peripheral blood mononuclear cells were isolated by ficoll density-gradient centrifugation from healthy blood donor samples (non-HLA-matched with CD34⁺ HSPCs). Total PBMCs or purified T cells (Pan T Cell Isolation Kit, Miltenyi, 130-096-535) were cultured in IMDM supplemented with 10% FBS, 1% penicillin–streptomycin (10,000 U ml⁻¹), 2% L-glutamine (200 mM), 5 ng ml⁻¹ IL-7 (Peprotech, 200-07) and 5 ng ml⁻¹ IL-15 (Peprotech, 200-15) in the presence of CD3-CD28 Dynabeads (Gibco, 11141D) at a 3:1 bead:cell ratio. On day 2–3 of culture, cells were counted and transduced with a third-generation bidirectional lentiviral vector encoding the CAR and an optimized version[36] of the truncated EGFR[53] marker at a multiplicity of infection (MOI) of 5 to 10. Dynabeads were removed at day 7 and the culture was continued for a total of 14 days before either vital freezing or in vitro/in vivo experiments. The CAR T cell phenotype and transduction efficiency was evaluated by flow cytometry.

## CAR T cell co-culture assays
Target cells (K562 cells, AML PDXs or CD34⁺ HSPCs) were plated in flat-bottom 96-well plates, 10,000 cells per well in IMDM supplemented with 10% FBS. Freshly cultured or thawed CAR T or unstransduced T cells were marked with CellTrace Yellow or Violet reagent (Invitrogen, C34567 and C34557) for 15 min at 37 °C and then washed with complete medium. T cells then were added to the target cells at several effector:target ratios (10, 5, 2.5, 1.25, 0.625, 0) in a total volume of 200 μl in triplicate or quadruplicate. The plates were incubated at 37 °C in a humidified incubator with 5% CO₂ for 6 h (early timepoint) or 48 h (late timepoint). For analysis, cells were transferred to V-bottom plates supplemented with 10–15 μl CountBeads (BioLegend, 424902), centrifuged and resuspended in staining mix containing human Fc-blocking reagent (Miltenyi, 130-059-901), and CD3 (or CD4/CD8 cocktail), CD69 and CD107a antibodies unless stated otherwise. In K562 cell experiments, FLT3 BV10A4 and/or CD123 9F5 or KIT 104D2 antibodies were included in the staining mix to evaluate the residual target expression at the end of co-culture. In CD34⁺ HSPC experiments, cells were stained for CD34, CD90 and CD45RA. After incubation at room temperature for 15 min, the cells were washed and resuspended in annexin V binding buffer (BioLegend, 422201) containing annexin V FITC or PacificBlue (BioLegend, 640945 or 640918) and 7-AAD. The plates were analysed using the BD Fortessa high-throughput system (HTS) and acquired using BD FACS Diva (v.6). Absolute counts of live (Annexin V⁻7-AAD⁻) target cells were calculated using CountBeads and normalized to the E:T = 0 (no T cells) condition (absolute count of target cells divided by the median of the target cells in E:T = 0 replicates). The MFI of target genes (*FLT3, CD123, KIT*) was normalized in the same manner to account for slight differences in transgene expression.

## Base editor mRNA in vitro transcription
Base editor mRNA was prepared by T7 run-off in vitro transcription using a custom plasmid template encoding for a T7 promoter, a minimal 5′-UTR, the base editor reading frame, 2× HBB 3′-UTR and a 110–120 bp poly(A) sequence[54]. The plasmid template was linearized by BbsI-HF restriction digestion (NEB, R3539) and purified by phenol–chloroform extraction (Sigma-Aldrich, P2069). mRNA IVT was performed using the NEB HiScribe T7 kit (E2040S) and co-transcriptionally capped with ARCA (3′-O-Me-m7G(5′)ppp(5′)G RNA cap analogue, NEB, S1411L) or CleanCap AG (Trilink, N-7113). Partial (75–85%) or total UTP substitution with N1-methyl-pseudo-UTP (Trilink, N-1103) or 5-methoxy-UTP (Trilink, N-1093) was performed as indicated. Dephosphorylation with QuickCIP (NEB, M0525L) or DNase treatment (NEB, M0303L) was added after IVT reaction (30 min at 37 °C). IVT mRNA was purified using the NEB Monarch RNA Clean up kit (T2050L) or sparQ PureMag magnetic beads (MagBio, 95196) and resuspended in RNase-free water. mRNA was quantified using the Nanodrop-8000 spectrophotometer and quality control was performed using the Agilent Fragment Analyzer with RNA Kit-15NT (Agilent, DNF-471).

## CD34⁺ base editing
Mobilized peripheral-blood-derived human CD34⁺ HSPCs were cultured and electroporated as previously described[55]. In brief, cryopreserved cells were thawed and cultured in StemSpan SFEMII medium (StemCell, 09655) supplemented with 1% penicillin–streptomycin (10,000 U ml⁻¹), 1% L-glutamine (200 mM), 125 ng ml⁻¹ hSCF (Peprotech, 300-07), 125 ng ml⁻¹ hFLT3L (Peprotech, 300-19), 62.5 ng ml⁻¹ hTPO (Peprotech, 300-18), 0.75 μM StemRegenin-1 (SR1, StemCell, 72344) and 35 nM UM171 (Sellekchem, S7608) unless stated otherwise. Then, 48 h after thawing (unless stated otherwise), cells were collected and electroporated using the Lonza 4D-Nucleofector system by resuspending the cells in P3 solution (Lonza, V4XP-3024) supplemented with 100–250 nM base editor mRNA, 15–20 μM sgRNA (IDT) and 1.2 U μl⁻¹ RNase inhibitor (Promega RNAsin Plus, N2611) or 1.5–3% (v/v) glycerol. After electroporation, cells were then counted and transplanted after 24 h for in vivo experiments, or cultured for an additional 5–7 days in the same medium described above at 0.5 M ml⁻¹ for in vitro experiments.

## Transduction and expansion of human AML xenografts

Human patient-derived AML xenografts (PDX) were thawed and cultured in StemSpan SFEMII medium (StemCell, 09655) supplemented with 1% penicillin–streptomycin (10,000 U ml⁻¹), 1% L-glutamine (200 mM), 50 ng ml⁻¹ hSCF (Peprotech, 300-07), 50 ng ml⁻¹ hFLT3L (Peprotech, 300-19), 25 ng ml⁻¹ hTPO (Peprotech, 300-18), 10 ng ml⁻¹ IL-3, 10 ng ml⁻¹ G-CSF, 0.75 µM StemRegenin-1 (SR1, StemCell, 72344), 35 nM UM171 (Sellekchem, S7608) and 10 µM PGE2. Cells were transduced with a third-generation LV vector expressing mNeonGreen under a hPGK promoter (titre, ~2×10¹⁰ TU ml⁻¹) at an MOI of 100 and cultured overnight at 37 °C in a humidified incubator under 5% CO₂. Cells were collected the next day and transplanted into 4–8-week-old NBSGW female mice by tail-vein injection. Engraftment was monitored by peripheral blood collection and FACS analysis. Mice were euthanized when PDX AML cells exceeded around 20% of total white blood cells. BM and spleen were collected and either vitally frozen or FACS-sorted (BD FACS Melody sorter) to isolate mNeonGreen⁺ cells and retransplanted into new NBSGW recipients to obtain fully transduced PDXs.

## In vivo xenotransplantation experiments

All of the animal experiments were performed in accordance with regulations set by the American Association for Laboratory Animal Science and Institutional Animal Care and Use Committee (IACUC) approved protocol (DFCI, 21-002). Mice were housed in sterile individually ventilated cages and fed autoclaved food and water, under a standard 12 h–12 h day–night light cycle. Female NOD.Cg-Kit^W-41J^Tyr⁺Prkdc^scid^Il2r g^tm1Wjl^/ThomJ mice (aged 4–8 weeks; termed NBSGW, Jackson, 026622) were xenotransplanted with 1 million human CD34⁺ HSPCs per mouse by tail-vein injection. Human engraftment was monitored by peripheral blood FACS analysis at 7–9 weeks. In AML co-engraftment experiments, mice then received 0.75 M human patient-derived AML xenograft cells (PDX), previously transduced with a mNeonGreen, by tail-vein injection. Then, 10 days after AML transplantation, mice received 2.5 M CAR T cells by tail-vein injection unless stated otherwise. The mice were monitored by peripheral blood analysis for a period of 2 weeks after CAR T cell administration and then euthanized. At euthanasia, the bone marrow, spleen and peripheral blood were collected for FACS and genomic analysis.

## Colony-forming assays

CFU assays were performed by plating 1,000 CD34⁺ cells per well for in vitro CD34⁺ HSPCs experiments, or 25,000 total bone marrow cells per well for xenotransplanted BM-derived assays, unless stated otherwise. Cells were resuspended in Methocult H4034 medium (StemCell, 04034) and plated into SmartDish meniscus-free six-well plates. Wells were imaged and analysed after 2 weeks using the StemCell STEMvision system and STEMvision software using the 14 day bone marrow setting. For flow cytometry analysis, methylcellulose medium was softened with prewarmed PBS, collected and washed twice before analysis. The percentage of leukaemia cells within the BM-derived CFU was determined by mNeonGreen fluorescence.

## Lineage differentiation cultures from CD34⁺ HSPCs

CD34⁺ HSPCs were thawed and edited as previously described. At day 7 after electroporation, cells were placed into differentiation media as follows: myeloid culture (SFEMII, penicillin–streptomycin 1%, Q 1%, FLT3L 100 ng ml⁻¹, SCF 100 ng ml⁻¹, TPO 50 ng ml⁻¹, GM-CSF 100 ng ml⁻¹, IL-6 50 ng ml⁻¹ and IL-3 10 ng ml⁻¹), macrophage (RPMI, FBS 10%, M-CSF 100 ng ml⁻¹ and GM-CSF 50 ng ml⁻¹) on non-tissue-treated plates, DC (RPMI, FBS 10%, GM-CSF 50 ng ml⁻¹ and IL-4 20 ng ml⁻¹), megakaryocyte (SFEMII, penicillin–streptomycin 1%, Q 1%, SCF 25 ng ml⁻¹, TPO 100 ng ml⁻¹, IL-6 10 ng ml⁻¹, IL-11 5 ng ml⁻¹), granulocyte (SFEMII,

penicillin–streptomycin 1%, Q 1%, GM-CSF 100 ng ml⁻¹, then transitioned to RPMI + FBS 20% + G-CSF 50 ng ml⁻¹ after 7 days). The immunophenotype of differentiated cells was evaluated by flow cytometry.

## Flow cytometry

Cell lines were collected, washed in PBS + 2% FBS, resuspended in 100 µl, incubated with human or mouse Fc-blocking reagent (Miltenyi 130-059-901 and 130-092-575) and then stained with the indicated antibodies for 20–30 min at 4 °C and washed. Viability was assessed by LiveDead yellow (Invitrogen, L34959) 7-AAD (BioLegend, 420404) or propidium iodide (PI, BioLegend, 421301) staining. Immunophenotyping of in vitro base-edited CD34⁺ HSPCs was evaluated by staining for 30 min at 4 °C with CD34 BV421, CD45RA APC-Cy7, CD90 APC, CD133 PE and, in some experiments, CD49f PE-Cy7 antibodies after incubation with Fc-blocking reagent (Miltenyi, 130-059-901). Viability was assessed by 7-AAD or PI staining.

For assessment of reactive oxygen species production, cells from the myeloid differentiation culture were collected and incubated with PMA (5 ng µl⁻¹) for 15 min at 37 °C, then CellROX Green reagent (Invitrogen, C10444) was added at a final concentration of 500 nM and the sample was kept for an additional 30 min at 37 °C (ref. 17). Cells were then stained with hFC block and HLA-DR PacificBlue, CD11b APC-Fire750, CD14 BV510, CD33 PE-Cy7 and CD15 AF700 antibodies, and washed and analysed. PI was included for viability. To test phagocytosis, in vitro differentiated macrophages were incubated with pHrodo green *E. coli* bioparticles (Thermo Fisher Scientific, P343666) for 60 min at 37 °C, washed and analysed by flow cytometry[17]. To assess signal transduction of in vitro differentiated myeloid culture, cells were collected, cytokine-starved overnight in SFEMII (not supplemented with cytokines) and then stimulated with GM-CSF 50 ng ml⁻¹, IL-4 50 ng ml⁻¹, IL-6 100 ng ml⁻¹, type I IFNα A/D 5,000 U ml⁻¹, PMA 100 ng ml⁻¹ plus ionomycin 1 µg ml⁻¹ or LPS 100 ng ml⁻¹ for 30 min at 37 °C. Cells were then fixed with BioLegend fixation buffer (420801) and permeabilized with True-Phos Perm Buffer (425401) according to the manufacturer's instructions. Cells were then aliquoted into two fractions, stained with the following antibody panels and analysed using flow cytometry: (1) pSTAT1 PE-Cy7, pSTAT3 BV421, pSTAT5 PE and pSTAT6 AF647; (2) pERK1/2 BV421, pCREB AF647, p38/MAPK PE and pRPS6 PE-Cy7. In vitro differentiated dendritic cells were stained with hFC block, CD1c APC, CD33 PE-Cy7, CD303 APC-Cy7, CD141 KiraviaBlue520 and FLT3 PE antibodies and 7-AAD for viability. To assess upregulation of co-stimulatory molecules, dendritic cells were incubated with 100 ng ml⁻¹ LPS for 30 min at 37 °C and stained with hFC block, CD1c APC, CD303 APC-Cy7, CD11c BUV661, FLT3 PC7, HLA-DR Pacific Blue, CD80 BV605 and CD86 FITC antibodies, plus PI for viability. To evaluate macrophage differentiation towards an M1-like (CD80⁺CD86⁺HLA-DR⁺) or M2-like (CD86⁻CD200R⁺CD206⁺HLA-DR⁻) phenotype, in vitro differentiated macrophages were plated in a 96-well plate and stimulated overnight with LPS 100 ng ml⁻¹ or IL-4 20 ng ml⁻¹, respectively. Macrophages were then detached and stained with hFC block, CD11b APC-F750, CD33 BB515, CD200R PE-Cy7, CD206 PE, CD209 AF647, HLA-DR PacificBlue, CD80 BV605 and CD86 AF700 antibodies, and PI for viability and analysed by flow[56]. Neutrophil extracellular trap (NETosis) induction in differentiated granulocytes was performed as previously described[56]. In brief, cells were seeded in a 96-well plate, stimulated with PMA 50 ng ml⁻¹ for 4 h at 37 °C and then stained with hFC block and CD15 FITC, CD66b PE-Cy7 and CD33 APC antibodies for 15 min at room temperature. Granulocytes were then washed and fixed with PFA, supplemented with DAPI and PI staining solution to stain exocytosed nucleic acid, and analysed by flow cytometry. In vitro differentiated megakaryocytes were analysed by staining for 30 min at 4 °C with hFC block and CD41 PE, CD42b AF647, CD61 FITC, CD34 BV421 and CD45 BV786 antibodies. For megakaryocyte ploidy, cells were stained with stained with hFC block and CD42b AF647 and CD61 FITC antibodies, and then washed and resuspended in 500 µl FxCycle

PI/RNase staining solution (Molecular Probes, F10797) and analysed by flow cytometry[57].

Peripheral blood collected from in vivo experiments was lysed twice for 10 min at room temperature using ACK reagent (StemCell, 07850), washed, incubated with human and mouse Fc-blocking reagent (Miltenyi, 130-059-901 and 130-092-575), and stained at room temperature for 15 min with hCD45 BV421, mCD45 PE, CD3 AF647, CD19 BV605 and CD33 PE-Cy7 antibodies, unless stated otherwise. 7-AAD was included as viability stain. To comprehensively identify haematopoietic populations in BM samples[58], collected cells were lysed using ACK reagent, washed twice and supplemented with Count-Beads for absolute quantification. The samples were then incubated with human and mouse Fc-blocking reagent and stained for 30 min on ice with hCD45 BV786, mCD45 PerCP-Cy5.5, CD3 PE-Cy5, CD7 AF700, CD10 BUV737, CD11c BUV661, CD14 BV510, CD19 BV605, CD33 PE-Cy7, CD38 BUV396, CD45RA APC-Cy7, CD56 BUV496, CD90 APC, FLT3 PE or BV711, and either KIT BV711 or CD123 PE antibodies (depending on the experimental design) (Supplementary Table 2), with the addition of 50 µl per sample Brilliant Stain buffer (BD, 659611). A T-cell-focused panel was applied to the spleen and BM samples by staining for 30 min on ice with hCD45 BV786, mCD45 PerCP-Cy5.5, EGFR AF488, CD3 BUV396, CD4 APC, CD8 PacificBlue, CD62L BV605, CD69 PE-Cy7, CD45RA APC-Cy7, PD1 BUV737 and LAG3 PE antibodies. Propidium iodide (3.5 µl per 100 µl) was included as a viability stain shortly before analysis. Cells were acquired on the 5-laser BD Fortessa flow cytometer using BD FACS Diva (v.6) and analysed using FCS Express 6 (DeNovo Software).

## Gene expression analysis of edited CD34+ HSPCs

CD34+ cells from healthy donors were either base edited for *FLT3*, *CD123*, *KIT* or *AAVS1* or untreated (electroporation only). Then, 4 days after editing, to enable complete turnover of the edited receptors, cells were starved overnight in SFEMII without cytokines. On day 5 after editing, CD34+ HSPCs were stimulated with either FLT3L 500 ng ml$^{-1}$, SCF 500 ng ml$^{-1}$ or IL-3 100 ng ml$^{-1}$ according to the editing condition. *AAVS1* and the electroporation-only conditions were divided into three separate wells and incubated with the three cytokines, to provide appropriate controls for each donor and stimulation. After 24 h of stimulation, cells were collected and RNA was extracted using the Qiagen RNAeasy Mini kit (74104) with on-column DNase I treatment (Qiagen, 79254). RNA was quantified using the Nanodrop and sent for sequencing (DNBSEQ stranded mRNA library, BGI). Variations in the gene expression profile in response to specific ligand stimulation were compared among *FLT3*-, *KIT*- and *CD123*-edited samples, control *AAVS1*-edited samples and electroporation-only samples. Raw sequencing files were filtered for quality control and aligned to the reference human genome hg38 using STAR workflow[59], obtaining, as a result, the gene-based count matrices. On the basis of a preliminary principal component analysis, we identified one sample (HD-2, *KIT*-edited, unstimulated) as an outlier and removed it from the subsequent analysis. We also filtered poorly detected genes with cumulative read counts smaller than 10 or measured (read count > 0) in less than two samples. Statistical analysis was performed in R software, using the DEseq2 package[60]. Gene counts were modelled using a negative binomial model, the significance of specific coefficients was tested using the Wald test, and *P* values were adjusted for multiplicity using the Holm method[61]. We analysed the expression pattern for each gene by using the full model configuration: FullModel: GeneExpr_i = b_0 + b_Don. Donor + b_Edit. Editing+ b_Stim Stimulation + b_(Edit.Stim) Editing x Stimulation. We then tested the significance of the following parameters: b_Edit(UT VS AAVS1): genes differentially expressed in UT with respect to the reference group *AAVS1*; b_Edit(FLT3 | KIT | CD123 VS AAVS1): genes differentially expressed in sample edited using *FLT3*, *KIT* or *CD123* sgRNA with respect to the reference group *AAVS1*; b_Stim: genes differentially expressed in stimulated samples (FLT3L, SCF or IL-3) with respect to the unstimulated group; b (Edit.Stim): genes differentially expressed in an editing group-specific manner after stimulation.

The expression values for the significant genes identified in each comparison are displayed as heat maps in Extended Data Fig. 5c,e. The figures were generated using the R package ggplot2[62].

## Phosphoproteome analysis by mass spectrometry

CD34+ cells were base edited for *FLT3*, *CD123*, *KIT* or *AAVS1*. Four days after editing, cells were starved overnight in SFEMII without cytokines. On day 5 after editing, CD34+ HSPCs were incubated at 37 °C with FLT3L (500 ng ml$^{-1}$), SCF (500 ng ml$^{-1}$) or IL-3 (100 ng ml$^{-1}$) according to the editing condition (*AAVS1* was stimulated independently with each of the three cytokines). After 6 h, cells were collected and dry pellets were flash-frozen in liquid nitrogen. Cell pellets were lysed in 5% SDS, 50 mM TEAB pH 8.5 with protease inhibitors (Roche) and phosphatase inhibitors (Sigma-Aldrich, phosphatase inhibitor cocktails 2 and 3) for 30 min with shaking. The protein concentration was determined by BCA (Thermo Fisher Scientific). Aliquots of protein (30 µg) were treated with 250 U of benzonase and incubated for 5 min at room temperature. Proteins were then reduced with DTT (final concentration, 10 mM; 56 °C for 30 min), alkylated with iodoacetamide (25 mM) for 30 min at room temperature, acidified 1:10 with 12% phosphoric acid, further diluted 6× with binding buffer (90% methanol and 100 mM TEAB) and then applied to S-trap micro columns (Protifi). The columns were washed four times with 150 µl binding buffer, and proteins digested with trypsin overnight at 37 °C. Peptides were extracted from the column by centrifugation after applying elution buffers (1–50 mM TEAB, 2–0.2% formic acid and 3–50% acetonitrile), and dried by vacuum centrifugation. Peptides were resuspended in 100 mM HEPES, pH 8.0, and treated with TMT-Pro isobaric labelling reagents for 1 h at room temperature using 8 out of the 16 possible reagents (126, 127N, 128N, 129N, 130N, 131N, 132N and 133N, or 127C, 128C, 129C, 130C, 131C, 132C, 133C and 134N). Each experiment included proteins from target and control edits ± ligand stimulation for 2 separate donors. After labelling, the reactions were quenched with 5% hydroxylamine in 50 mM TEAB, combined, dried, reconstituted in 0.1% TFA and desalted by C18. Phosphopeptides were then enriched by Fe$^{3+}$-NTA-IMAC as described previously[63] and analysed by nanoflow liquid chromatography coupled with tandem mass spectrometry (LC–MS/MS)[64]. Peptides were loaded onto an analytical column with an integrated ESI emitter tip (75 µm internal diameter, fused, packed with 25 cm of 1.9 µm Reprosil C18, ESI Source Solutions), desalted and resolved with an LC gradient (1–5% B in 5 min, 5–18% B in 115 min, 18–27% B in 50 min, 27–39% B in 30 min, 39–80% B in 15 min, where A is 0.1% formic acid in water and B is 0.1% formic acid in acetonitrile). Peptides were analysed on the TimsTOF Pro II mass spectrometer with 6 PASEF ramps (mobility range 0.8–1.4 V s cm$^{-2}$, ramp time 100 ms), MS/MS stepping enabled, a 1.5 Da isolation width and a target intensity of 17,500. Active exclusion was enabled with a release time of 30 s. Data files were searched against a forward reverse human protein database (UCSG) using MSFragger with precursor and product ion tolerances of 25 ppm and 0.05 Da, respectively. Search parameters specified variable oxidation of methionine, variable phosphorylation (STY), fixed carbamidomethylation of cysteine and fixed TMT-Pro modification of peptide N termini and lysine residues. Custom Python scripts[65] were used to filter search results to 1% FDR, as well as extract TMT reporter ion intensities and correct for isotopic impurities. After filtering out PSMs with missing TMT values, reporter ion evidence was summed for peptides of the same modification and charge state, with values of less than 20 counts imputed to the estimated noise level (20 counts). Data were then quantile normalized after removing peptides with a median reporter ion intensity of less than 100 counts. We analysed the phosphorylation pattern for each peptide using the full model configuration: FullModel: PhosphoPeptide_i = b_0 + b_Don. Donor + b_Edit. Editing+ b_Stim Stimulation; and tested the significance of b_Edit and b_Stim using a likelihood ratio test and the following

reduced models: ReducedModel for testing b_Edit: PhosphoPeptide_i = b_0 + b_Don. Donor + b_Stim Stimulation; ReducedModel for testing b_Stim: PhosphoPeptide_i = b_0 + b_Don. Donor + b_Edit. Editing.

Given the high signal-to-noise ratio of this methodology and the limited sample size, the results were analysed descriptively. We calculated the average (across donor) fold change between the phosphorylation levels in the unstimulated and stimulation conditions for both editing settings.

## Off-target evaluation

GUIDE-seq was conducted in HEK293T cells[66]. The GUIDE-seq dsODN was prepared by annealing PAGE-purified oligos GUIDE-seq dsODN sense and antisense (a list of the sequences is provided in Supplementary Table 1) using the following conditions: 95 °C for 2 min, slow ramp down to 25 °C with an increment of −0.1 °C per cycle for 700 cycles. HEK293T cells were electroporated with 4 µg SpRY Cas9 mRNA, 200 pmol of dsODN and 600 pmol of guide RNA using the SF Cell Line 4D Nucleofector X Kit S (Lonza, V4XC-2024). The dsODN concentration was selected after a titration to identify the maximal dose permitting at least 60% cell viability 24 h after delivery by electroporation. Genomic DNA was collected 4–6 days after electroporation. We performed GUIDE-seq library construction with some modifications as previously described[67]. Adapter oligos (from IDT) were annealed using the following conditions: 95 °C for 2 min, slow ramp down to 25 °C with an increment of −0.1 °C per cycle for 700 cycles. We added Tn5-transposase (86% reaction volume) to pre-annealed oligo (14% reaction volume) for transposome assembly at room temperature for 1 h. We prepared tagmentation reactions by titrating the assembled transposome into 200 ng of genomic DNA (gDNA) in the presence of 1× TAPS-DMF, incubating at 55 °C for 7 min. 1× TAPS-DMF buffer contains 50 mM TAPS-NaOH at pH 8.5 (at room temperature; Thermo Fisher Scientific, J63268AE), 25 mM magnesium chloride ($MgCl_2$; Thermo Fisher Scientific, AM9530G), 50% dimethylformamide (Thermo Fisher Scientific, 20673). The transposome was inactivated with 0.2% SDS at room temperature for 5 min. Tagmentation conditions were selected that yielded DNA fragment size distribution concentrated at 500–1,000 bp. After the tagmented DNA was isolated, we performed a two-step PCR amplification with the first step (PCR 1) for GUIDE-seq oligo-specific amplification and the second step (PCR 2) for indexing. The PCR 1 reaction mix contained 50 mM magnesium chloride ($MgCl_2$; Thermo Fisher Scientific, AM9530G), 2× Platinum SuperFi PCR Master mix (Thermo Fisher Scientific, 12358050), 0.5 M tetramethylammonium chloride solution (TMAC; Sigma-Aldrich, T3411), 10 µM GUIDE-Seq oligo-specific primer (GSP, IDT) (Supplementary Table 1), 10 µM i5 amplification primer (IDT, 5′-AATGATACGGCGACCACCGAGATC-3′) and tagmented DNA. The PCR 2 reaction mix was the same as for PCR 1, but without 50 mM $MgCl_2$, and we added 10 µM i7 barcode primer (IDT) instead of GSP. The PCR 1 product was size-selected with 45 µl (0.9× original reaction) of AMPure XP Beads (Agencourt, A63882) and eluted in nuclease-free water before proceeding to the next step. The PCR 2 amplified product was double-size selected with 25 µl (0.5× original reaction) of AMPure XP Beads and then with 17.5 µl (0.35× original reaction) of the beads to isolate PCR products in the range of 200–500 bp, quantified using the Qubit dsDNA High-Sensitivity Assay (Thermo Fisher Scientific, Q32851), checked for size and purity using the TapeStation (Agilent 4200 System) and KAPA qPCR (KAPA Library Quantification Kit Illumina Platform), before samples were equimolar pooled and sequenced. The libraries were deep-sequenced as a pool using a paired-end 150 bp run on the Illumina MiniSeq system (~1.5 million reads per sample) with the following parameters: 25–35% PhiX and 151|8|17|141 (Read1|Index1|Index2|Read2). Deep sequencing data from the GUIDE-seq experiment were analysed using GS-Preprocess (https://github.com/umasstr/GS-Preprocess) and Bioconductor Package GUIDEseq (v.1.4.1)[68]. The GS-Preprocessing script demultiplexed the raw Illumina FASTQ files based on the index information and mapped the sequence files to the reference genome. It generated the input files (BAM files of plus and minus strands, gRNA fasta files and a unique molecular index (UMI) for each read) for Bioconductor Package GUIDEseq. The window size for peak aggregation was set to 50 bp. Off-target site identification parameters were set for SpRY-Cas9 as follows: PAM.pattern = "NNN", max.mismatch = 10, includeBulge = TRUE, max.n.bulge = 2L, upstream = 50, downstream = 50, max.mismatch = 6 and allowed.mismatch.PAM = 3. To filter false-positive signals, we called the peaks and calculated their frequencies in the control group. These background peaks were used to filter out potential hot spots or sequencing noise in Cas9–sgRNA treatment groups. This analysis provided a list of potential off-target sequences that were ranked with regard to unique reads on the basis of the incorporated UMI and the adapter ligation position, which has been correlated with the activity of SpRY-Cas9 at each sequence (Supplementary Table 16). The results were filtered by the number of mismatch + bulge < 6 to identify high-risk off-target sites.

Candidate in silico predicted off-target sites were selected on the basis of the CRISPOR online tool (top six intronic and top six exonic sites based on sequence homology for *FLT3*, *CD123* and *KIT* sgRNAs when using a SpRY-Cas9 PAM (NRN). We amplified the off-target sequences by PCR on genomic DNA extracted from edited or control CD34+ cells, using three different healthy donors for each condition. PCR products were designed to be <400 bp in size. PCR amplicons for each condition were pooled together and sent for library preparation and NGS sequencing (300 bp paired-end reads, Azenta). One exonic and two intronic sites for *CD123* sgRNA-R were eliminated from the final analysis owing to an insufficient number of reads. Reads were processed using the CRISPResso2 tool (CRISPRessoPooled), setting as the quantification window the amplicon interval overlapping with the sgRNA. We modelled the proportion of EditedReads over TotalReads (unedited + edited) using a binomial response generalized linear model (logistic regression) including the following covariates: DonorID, factor distinguishing cell donor; treatment, a binary variable having value treatment = 1 for edited and treatment = 0 for mock samples. Standard logistic regression estimation frameworks have convergence issues when the input data table is sparse (multiple samples with EditedReads = 0). To address this issue, we adopted a bias-reduction approach[69], a second-order unbiased estimator that returns finite standard errors for the coefficients under all conditions. We performed the regression and analysed the results for each locus separately and calculated: (1) EditedReads proportion estimation and related 95% confidence interval for all samples; (2) the significance of the treatment coefficient (Wald test for the null hypothesis $\beta = 0$ and $\alpha = 0.05$). Suppose the coefficient estimate is positive and significant. In that case, the expected EditedReads proportion in the edited samples (treatment = 1) is greater than in the mock samples, and therefore bona fide, off-target, base editor activity is detected. For the quantification of on-target indels proportion, a similar procedure was used by selecting from the edited reads (Unedited="FALSE") only those having either n_deleted > 0 or n_inserted > 0. The analysis was performed using R software. For the estimation of the logistic model, we used the R package brglm2 (ref. 70).

## RNA-editing analysis

The presence and extent of guide-independent RNA editing activity caused by ABE have been verified using the data generated in the RNA-seq experiment described in the 'Gene expression analysis of edited CD34+ HSPCs' section. Each sample's R1 and R2 reads files were processed using REDItools2[71] using a base quality filtering threshold (-bq 30) and a hg38 transcript annotation database. On the basis of the output generated by REDItools2, for each sample, we selected the adenine nucleotides in the reference genome with the highest coverage (top 5%) by transcriptomics reads. We then calculated the percentage of detected A>G transition events (number of A>G transition)/A coverage). The distribution of A>G transition rates across all A nucleotides

considered for each sample is graphically represented using box plots in Extended Data Fig. 8c. The sample-specific coverage threshold and the total number of A nucleotides considered is reported in Supplementary Table 21.

## Statistical analyses

The $n$ values indicate the number of biologically independent samples, animals or experiments. Data are summarized as mean ± s.d. Inferential techniques were applied in presence of adequate sample sizes ($n \geq 5$), otherwise only descriptive statistics are reported. Comparisons between two groups were performed using unpaired $t$-tests. FDR-adjusted $P$ values are reported when appropriate. When one variable was compared among more than two groups, one-way ANOVA was used. When multiple variables were compared among more than two groups, two-way ANOVA was used. The $P$ values of the row or column effect is reported as appropriate to describe the significance of the selected parameter on the measured variable. Multiple comparisons between groups are performed with two-stage step-up procedure of Benjamini, Krieger and Yekutieli to control the FDR ($Q = 0.05$) when appropriate. In all of the analyses, the significance threshold was set at 0.05; NS, not significant. Analyses were performed using GraphPad Prism v.9.4 (GraphPad). The methods for the statistical analyses of genomic off-target sites, RNA-seq, phosphoproteomic profiling by MS, RNA editing and antibody/ligand affinity curves are reported in each Methods section.

## Reporting summary

Further information on research design is available in the Nature Portfolio Reporting Summary linked to this article.

## Data availability

RNA-seq datasets have been deposited with links to BioProject accession number PRJNA986596 in the NCBI BioProject database (https://www.ncbi.nlm.nih.gov/bioproject/). Targeted amplicon sequencing datasets of off-target sites have been deposited at BioProject (PRJNA986845). All MS data files are available for download (ftp://massive.ucsd.edu/MSV000092272). All other data are available in the Article and its Supplementary Information. Source data are provided with this paper.

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

**Acknowledgements** We thank D. A. Williams, G. Schiroli and all of the members of the Genovese's laboratory for discussions; L. Sereni, G. Ceglie, A. Martinuzzi, V. Cinella, S. Rizzato, A. Bianchi and M. Freschi (Genovese laboratory) for help with experiments; A. Nguyen, N. Neri and A. Verma (Bauer laboratory) for the help in performing and analysing GUIDE-seq analysis; L. J. Zhu, T. Rodriguez and S. A. Wolfe for support during the analysis and supplying the Tn5 transposase (UMass Med); F. Perner and C. Marinaccio (Armstrong laboratory) for providing the PDXs and BaF3 cell line. G.C. conducted this study as partial fulfilment of his PhD in Translational and Molecular Medicine—DIMET, Milano-Bicocca University (Italy). This work was supported by grants to P.G. from the Children's Cancer Research Fund (Emerging Scientist Award), the Leukemia Research Foundation (New Investigator Blood Cancer Research Grant Program), the National Blood Foundation (NBF2021GRANT-PG), the American Cancer Society (Discovery Boost Grants, DBG-23-1039598-01-IBCD), the National Institutes of Health (NIH) (grant R01AI155796) and the Boston Children's Hospital (Gene Therapy Program startup package and Pilot Research Award from the Office of Sponsored Programs). J.A.M. acknowledges support from the Mark Foundation for Cancer Research, the NIH (grants CA233800, CA247671 and U24DK116204) and the Massachusetts Life Science Center. S.A.A. was supported by NIH (grants CA176745, CA259273 and CA066996). G.C. and A.C. were partially supported by fellowships from the American Italian Cancer Foundation (AICF).

**Author contributions** G.C. designed and performed research, interpreted data and wrote the manuscript. A.C., A.M., M.S.M., I.U.Z. and D.K. performed experiments and interpreted data. J.Z. and D.E.B. performed and interpreted GUIDE-seq analysis. S.B.F. and J.A.M. performed, analysed and interpreted phospho-proteomic profiling. D.P. oversaw statistical analyses and performed bioinformatic analyses of RNA-seq, phospho-MS and NGS off-target data. C.B., A.R., J.R., J.A.M., D.E.B. and S.A.A. provided discussion and some primary samples. P.G. designed and supervised research, interpreted data, wrote the manuscript and coordinated the work.

**Competing interests** G.C. and P.G. are listed as inventors on patent applications related to this work that are owned and managed by the Boston Children's Hospital and Dana-Farber Cancer Institute. S.A.A. has been a consultant and/or shareholder for Neomorph Inc, Imago Biosciences, Cyteir Therapeutics, C4 Therapeutics, Nimbus Therapeutics and Accent Therapeutics. S.A.A. has received research support from Janssen and Syndax. S.A.A. is named as an inventor on a patent application related to MENIN inhibition WO/2017/132398A1. J.A.M. is a founder, equity holder and advisor to Entact Bio, serves on the scientific advisory board of 908 Devices and receives sponsored research funding from Vertex, AstraZeneca, Taiho, Springworks and TUO Therapeutics. The other authors declare no competing interests.

**Additional information**
**Correspondence and requests for materials** should be addressed to Pietro Genovese.

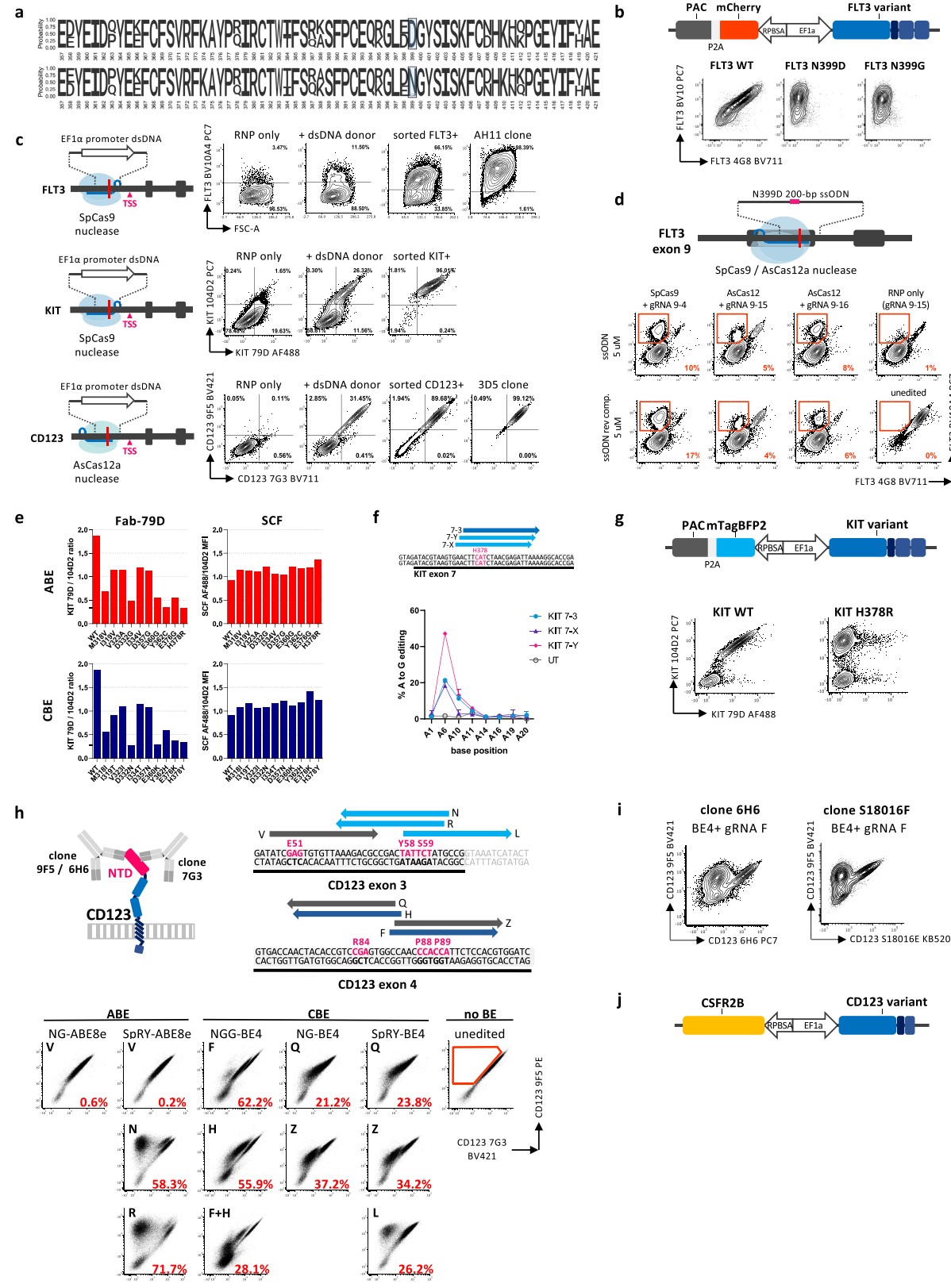

**Extended Data Fig. 1** | See next page for caption.

**Extended Data Fig. 1 | FLT3, KIT and CD123 epitope-engineering can be achieved by base editing. a**. Full length sequence logo of the FLT3 EC4 combinatorial library showing the amino-acid frequency at each position of ECD4 (357 to 421) in FACS-sorted 4G8- and 4G8+ cells. **b**. Top, design of Sleeping Beauty transposon encoding for FLT3 variants with a mCherry and puromycin N-acetyltransferase (PAC) reporter/resistance cassette. Bottom, flow cytometry plots showing loss of 4G8 recognition for N399D and N399G variants expressed in K562 cells. **c**. Generation of FLT3, KIT and CD123 reporter K562 cells through targeted homology-directed repair integration of a EF1α promoter upstream of the gene transcriptional start site (TSS). A dsDNA donor with 50-bp long homology arms was generated by PCR on a plasmid template encoding for a full EF1α promoter. K562 cells were electroporated with SpCas9 (FLT3, KIT) or AsCas12a nuclease (CD123) and gRNAs recognizing a region upstream of the coding sequence of each gene. 0.5 to 10 μg of dsDNA donor template was co-electroporated with Cas RNPs in 20 μL electroporation volume. Representative flow cytometry plots show the population of cells positive for the over-expressed gene, which were FACS-sorted and expanded. For FLT3 and CD123, single cell cloning of sorted cells was performed to isolate clones with the highest surface expression. All epitope-editing tests and optimization were performed on K562 reporter cells, unless otherwise specified. Dual FLT3/CD123 reporters were obtained through a second round of CD123-targeted RNP+donor electroporation on FLT3-expressing K562 cells (data shown in Fig. 6b). Similarly, triple FLT3/CD123/KIT+ K562 were generated by editing the KIT promoter in dual FLT3/CD123 reporter cells. **d**. Introduction of the FLT3 N399D mutation through CRISPR-Cas mediated homology directed repair. K562 reporter cells were electroporated with SpCas9 or AsCas12a nuclease, gRNAs and 200-bp ssODN template donor (or their reverse complement, rev.comp.) encoding for the N399D mutation. Additional silent mutations were included to reduce the risk of nuclease re-cutting after HDR repair. Cells were evaluated by flow cytometry 72h after editing. The percentage of FLT3+ cells (by control mAb BV10A4) but 4G8- is reported in the right bottom corner. **e**. Characterization of KIT mutations derived from epitope mapping. For amino-acid positions deriving from the KIT epitope mapping, substitutions that could be obtained with adenine BE (ABE, red) or cytidine BE (CBE, blue) were individually cloned in a Sleeping Beauty transposon and electroporated into HEK-293T cells. After puromycin selection, cells were stained with both Fab-79D and control Ab 104D2. The ratio between Fab-79D MFI and 104D2 MFI is reported for each mutation. To exclude variants affecting SCF binding to KIT, the same variants were incubated with AF488-conjugated SCF and control mAb 104D2 (which does not impair SCF binding). The ratio of SCF to 104D2 median fluorescence is reported in the bar plots. Horizontal lines show the reference mutation H378R. **f**. KIT H378R adenine base editing optimization. sgRNAs targeting codon H378 within exon 7 were co-electroporated with SpRY-ABE8e in K562 cells. Editing efficiency on gDNA is reported for each adenine within the protospacer (with position numbers relative to KIT-Y sgRNA). **g**. Top, design of Sleeping Beauty transposon encoding for KIT variants with a mTagBFP2 and puromycin N-acetyltransferase (PAC) reporter/resistance cassette. Bottom, flow cytometry plots showing loss of Fab79D recognition for KIT H378R expressed in HEK-293T cells. **h**. CD123 epitope screening by base editing. sgRNAs for targeted base editing of 7G3 contact residues were co-electroporated with 500 ng of adenine (ABE) or cytidine base editor (CBE) expression plasmids in CD123-reporter K562 cells. NGG (wild-type), NG- and SpRY- PMA-flexible Cas9 variants of the base editors were exploited to achieve on-target base deamination. The percentage of cells positive for control mAb 9F5 and negative for therapeutic clone 7G3 is reported in each plot. The unedited condition shows the gating strategy. BE4, evoAPOBEC1-BE4max. **i**. CD123 CBE with sgRNA-F results in loss of clone 6H6 and S18016F binding. The same conditions from the BE screening reported in Extended Data Fig. 1h were stained with CD123-targeting clones 6H6 and S18016F which have a different epitope than clone 7G3. **j**. Design of Sleeping Beauty transposon encoding for CD123 variants with co-expression of the common β-chain CSFR2B to allow intracellular signal transduction.

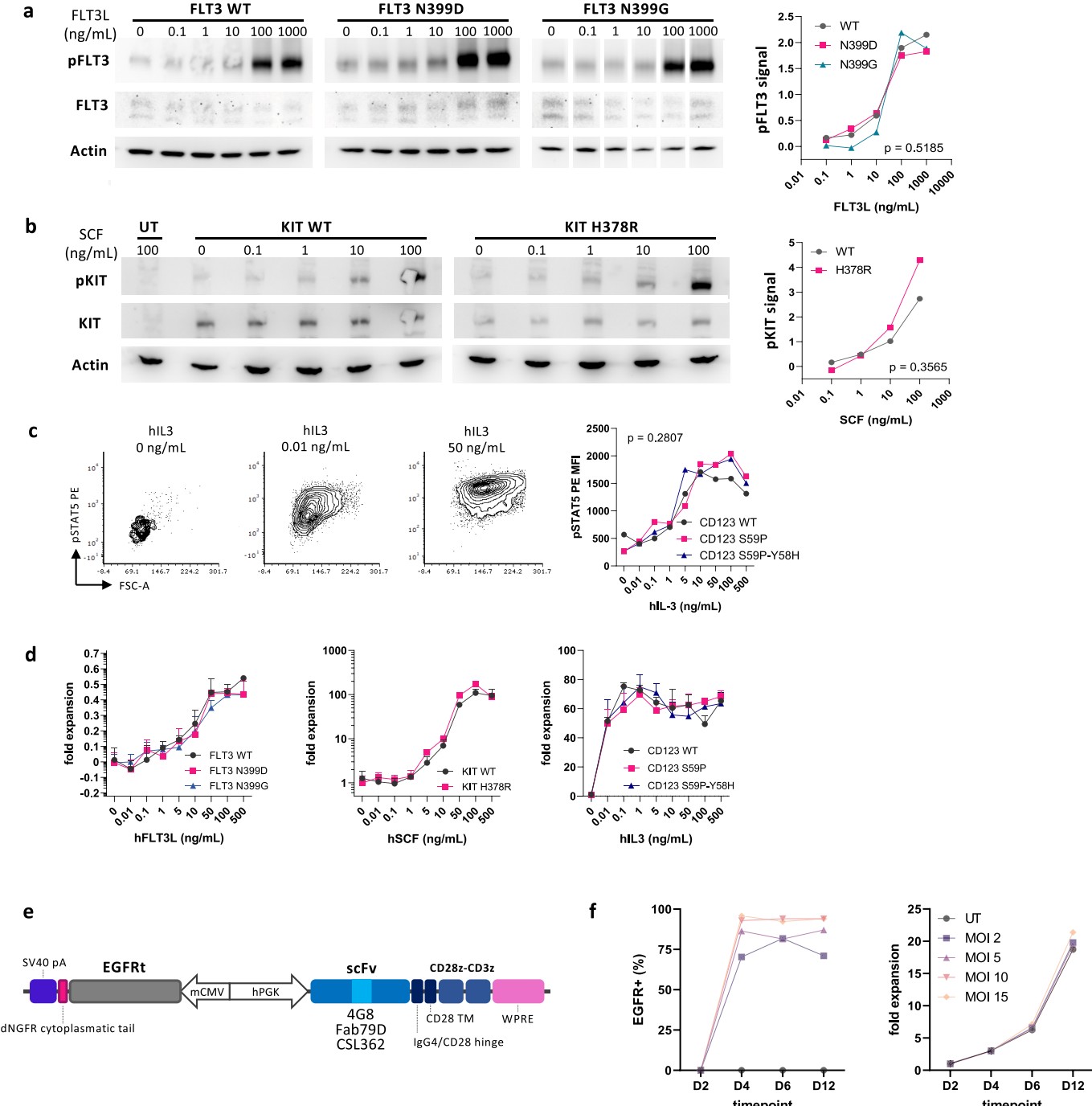

**Extended Data Fig. 2** | See next page for caption.

**Extended Data Fig. 2 | Epitope variant receptors show preserved ligand-mediated activation. a**. FLT3 epitope engineered variants preserve kinase activation. Western blot of proteins extracts from K562 cells expressing FLT3 variants by Sleeping Beauty transposase. Cells were serum-starved overnight and stimulated with different concentrations of human FLT3L for 10 min at 37 °C. pFLT3 Y589-591, total FLT3 and Actin were probed on the same lysates. Total FLT3 was probed after stripping of the pFLT3 membrane. Normalized pFLT3 signal intensity (on actin) is reported on the right. Two-way ANOVA, the p-value of the editing effect is reported. Uncropped blots are reported in Supplementary Fig. 2. **b.** KIT epitope engineered variant preserves kinase activation. Western blot of proteins extracts from NIH-3T3 cells expressing KIT variants by Sleeping Beauty transposase. Cells were serum-starved overnight and stimulated with different concentrations of human SCF for 10 min at 37 °C. pFLT3 Y719, total KIT and Actin were probed on the same lysates. Normalized pKIT signal intensity (on total KIT) is reported in the right plot. Two-way ANOVA, the p-value of the editing effect is reported. Uncropped blots are reported in Supplementary Fig. 2. **c.** CD123 epitope engineered variants preserve STAT5 activation. BaF3 cells expressing CD123 variants by Sleeping Beauty transposase were starved for murine IL-3 and stimulated with different concentrations of human IL-3. Cells were evaluated for STAT5 phosphorylation by intracellular flow cytometry after 48h (left, representative FACS plots show the CD123 S59P condition at different hIL-3 doses; right, pSTAT5 PE MFI). Two-way ANOVA, the p-value of the editing effect is reported. **d.** FLT3, KIT, CD123 epitope engineered variants induce proliferative responses similar to WT receptors. BaF3 cells expressing FLT3, KIT and CD123 variants by Sleeping Beauty transposase were starved for murine IL-3 overnight and stimulated with different concentrations of human FLT3, SCF and IL-3, respectively. Cells were cultured for 5 days and analysed by flow cytometry to obtain absolute counts (CountBeads). Plots report absolute counts normalized to the unstimulated condition. N = 4. **e.** Top: Bidirectional lentiviral vector expressing a 2nd generation CAR and a truncated Epidermal Growth Factor Receptor (EGFRt). **f.** Percentage of EGFRt+ (left) and fold expansion (right) of T cells after transduction with 4G8-CAR at different multiplicity of infection (MOI). Days, D.

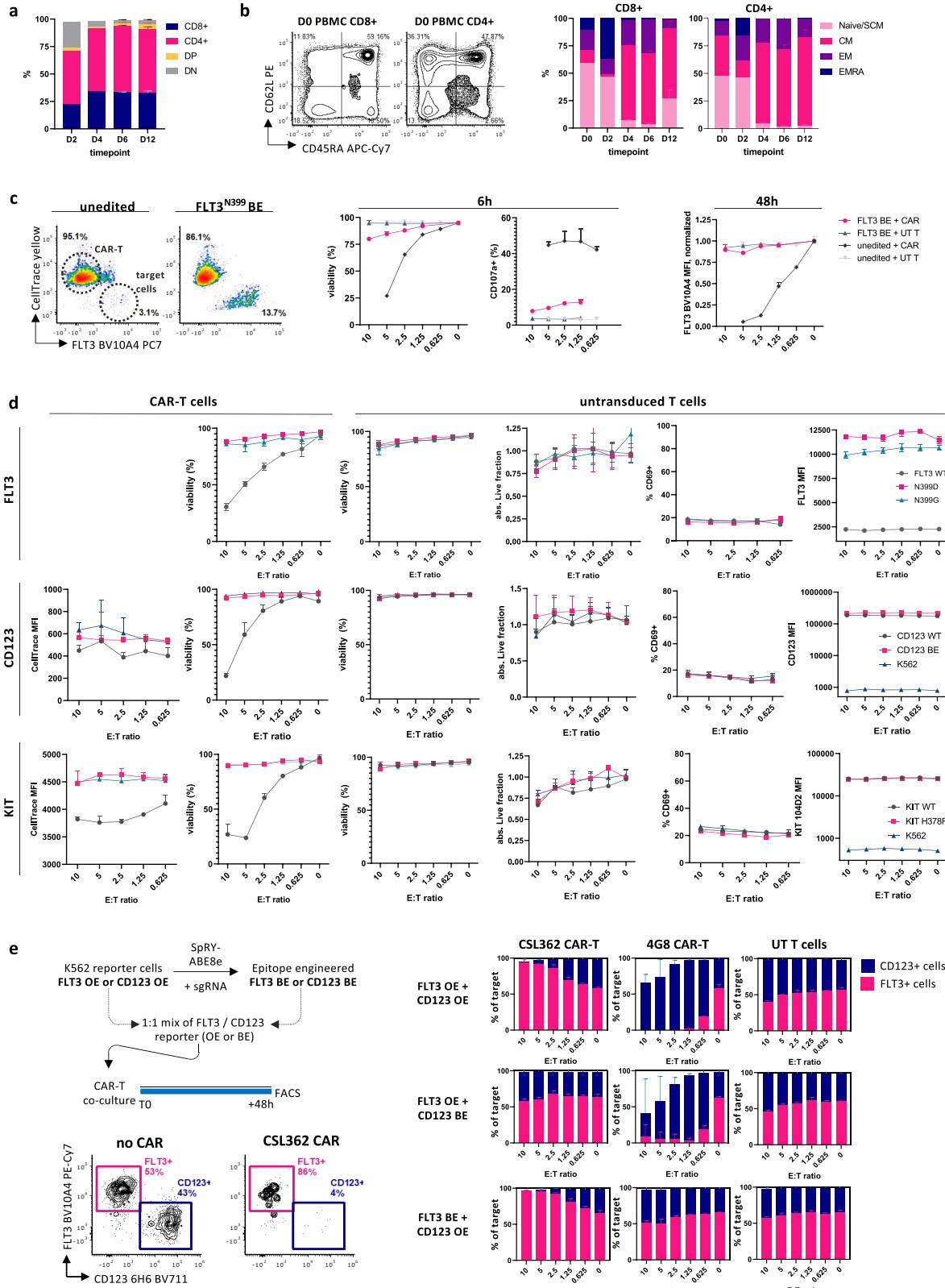

**Extended Data Fig. 3** | See next page for caption.

**Extended Data Fig. 3 | Epitope-edited cells are protected by CAR-T cell killing. a**. CAR-T cell CD4/CD8 composition during *in vitro* culture. Fresh healthy donor-derived PBMC were cultured with CD3/CD28 Dynabeads (bead:cell ratio = 3:1), IL-7 5 ng/mL and IL-15 5 ng/mL and transduced at day (D) 2 with a lentiviral vector (LV) encoding for the 4G8 CAR. The culture composition was evaluated by flow cytometry at days 2, 4, 6, 12. The plot reports N = 5 conditions LV-transduced with different multiplicity of infection (MOI). Mean ± SD. **b**. CAR-T cell phenotype by flow cytometry. T cell subsets were evaluated by CD62L and CD45RA staining (CD45RA+62L+, Naïve/T stem memory cells; CD45RA-62L+, central memory, CM; CD45RA-62L-, effector memory, EM; CD45RA+62L-, terminally differentiated EM cells re-expressing CD45RA, EMRA). Representative FACS plots (left) and the culture composition by CD4+ and CD8+ subsets (right) are reported. D0 refers to uncultured peripheral blood T cells after Ficoll separation. Mean ± SD (N = 5). **c**. FLT3$^{WT}$ cells are eliminated by 4G8 CAR-T cell while FLT3$^{N399}$ BE cells show selective resistance. 4G8 CAR-T cells co-culture assay with FLT3 reporter K562 cells either unmodified or FLT3$^{N399}$ base edited. (Left) Representative flow cytometry plots at early timepoint (6h) gated on live cells (AnnexinV-7AAD-). T cells are identified by CellTrace marking, while K562 targets by FLT3 expression. (Left to Right)

Target cell viability at 6h (%), T cell degranulation by CD107a surface staining at 6h (%) and FLT3 expression on surviving target cells at 48h (MFI, normalized on E:T = 0). N = 2. Two-way ANOVA, the p-value of the editing effect is reported. **d**. Epitope engineered receptors provide protection from CAR-T cells. Each row reports additional plots from co-cultures with FLT3, CD123 and KIT expressing K562 cells (same experiments reported in Fig. 2c, d, e). By column from left to right: CellTrace MFI of CAR-T cells at 48h of co-culture, Target cell viability after 48h of co-culture with CAR-T cells (%), Target cell viability after 48h of co-culture with untransduced T cells (%), absolute counts of live cells (AnnexinV-7AAD-, normalized on E:T = 0) after 48h of co-culture with untransduced T cells, CD69+ untransduced T cells after 48h of co-culture (%), FLT3, CD123 or KIT MFI on target cells after 48h of co-culture with untransduced T cells. Mean ± SD, N = 4. **e**. Experimental layout for co-culture assays with two populations of target cells, one expressing FLT3 and the other expressing CD123. Unmodified or epitope edited FLT3 and CD123 K562 reporter cells were mixed at ~1:1 ratio and co-cultured with either expressing 4G8 CAR, CSL362 CAR or untransduced T cells. The FLT3+/CD123+ % composition of live target cells (pre-gated on FLT3+ or CD123+) is reported as bar plots for each combination at different effector:target (E:T) ratios. Mean ± SD, N = 4.

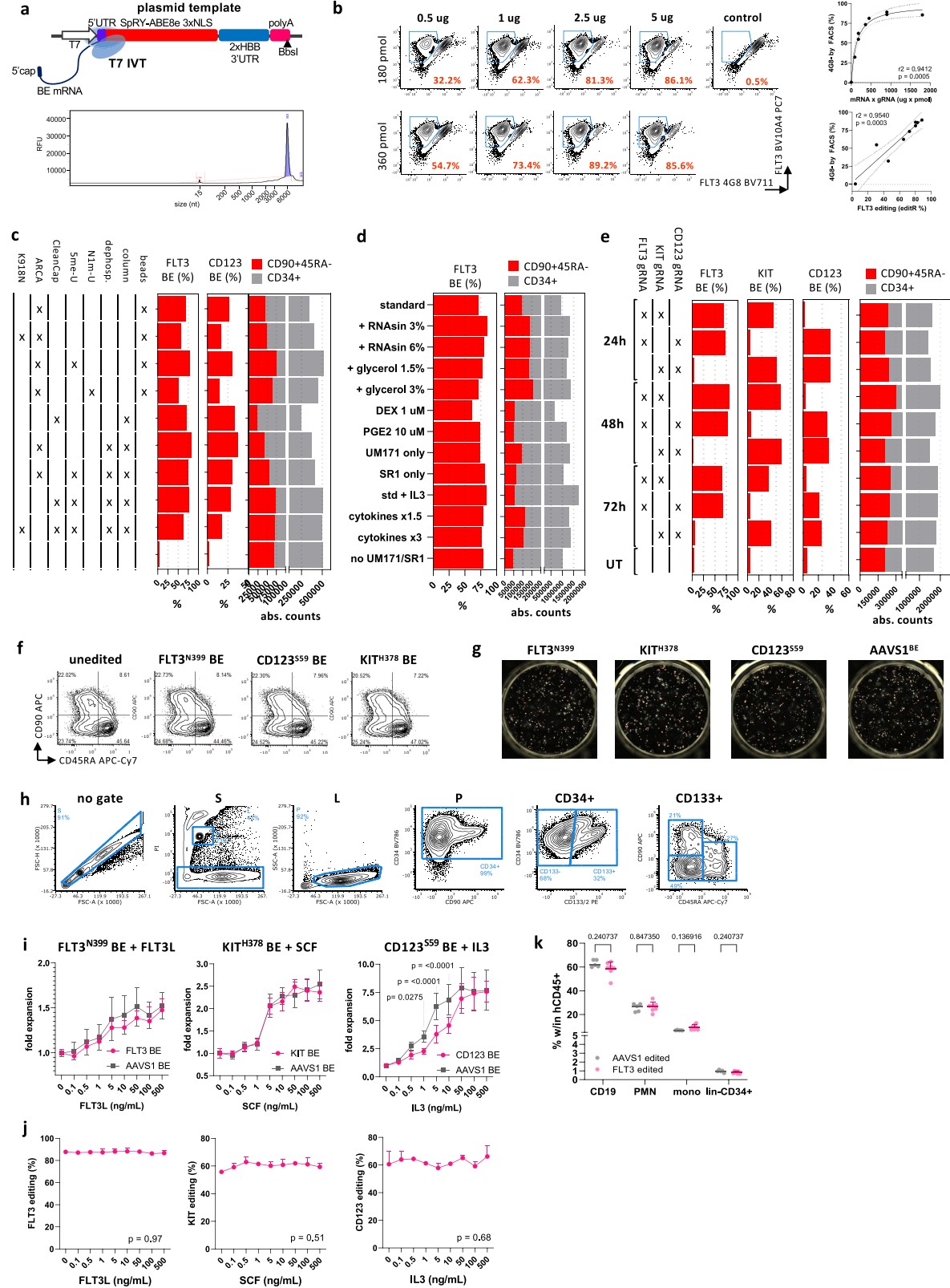

**Extended Data Fig. 4** | See next page for caption.

**Extended Data Fig. 4 | Efficient base editing of FLT3, CD123 and KIT in CD34+ HSPCs. a.** Schematic representation of the plasmid template used for *in vitro* transcription (IVT) of base editor mRNAs. Type-IIS restriction enzyme BbsI was used to linearize the template. T7, T7 RNA polymerase promoter; UTR, untranslated region; HBB, haemoglobin β gene; polyA, poly-adenine sequence (~110-120 nt). (Bottom) Representative plot of purified IVT SpRY-ABE8e mRNA analysed with Agilent Fragment Analyzer RNA for quality control. >90% of IVT mRNA corresponds to the predicted size. **b.** SpRY-ABE8e V106W mRNA dose finding test on FLT3-reporter K562 cells base edited for FLT3$^{N399}$ with sgRNA-18. Right, correlation of FLT3 editing efficiency by flow cytometry with mRNA x sgRNA dose and correlation of FLT3 editing efficiency by flow cytometry and by gDNA analysis. Spearman $r^2$ and p values are reported. **c.** Optimization of CD34+ HSPC base editing by mRNA electroporation. Several SpRY-ABE8e mRNA variants were tested in a dual FLT3/CD123 editing experiment. Tested variables include: mRNA purification method (beads, sparQ PureMag magnetic beads; column, NEB Monarch RNA columns), dephosphorylation, substitution of UTP with N1-methyl-pseudo-uridine (N1m-U) or 5-methoxy-uridine (5me-U), capping technology (CleanCap, Trilink CleanCap AG; ARCA, NEB 3′-O-Me-m7G(5′)ppp(5′)G RNA Cap Structure Analog) and the addition of the K918N SpCas9 mutation (which has been reported to improve SpCas9 catalytic activity)[72]. Bar plots showing FLT3 and CD123 editing efficiencies by genomic DNA (gDNA) analysis (%) and absolute counts of bulk (CD34+) and stem-enriched (CD90+45RA-) cells at the end of *in vitro* culture. **d.** Optimization of culture conditions for base editing. CD34+ HSPCs were base edited with SpRY-ABE8e mRNA and FLT3$^{N399}$ sgRNA with addition of supplements during electroporation (RNAsin, Promega RNAsin RNAse-inhibitor; glycerol) or with different culture conditions, including modulation of cytokine concentrations (standard: 100 ng/mL FLT3L, SCF and 50 ng/mL TPO; 1.5x: 150 ng/mL FLT3L, SCF and 75 ng/mL TPO; 3x: 300 ng/mL FLT3L, SCF and 150 ng/mL TPO; + IL-3: standard with addition of hIL-3 20 ng/mL), different stem-cell preserving compounds (standard: SR-1 0.75 μM, UM171 35 nM; SR-1 only 0.75 μM; UM171 only 35 nM; no SR-1/UM171), addition of anti-inflammatory compounds (PGE2, Prostaglandin-E2 10 μM; DEX, dexamethasone 1 μM). Bar plots showing FLT3 editing efficiencies by gDNA analysis (%) and absolute counts of bulk (CD34+) and stem-enriched (CD90+45RA-) cells at the end of *in vitro* culture. **e.** CD34+ HSPCs were electroporated at different timepoints (24h, 48h, 72h) after thawing to select the best timing for editing. Each condition was edited once for all combinations of two of our selected targets (FLT3, CD123, KIT). Bar plots show the editing efficiencies by gDNA analysis (%) and absolute counts of bulk (CD34+) and stem-enriched (CD90+45RA-) cells at the end of *in vitro* culture. **f.** Representative flow cytometry plots of the stem cell surface phenotype (CD90/CD45RA subsets) of FLT3, CD123, KIT and AAVS1 base edited CD34+ HSPCs. Plots are pre gated on Live CD34+133+ cells. **g.** Uncropped photomicrograph of colony forming assays plated with in vitro epitope edited CD34+ HSPCs (same conditions as Fig. 3h). **h.** Representative flow cytometry plots showing the gating strategy used for analysis of edited CD34+ HSPCs. From left to right, cells were gated for singlets (FSC-H/FSC-A plot), Live (PI/FSC-A plot; PI, propidium iodide), physical parameters (SSC-A/FSC-A plot), CD34+ (CD34/CD90 plot), CD133+ (CD34/CD133 plot), and CD45RA-90-, CD45RA-90-, CD45RA+90- (CD45RA/90 plot). After pre-gating on singlets (S), a bead (B) gate identifies CountBeads (FSC-A$^{low}$PI$^{high}$, which are further gated on two additional fluorescent parameters to exclude debris (not shown). **i.** Epitope-edited HSPCs retain proliferative response to cytokine stimulation. FLT3, KIT and CD123 base edited CD34+ HSPCs were plated with different concentration of the respective ligand and cultured for 4 days. Absolute counts were obtained by flow cytometry using CountBeads. Editing efficiencies at experiment endpoint are reported (technical triplicate were pooled together for gDNA analysis). Mean ± SD. N = 4 on 2 healthy donors. Two-way ANOVA, the p-value of significant comparisons at each concentration are reported. **j.** Editing efficiencies of epitope-edited HSPCs cultured for 4 days with different concentration of the respective receptor ligand (same experiment at Extended Data Fig. 4h). No counterselection of the edited cells was observed. One-way ANOVA. **k.** BM lineage composition of NBSGW xeno-transplanted with FLT3- or AAVS1-edited HSPCs. Same primary xenotransplant reported in Fig. 3i. Mean ± SD. Multiple t-tests. FLT3N399 N = 7, AAVS1-BE N = 4. PMN, polymorphonucleate granulocytes; mono, monocytes; lin-, lineage-negative.

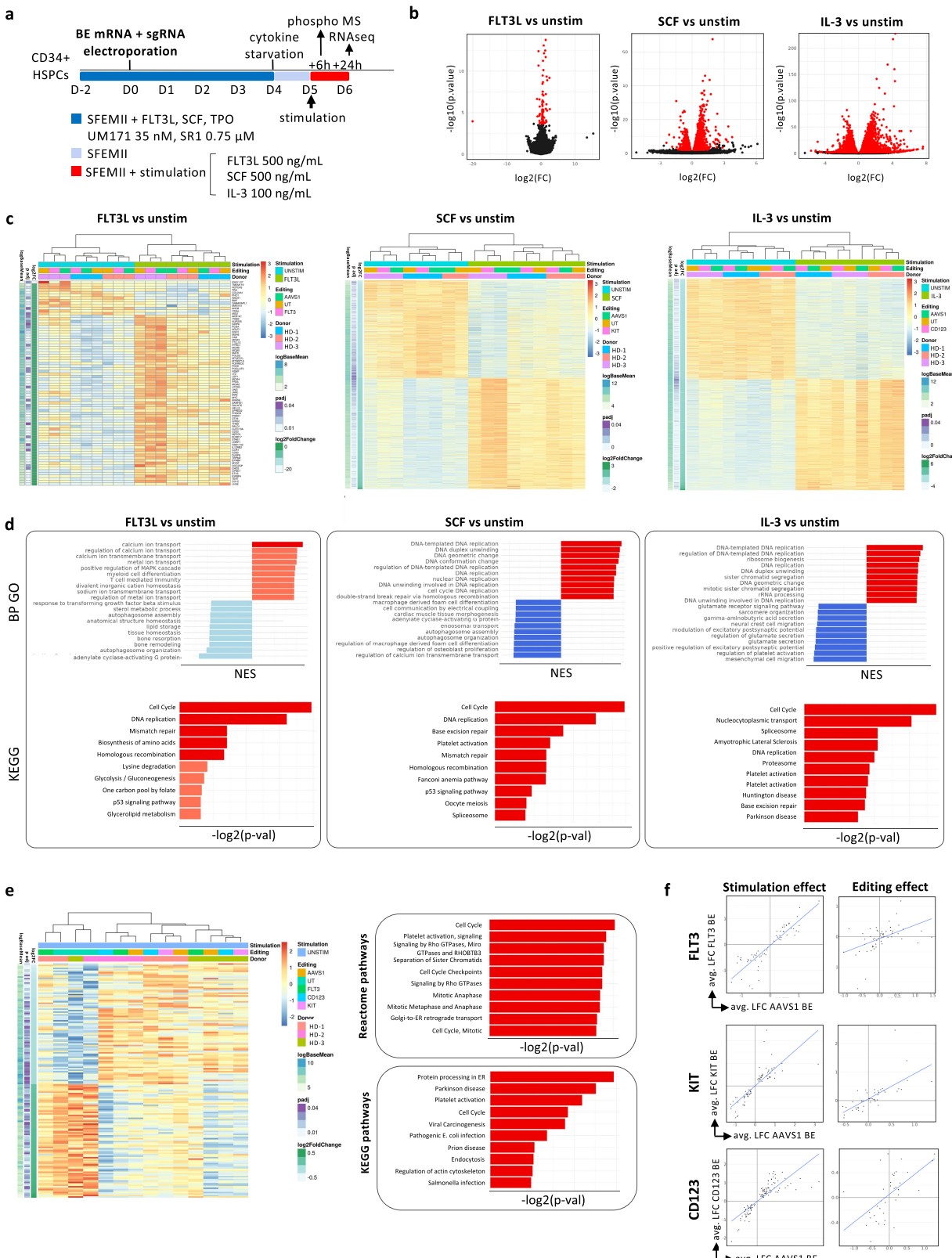

**Extended Data Fig. 5 |** See next page for caption.

**Extended Data Fig. 5 | Epitope editing does not affect downstream signalling and transcriptional response of CD34+ HSPCs upon ligand stimulation. a**. Experimental layout for RNAseq and phospho-profiling by mass spectrometry of epitope edited CD34+ HSPCs. Cells were thawed and edited as previously described, cultured for 4 days to allow substitution of the receptor protein on the cell surface, starved for cytokines (24h) and stimulated with the respective ligand (FLT3L, IL-3, SCF). Cells were harvested at 6h and 24h after the start of stimulation for phospho-MS and RNAseq respectively. The experiments were performed in biological replicate (N = 3 and N = 2 CD34+ donors for RNAseq and MS, respectively). **b**. Volcano plots showing significantly differentially expressed genes (DEGs) in response to each receptor stimulation. The full list is available in Supplementary Tables 3 to 5. **c**. Heatmaps reporting the significant DEGs for each receptor/ligand couple (FDR-corrected p<0.05). Each column corresponds to a sample (see mapping reported above the heatmap and legend on the right), while rows correspond to gene transcripts. As depicted by the dendrogram, unbiased clustering shows predominant effects of the (1) stimulation and (2) donor, while editing conditions are distributed within each cluster. Full comparisons are available in Supplementary Tables 10–12. **d**. Enrichment analyses for the comparison of (from left to right) FLT3L, IL-3 and SCF vs unstimulated. Top row reports the GSEA analysis for the Biological Process (BP) gene ontology. The top 10 terms by significance for NES<0 or NES>0 are reported. Dark red, FDR<0.05 and NES>0, Light red FDR>0.05 and NES>0, Dark blue FDR<0.05 and NES<0, Light blue FDR>0.05 and NES<0. The bottom row reports top 10 terms by significance for KEGG pathways enrichment analysis. Dark red, FDR < 0.05, Light Red FDR > 0.05. The full list of terms is available in Supplementary Tables 7–9. **e**. Left, Heatmap of differentially expressed genes (DEGs) from the UT vs AAVS1 comparison (editing effect) on CD34+ HSPC RNAseq samples. Full list of DEGs is available in Supplementary Table 6. Right, enrichment analysis of the UT vs AAVS1 DEGS, the top 10 terms by significance for KEGG and Reactome pathways are reported. Full lists are available in Supplementary Tables 7–9. **f**. Phospho-proteomic analysis of edited CD34+ HSPCs by mass spectrometry. Scatter plot showing the concordance of the log Fold Change (LFC) of phosphorylated sites by MS upon stimulation in receptor edited vs AAVS1 control condition. Sites differentially with differential phosphorylation associated with the Stimulation (left column) or the Editing (right column) are reported in the scatter plots. Heatmaps reporting the phospho-sites for each comparison and editing condition are reported in Supplementary Fig. 3. Full lists of phospho-sites for each comparison are available as Supplementary Tables 13–15.

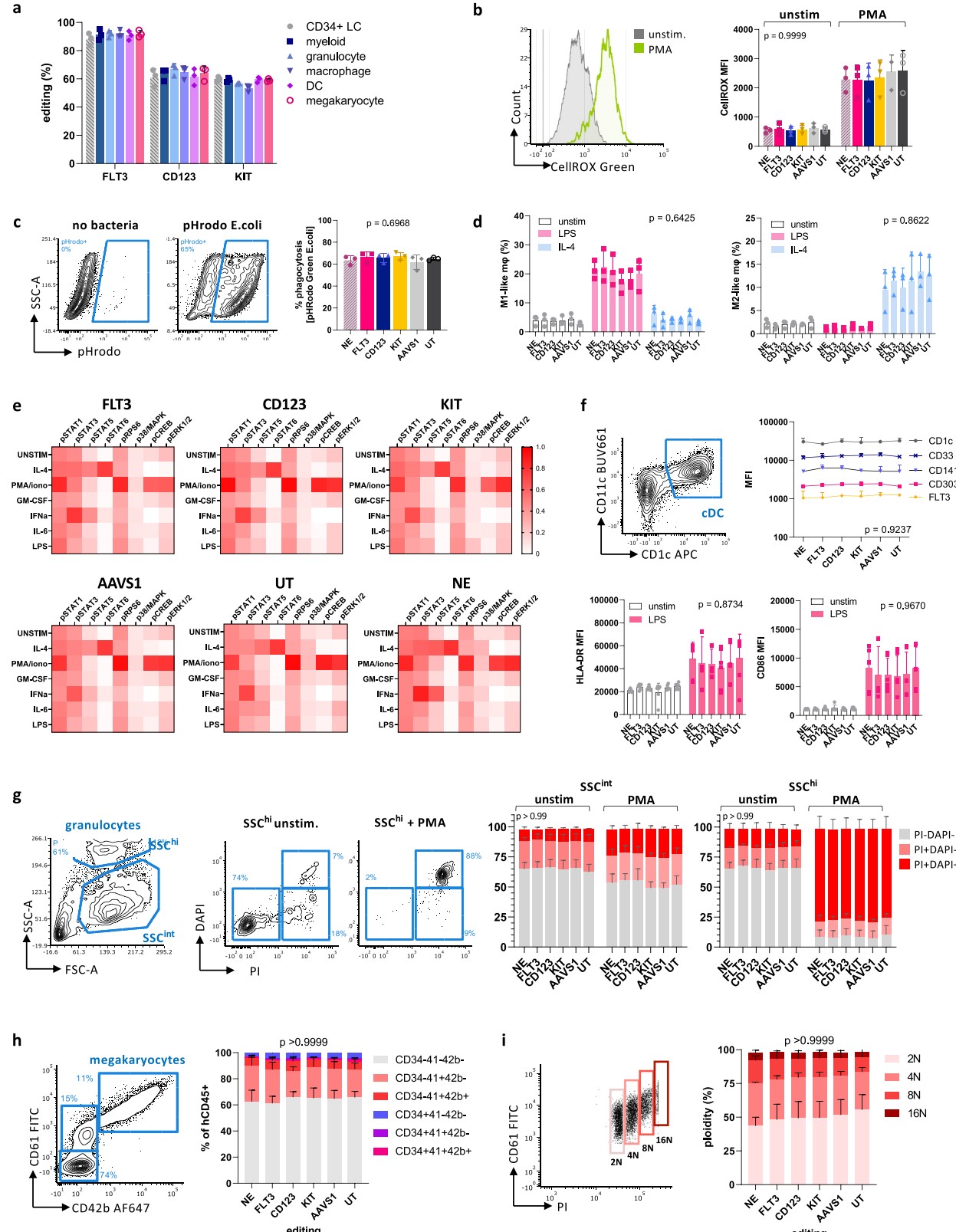

**Extended Data Fig. 6** | See next page for caption.

**Extended Data Fig. 6 | Hematopoietic cells derived from epitope-edited HSPCs show preserved function. a**. Editing efficiency of FLT3-, CD123- and KIT- edited CD34+ HSPCs differentiated in vitro toward myeloid, granulocyte, macrophage, dendritic cell and megakaryocyte lineages. N = 3 biological replicates. **b**. Reactive Oxygen Species (ROS) production of differentiated myeloid cells unstimulated or stimulated with PMA 5 ng/ul for 15 min at 37 °C, as measured by CellROX Green fluorescence. N = 3 biological replicates. Two-way ANOVA, the p-value of the editing effect is reported. **c**. Phagocytosis of E.coli loaded with pHrodo Green dye, which becomes fluorescent upon acidification of the phagolysosome. In vitro differentiated macrophages were incubated with E.coli for 60 min at 37 °C and then analysed by flow cytometry. Left, representative FACS plots. Right, % of phRodo+ macrophages. N = 3 biological replicates, technical duplicate. One-way ANOVA. **d**. Polarization of in vitro differentiated macrophages incubated with LPS or IL-4. Cells were stimulated with LPS 100 ng/mL or IL-4 20 ng/mL and then analysed by flow cytometry. The % of cells with and M1-like or M2-like phenotypes are reported in the bar plots. N = 3 biological replicates. Two-way ANOVA, the p-value of the editing effect is reported. **e**. Phospho-flow of in vitro differentiated myeloid cells stimulated with IL-4, PMA/ionomycin, GM-CSF, IFN type-I, IL-6 or LPS. MFI of each phosphorylated marker was scaled to the range between the FMO control and the highest measured value. The heatmaps show comparable phosphorylation patterns between editing conditions. **f**. In vitro differentiation of classical dendritic cells and expression of co-stimulatory surface markers upon LPS stimulation. Top left, representative FACS plot showing gating for CD1c+CD11c+ DCs. Top right, MFI of surface markers on gated cDC. Two-way ANOVA, the p-value of the editing effect is reported. Bottom, HLA-DR and CD86 MFI on cDC stimulated with LPS. N = 3 biological replicates. Two-way ANOVA, the p-value of the editing effect is reported. **g**. In vitro differentiation of granulocytes and NETosis induction by PMA stimulation. Left, representative FACS plots showing composition of differentiation culture with SSC$^{int}$ and SSC$^{hi}$ populations (both are >80% CD33+66b+). Upon PMA stimulation and NETosis induction, the released nucleic acids are stained by DAPI and PI. Right, bar plots reporting the culture composition by DAPI+ and/or PI+ (%). N = 3 biological replicates, technical duplicates. Two-way ANOVA, the p-value of the editing effect is reported. **h** Differentiation of megakaryocytes from CD34+ HSPCs. Left, representative FACS plot showing surface expression of CD61 and CD42b. Right, culture composition of in vitro differentiated megakaryocytes. Bar plots report the % of CD34, C41, CD42b positive cells. N = 3 biological replicates. Two-way ANOVA, the p-value of the editing effect is reported. **i** DNA staining (PI) shows the generation of polyploid megakaryocytes up to 16N. Left, representative FACS plot showing the relationship between surface CD61 and DNA content. Right, culture composition of gated megakaryocytes by ploidy. Two-way ANOVA, the p-value of the editing (column) effect is reported.

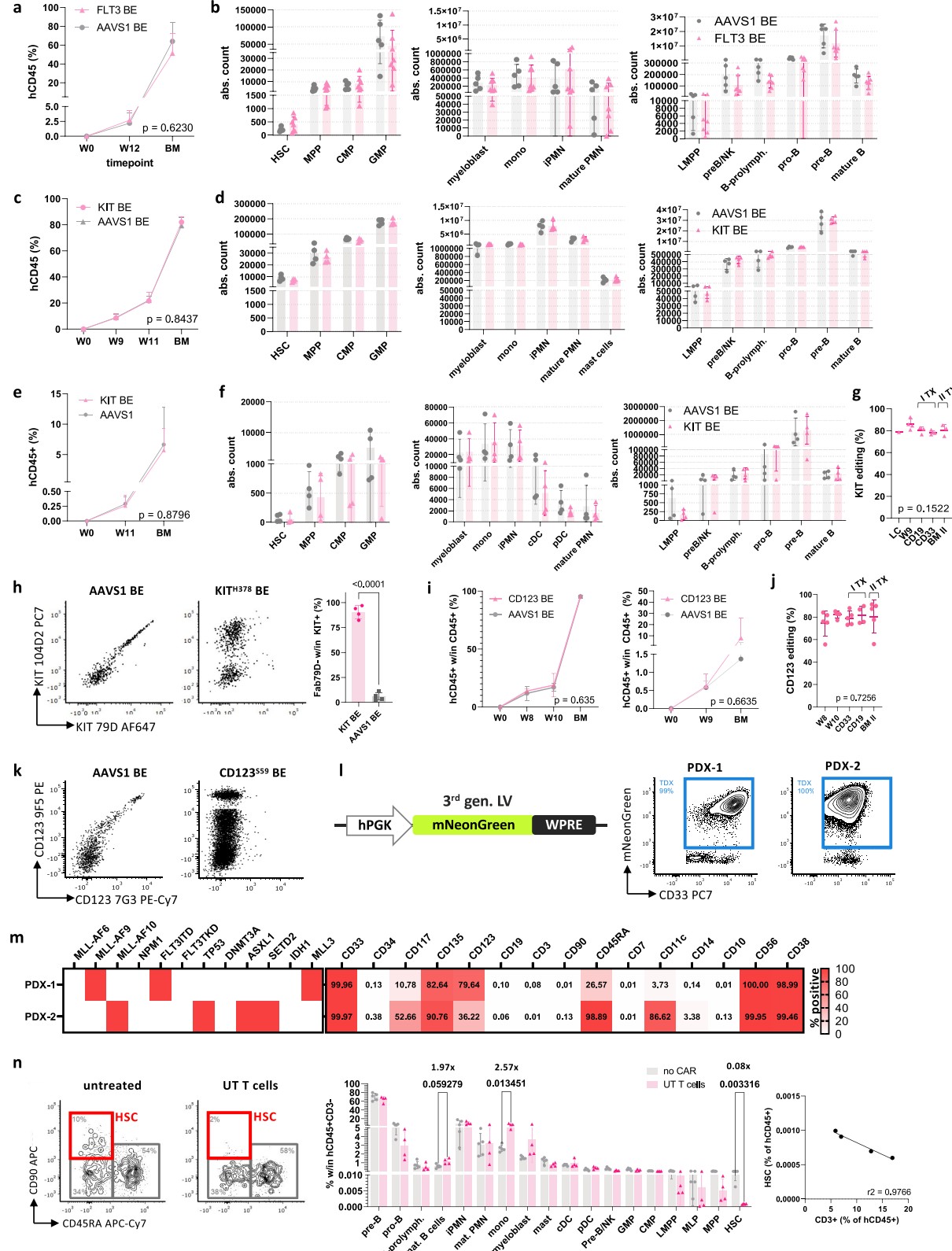

**Extended Data Fig. 7** | See next page for caption.

**Extended Data Fig. 7 | Epitope-edited HSPCs show preserved hematopoietic reconstitution and multilineage differentiation capacity. a**. Human engraftment by flow cytometry (%hCD45+) in the peripheral blood at 12 weeks (W12) and in the bone marrow at week 17 (BM) at endpoint of secondary recipient NBSGW mice xenotransplanted with BM cells from the experiment depicted in Fig. 3i. Each primary was transplanted in one secondary recipient. Mean ± SD. Comparison by 2-way ANOVA, the p-value of the editing effect is reported. **b**. Absolute counts of progenitor (left), myeloid (centre) and lymphoid (right) lineages in the BM of secondary xenotransplanted mice. *AAVS1*[BE] N = 4, FLT3[N399] N = 7. Mean ± SD. HSC, hematopoietic stem cells; MPP, multipotent progenitors; LMPP, lymphoid-primed multipotent progenitors; CMP, common myeloid progenitors; GMP, granulo-mono progenitors; myeloblasts, defined as CD33/66b+19-14-11c-34-SSC[low]; mono, monocytes; iPMN, immature polymorphonucleate granulocytes; mature PMN, mature granulocytes. Multiple t-tests (only significant comparison are reported). **c**. Human engraftment by flow cytometry (%hCD45+) in the peripheral blood at 9, 11 weeks (W9, W11) and in the BM of NBSGW xenotransplanted with 1M CD34+ HSPCs, either *AAVS1*[BE] or KIT[H378] edited. Mean ± SD. Comparison by two-way ANOVA, the p-value of the editing effect is reported. **d**. Absolute counts of progenitor (left), myeloid (centre) and lymphoid (right) lineages in the BM of mice from C. *AAVS1* N = 4, KIT[H378] N = 4. Multiple t-tests (only significant comparison are reported). **e**. Human engraftment by flow cytometry (% hCD45+) in the peripheral blood at 11 weeks (W11) and in the bone marrow at week 14 (BM) of secondary recipient NBSGW mice xenotransplanted with BM cells from primary mice engrafted with KIT[H378] or *AAVS1*[BE] CD34+ HSPCs (experiment described in A). Each primary was transplanted in one secondary recipient. Comparison by two-way ANOVA, the p-value of the editing effect is reported. **f** Absolute counts of progenitor (left), myeloid (centre) and lymphoid (right) lineages in the BM of secondary xenotransplanted mice. *AAVS1* N = 4, KIT[H378] N = 4. Multiple t-tests (only significant comparison are reported). **g**. KIT editing efficiencies measured on liquid culture (LC), total blood cells at week 9 after transplant (W9), on FACS-sorted B (CD19) and myeloid (CD33) BM cells at the end of the experiment, and on total BM from secondary recipients. Mean ± SD. One-way ANOVA. **h**. Left, representative FACS plots showing KIT staining with both therapeutic (Fab-79D) and control (104D2) mAb clones on BM hCD45+33+ cells from secondary xenotransplanted mice. Loss of Fab-79D staining in vivo is consistent with the genomic KIT editing efficiency (reported in G). Right, % of Fab79D+ cells within total KIT+ cells by control staining with clone 104D2. Unpaired t-test. **i** Human engraftment by flow cytometry (% hCD45+) in the peripheral blood and BM at week 14 of NBSGW mice xenotransplanted with CD123[S59] or *AAVS1*[BE] CD34+ HSPCs, in primary (left) or secondary (right) transplants. Comparison by two-way ANOVA, the p-value of the editing effect is reported. **j**. CD123 editing efficiencies measured on liquid culture (LC), total blood cells at week 8 and 10 after transplant (W8, W10), on FACS-sorted B (CD19) and myeloid (CD33) BM cells at the end of the experiment, and on total BM from secondary recipients. Mean ± SD. One-way ANOVA. **k**. Representative FACS plots showing CD123 staining with both therapeutic (7G3) and control (9F5) mAb clones on BM lineage-CD34+ cells from secondary xenotransplanted mice. Loss of 7G3 staining in vivo is consistent with the genomic CD123 editing efficiency (reported in J). **l**. Left, schematic representation of a lentiviral vector encoding for the mNeonGreen fluorescent protein under a hPGK promoter used to transduce human PDXs. Right, representative flow cytometry plots show the transduction efficiency of patient-derived AML xenografts on bone marrow (PDX-1) or spleen (PDX-2) samples. PDX cells were transduced *ex vivo* overnight, transplanted into NBSGW mice for expansion, FACS-sorted for mNeonGreen+ cells and injected into secondary recipients. **m**. Genetic features (left) and surface immunophenotype (right) at thawing of AML PDX used for *in vivo* experiments. Genetic mutations and the % of positive cells for each marker is reported in the heatmap. ITD, internal tandem duplication; TKD, tyrosine kinase domain mutation. **n**. Effect of cultured, non-CAR-transduced T cells on healthy human engraftment in NBSGW mice. Mice were engrafted with 1M CD34+ HSPCs and treated with 2.5 M healthy donor untransduced T cells, expanded in vitro as previously described. While there is no Ag-specific impact on lineage composition, we observed significant depletion of HSCs (0.08x), slight expansion of monocytes (2.57x) and a trend towards B cell increase. Left, representative FACS plots of lineage-CD34+38-10-progenitors, with HSC gating (CD45RA-90+) highlighted in red. Centre, human BM graft composition as % of hCD45+CD3- (to exclude exogenous T cells). Multiple unpaired t-tests, only significant p values are reported. Right, correlation between T cell expansion (CD3+ w/in hCD45+) and decrease in HSC frequency in mice treated with untransduced T cells.

FLT3/KIT/CD123 sgRNAs + SpRY-Cas9 nuclease mRNA

gDNA extraction, library prep and NGS sequencing

293T cells    ssODN    D+4

**a**

| gRNA | offTarget | Peak score | Predicted cleavage score | offTarget_sequence | Guide Alignment to OffTarget | Strand | N guide mismatch | PAM | chr | RNA bulge | DNA bulge | Mismatch + bulge | Exon | entrezID | Gene Symbol | N distinct UMIs |
|---|---|---|---|---|---|---|---|---|---|---|---|---|---|---|---|---|
| FLT3-18 | chr13:-:28048268:28048290 | 4457 | 100 | TTGATAACGGATACAGGTGAGAC | ............... | - | 0 | GAC | chr13 | 0 | 0 | 0 | TRUE | 2322 | FLT3 | 4234 |
| FLT3-18 | chr6:+:3963706:3963728 | 526 | 1.5 | ATCATAACGGATACAGGTGTGAA | A.C............T | + | 3 | GAA | chr6 | 0 | 0 | 3 | FALSE | | | 522 |
| FLT3-18 | chr8:+:96288137:96288159 | 389 | 7.1 | ATGATAAAGGATACAGGTGAATG | A......A......... | + | 2 | ATG | chr8 | 0 | 0 | 2 | FALSE | 9791 | PTDSS1 | 388 |
| FLT3-18 | chr17:-:64112782:64112804 | 134 | 1.3 | ATGATAAAGGATACAGATGAAGA | A......A......A... | - | 3 | AGA | chr17 | 0 | 0 | 3 | FALSE | 2081 | ERN1 | 134 |
| FLT3-18 | chr8:-:97382561:97382583 | 114 | 3.1 | TTGATAAAGGATACAGATGAAGA | .......A......A... | - | 2 | AGA | chr8 | 0 | 0 | 2 | FALSE | | | 113 |
| FLT3-18 | chr16:+:19424663:19424685 | 80 | 7.1 | ATGATAAAGGATACAGGTGAATA | A......ATA | + | 2 | ATA | chr16 | 0 | 0 | 2 | FALSE | 79838 | TMC5 | 79 |
| FLT3-18 | chr11:+:62345996:62346018 | 77 | 1.3 | ATGATAAAGGATACAGATGAAGA | A......A......A... | + | 3 | AGA | chr11 | 0 | 0 | 3 | FALSE | 80150 | ASRGL1 | 77 |
| FLT3-18 | chr6:+:9974765:9974787 | 47 | 2.6 | AAGATAAGGGATACAGGTGAACA | AA.....G......... | + | 3 | ACA | chr6 | 0 | 0 | 3 | FALSE | 266553 | OFCC1 | 47 |
| FLT3-18 | chr14:-:91631139:91631161 | 38 | 0.5 | TAATAAACAGATACAGGTGATAA | .AATA...A......... | - | 5 | TAA | chr14 | 0 | 0 | 5 | FALSE | | | 38 |
| FLT3-18 | chr11:-:101019514:101019536 | 36 | 1.3 | ATGATAAAGGATACAGATGAAGA | A......A......A... | - | 3 | AGA | chr11 | 0 | 0 | 3 | FALSE | | | 36 |
| FLT3-18 | chr17:+:29555067:29555085;chr17:-:29555067:29555086 | 35 | 0 | TTCTTAACGGATACAGG-GAGAG | ..CT..........-.. | + | 2 | GAG | chr17 | 1 | 0 | 3 | FALSE | | | 35 |
| FLT3-18 | chr17:+:29555059:29555081 | 35 | 0 | TTCTTAACGGATACAGGGAGAGA | ..CT.........GAG | + | 5 | AGA | chr17 | 0 | 0 | 5 | FALSE | | | 35 |
| FLT3-18 | chr7:-:69956425:69956447 | 32 | 1.5 | GTAATAACGGTTACAGGTGAGTA | G.A.....T........ | - | 3 | GTA | chr7 | 0 | 0 | 3 | FALSE | 26053 | AUTS2 | 32 |
| FLT3-18 | chr17:-:76198654:76198676 | 28 | 0.1 | ATGAGAACGGATACAGGGTAACC | A...G........GT. | - | 4 | ACC | chr17 | 0 | 0 | 4 | FALSE | 114804 | RNF157 | 28 |
| FLT3-18 | chr8+:28253019:28253039;chr8:-:28253018:28253030 | 25 | 0.020057 | CAGATAACTGATACAGGGTGATAG | CA.....T......^... | + | 3 | TAG | chr8 | 0 | 1 | 4 | FALSE | | | 25 |
| FLT3-18 | chr8+:72672062:72672080;chr8:-:72672061:72672073 | 20 | 0 | TAGATAACTGATACAG-TGAGAG | .A.....T........ | + | 2 | GAG | chr8 | 1 | 0 | 3 | FALSE | 9312 | KCNB2 | 20 |
| FLT3-18 | chr17+:27217043:27217062 | 16 | 0.005185 | ATGATAAAGGA-ACAGGTGAAAG | A......A...-..... | + | 2 | AAG | chr17 | 1 | 0 | 3 | FALSE | | | 16 |
| FLT3-18 | chr4:-21167988:21168008 | 15 | 0 | ATGATAATGGATACAGAGTGATAG | A......T......^.... | + | 2 | TAG | chr4 | 0 | 1 | 3 | FALSE | 80333 | KCNIP4 | 15 |
| FLT3-18 | chr3:+:160033183:160033205 | 13 | 1.3 | ATGATAAAGGATACAGATGAAGA | A......A......A... | + | 3 | AGA | chr3 | 0 | 0 | 3 | FALSE | 101928376 | IL12A-AS1 | 13 |
| FLT3-18 | chrX:-:105899604:105899626 | 11 | 2.4 | TAAATAACGAATACAGGTGAAAG | .AA.....A........ | - | 3 | AAG | chrX | 0 | 0 | 3 | FALSE | 203447 | NRK | 11 |
| FLT3-18 | chr16:-5167263:5167278 | 9 | 0 | CAGATAATGGGATACAGGCGAGAA | CA....T..^.....C.. | - | 4 | GAA | chr16 | 0 | 1 | 5 | FALSE | 105371067 | LOC105371067 | 9 |
| FLT3-18 | chr16:-:21960577:21960599 | 6 | 1 | TTAAAAACGGATACAGGTCAAAA | ..A.A...........C. | - | 3 | AAA | chr16 | 0 | 0 | 3 | FALSE | | | 6 |
| CD123-R | chrX:-:1345408:1345430 | 978 | 100 | ATAGAATAGTCGGCGTCTTTAAC | .............. | - | 0 | AAC | chrX | 0 | 0 | 0 | TRUE | | | 967 |
| CD123-R | chr2:+:31671025:31671045 | 9 | 0.002307 | ATCAAATA-TAGGAGTCTTTTGA | ..CA....-.A..A.... | + | 4 | TGA | chr2 | 1 | 0 | 5 | FALSE | | | 9 |
| KIT-Y | chr4:+:54709436:54709458 | 7583 | 100 | ACTTCATCTAACGAGATTAAAAG | .............. | + | 0 | AAG | chr4 | 0 | 0 | 0 | TRUE | 3815 | KIT | 7012 |
| KIT-Y | chr4:+:63253108:63253130 | 6 | 5.7 | ACTTTATATAACGAGATTAAAAG | ....T..A......... | + | 2 | AAG | chr4 | 0 | 0 | 2 | FALSE | | | 6 |

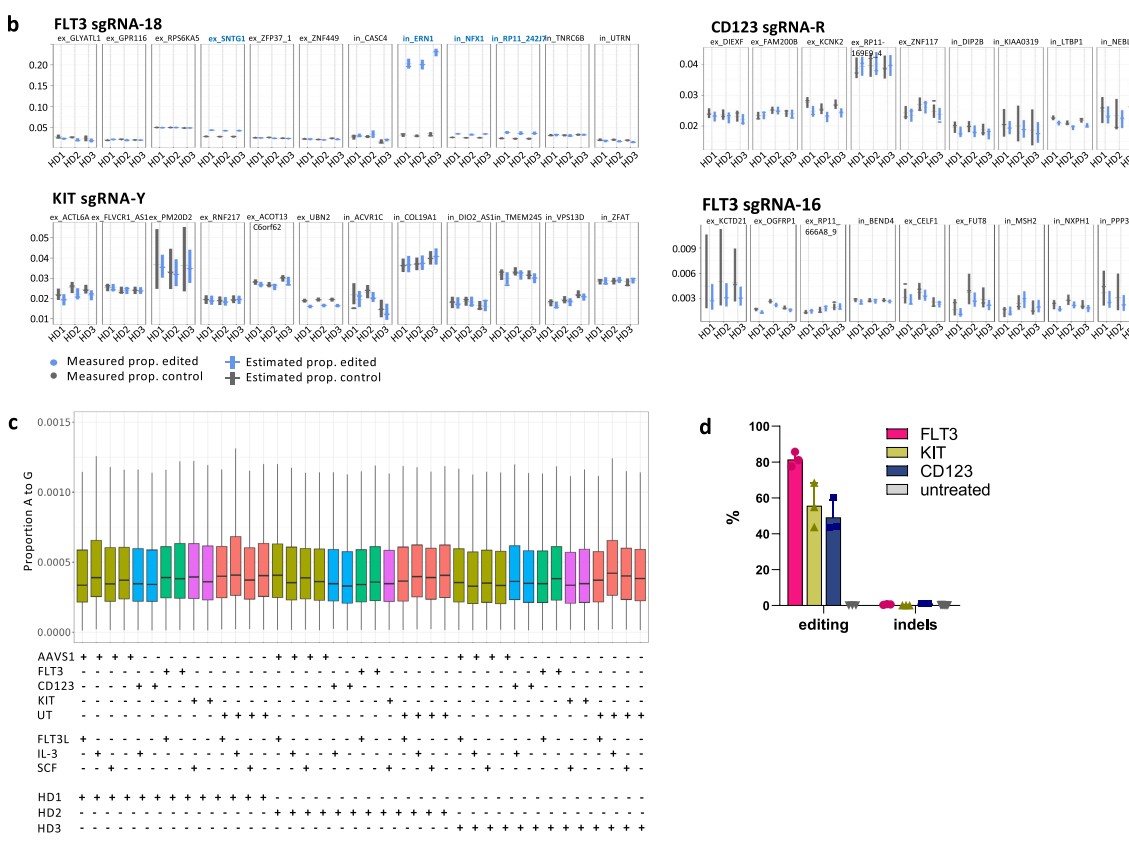

**b**

FLT3 sgRNA-18
ex_GLYATL1, ex_GPR116, ex_RPS6KA5, ex_SNTG1, ex_ZFP37_1, ex_ZNF449, in_CASC4, in_ERN1, in_NFX1, in_RP11_242J7, in_TNRC6B, in_UTRN

CD123 sgRNA-R
ex_DIEXF, ex_FAM200B, ex_KCNK2, ex_RP11-169E9_4, ex_ZNF117, in_DIP2B, in_KIAA0319, in_LTBP1, in_NEBL

KIT sgRNA-Y
ex_ACTL6A, ex_FLVCR1_AS1, ex_PM20D2, ex_RNF217, ex_ACOT13, C6orf62, ex_UBN2, in_ACVR1C, in_COL19A1, in_DIO2_AS1, in_TMEM245, in_VPS13D, in_ZFAT

FLT3 sgRNA-16
ex_KCTD21, ex_OGFRP1, ex_RP11-666A8_9, in_BEND4, in_CELF1, ex_FUT8, in_MSH2, in_NXPH1, in_PPP3CA

Measured prop. edited    Estimated prop. edited
Measured prop. control    Estimated prop. control

**c**

Proportion A to G

AAVS1, FLT3, CD123, KIT, UT, FLT3L, IL-3, SCF, HD1, HD2, HD3

**d**

% editing / indels — FLT3, KIT, CD123, untreated

**Extended Data Fig. 8** | See next page for caption.

**Extended Data Fig. 8 | Off-target analyses of FLT3, CD123 and KIT epitope editing. a** GUIDE-Seq experimental design on 293T cells. GUIDE-Seq results were filtered by number of gRNA mismatches+bulge <6. The detected on-target locus is highlighted in blue. **b** Estimation plots of off-target deamination at predicted sites by NGS sequencing for FLT3-sgRNA-18, CD123-sgRNA-R, KIT-sgRNA-Y and FLT3-sgRNA-16. Loci with significant excess deamination compared to control are highlighted in blue. Additional information on predicted OT sites is reported in Supplementary Fig. 3 and Supplementary Tables 17–20.

**c** Random deamination within RNAseq data of edited CD34+ HSPCs. The proportion of A-to-G conversion observed on the top 5% reads by coverage on RNAseq samples of edited CD34+ HSPCs is reported. Sample editing condition, whether it was stimulated with each ligand and donor source are reported below. Threshold and coverage are reported in Supplementary Table 21. **d**. On-target editing efficiency and indel formation in CD34+ HSPCs (%). Indel frequency was calculated on NGS samples as the proportion of reads harbouring any deletion or base insertion within the sgRNA sequence.

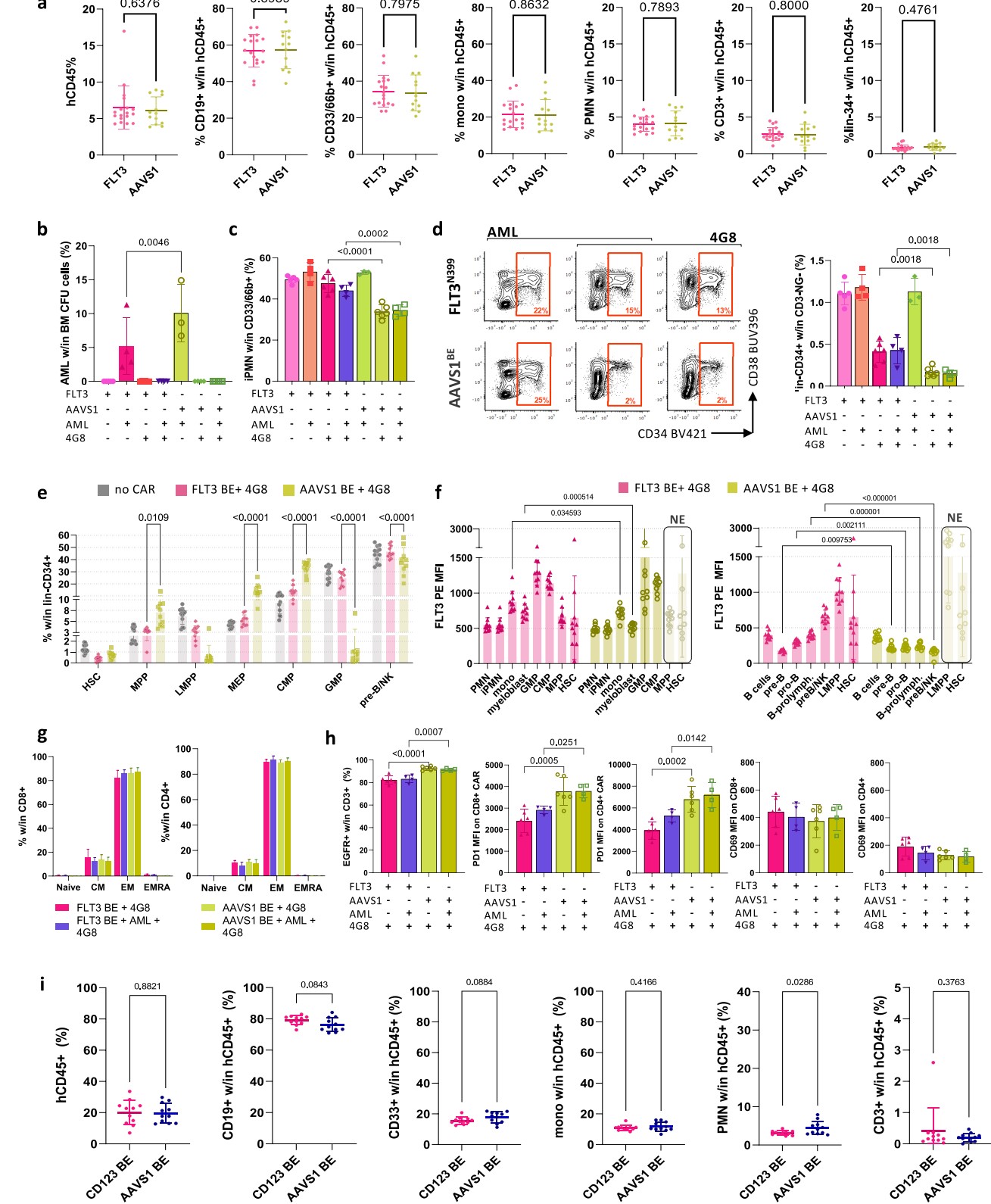

**Extended Data Fig. 9** | See next page for caption.

**Extended Data Fig. 9 | FLT3^N399 HSPCs are resistant to 4G8-CAR in vivo.**
**a**. Peripheral blood lineage composition of NBSGW mice xeno-transplanted with *AAVS1*^BE and FLT3^N399 HSPCs at 8 weeks. Mean ± SD. *AAVS1*^BE N = 13, FLT3^N399 N = 18. Unpaired t-tests. **b**. Percent of AML cells (by mNeonGreen fluorescence) on total BM-derived CFUs plated with samples from the experiment reported in Fig. 4. Mean ± SD. One-way ANOVA. **c**. Percent of immature granulocytes (CD33/66b+14-10-11c-SSChigh) within CD33/66b+ BM cells. Same experiment reported in Fig. 4. Mean ± SD. One-way ANOVA. **d**. Lineage-negative CD34+ progenitors are depleted by 4G8 CAR-T *in vivo* and protected by FLT3^N399 editing (experiment reported in Fig. 4). (Left) Representative flow cytometry plots of lineage-neg cells (mNeonGreen-CD3-19-14-11c-56-) with gating of CD34+ progenitors (% reported within the gate). (Right) % of lin-CD34+ cells within hCD45+3-mNeonGreen- BM cells. Mean ± SD. One-way ANOVA. **e**. Relative composition of BM lin-CD34+ of mice from the experiment reported in Fig. 4. Mean ± SD. two-way ANOVA with multiple comparison (only significant AAVS1 vs FLT3^N399 comparisons are reported). HSC, hematopoietic stem cells; MPP, multipotent progenitors; LMPP, lymphoid-primed multipotent progenitors; CMP, common myeloid progenitors; GMP, granulo-mono progenitors; MEP, mega-erythroid progenitors. **f**. FLT3 expression (MFI) on myeloid (left) and lymphoid (right) BM subsets at experimental endpoint. LMPP, MPP and HSC from 4G8 CAR treated *AAVS1*^BE conditions are not evaluable (NE) due to low cell number. Mean ± SD. Multiple unpaired t-tests. **g**. CAR-T cell phenotype by flow cytometry in the BM of mice from Fig. 4. CD45RA+62L+, Naïve; CD45RA-62L+, central memory, CM; CD45RA-62L-, effector memory, EM; CD45RA+62L, terminally differentiated EM cells re-expressing CD45RA, EMRA. **h**. Bar plots reporting (from left to right) % of EGFRt+ within BM CD3+ cells, PD1 (CD279) MFI and CD69 MFI on BM CD8+ and CD4+ CAR T cells. Mean ± SD. Comparisons between *AAVS1*^BE and FLT3^N399 conditions by one-way ANOVA are reported. **i**. Peripheral blood lineage composition of NBSGW mice xeno-transplanted with *AAVS1*^BE and CD123^S59 HSPCs at 10 weeks (experiment reported in Extended Data Fig. 10). Mean ± SD (N = 11). Unpaired t-tests.

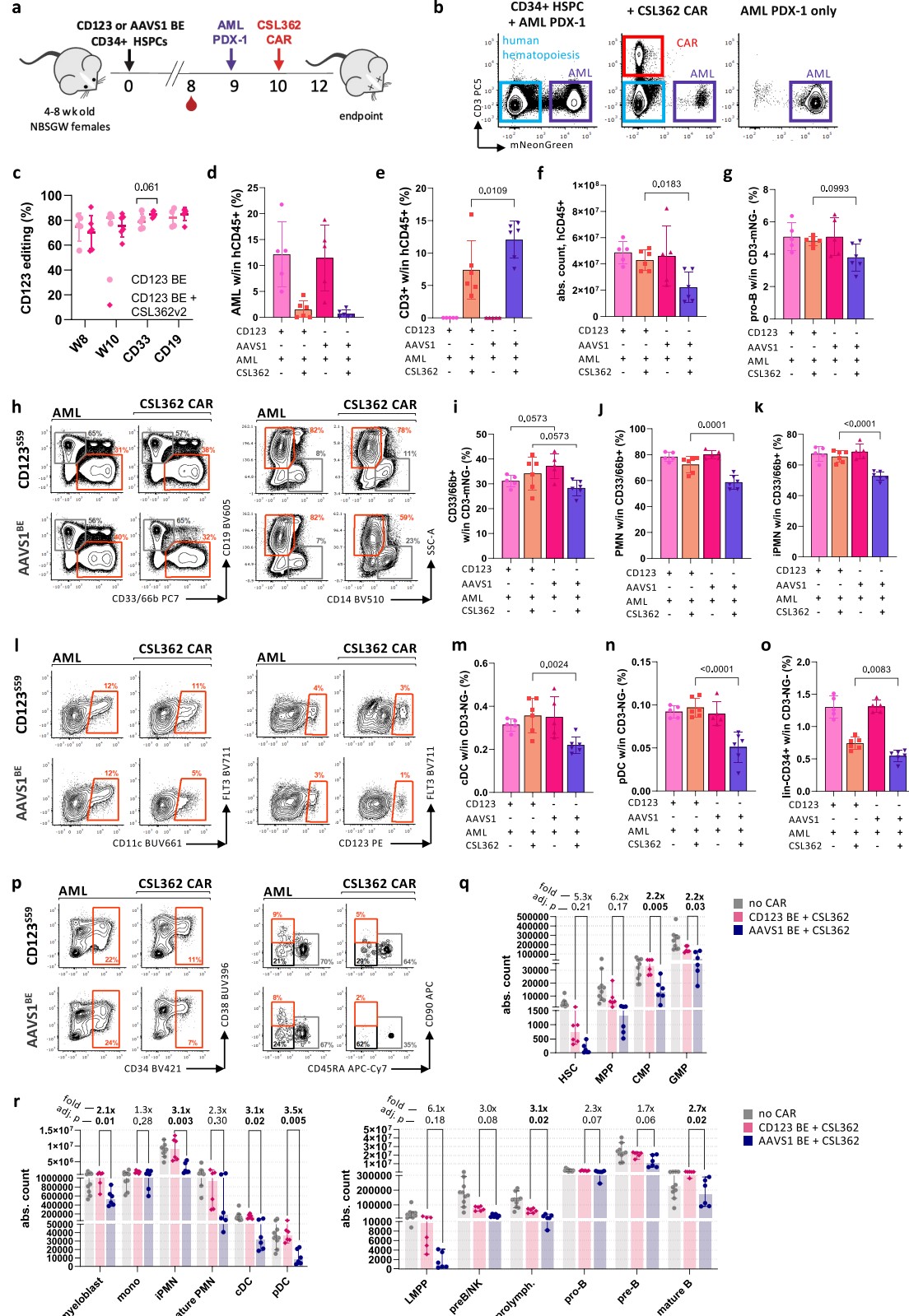

**Extended Data Fig. 10** | See next page for caption.

**Extended Data Fig. 10 | CD123^S59 HSPCs are resistant to CSL362-CAR in vivo.**
**a**. Xeno-transplants of CD123^S59 or *AAVS1*-BE HSPCs co-engrafted with AML PDX-1 and treated with 5M CSL362 CAR-T. **b**. Representative FACS plots of BM samples from mice engrafted with HSPCs+PDX-1 treated with CSL362 CAR-T or untreated. Plots are pre-gated on total human CD45+; CAR-T cells are identified by CD3 staining, while AML PDX cells are mNeonGreen+. **c**. CD123 editing on PB (week 8-10) and sorted CD33+/CD19+ BM cells. CD123^S59 N = 6, *AAVS1* BE N = 5. Mean ± SD. Statistical comparison by multiple unpaired t-test. **d**. Bar plots showing the % of AML PDX cells within hCD45+CD3- BM cells. Mean ± SD. **e**. Percentage of CD3+ cells within hCD45+mNeonGreen- BM cells. Mean ± SD. One-way ANOVA. **f**. Absolute counts of total hCD45+3-mNeonGreen- cells in the BM. Mean ± SD. One-way ANOVA. **g**. Percentage of pro-B cells (CD19+10-34-) within human CD45+3-mNeonGreen- BM cells. Mean ± SD. Statistical comparison of CD123^S59 *vs AAVS1-BE* conditions by one-way ANOVA. **h**. Left: representative FACS plots showing depletion of BM myeloid cells (CD33/66b+19-, highlighted by the orange gate) by CSL362 CAR-T in mice transplanted with CD123^S59 or *AAVS1* BE HSPCs. Right: representative FACS plots showing depletion of granulocytes (PMN, CD33/66b+19-14-SSC^high, orange gate) **i**. Bar plots showing the % of total myeloid cells (CD33/66b+19- within hCD45+ cells), (**j**) PMN (CD33/66b+19- within CD33/66b+19- cells), (**k**) immature PMN (CD33/66b+19-14-10-11c-SSC^high within CD33/66b+19- cells). Mean ± SD. Statistical comparison of CD123^S59 *vs AAVS1 BE* conditions by one-way ANOVA. **l**. Representative flow cytometry plots showing loss of dendritic cells (DC) subsets by CSL362 CAR-T in mice transplanted with CD123^S59 or *AAVS1* BE HSPCs. Left: conventional DC (cDC, CD33/66b+14-11c+FLT3+SSC^low), plots are gated on CD33/66b+14-SSC^low cells. Right: plasmacytoid DC (pDC, CD33/66b+14-11c-FLT3+CD123^high SSC^low), plots are gated on CD33/66b+14-11c-SSC^low cells. **m**. and **n**. Percentage of cDC and pDC within hCD45+3-mNeonGreen- cells, respectively. **o**. Percentage of lineage-CD34+ progenitors within hCD45+3-mNeonGreen- cells. Mean ± SD. Statistical comparison of CD123^S59 *vs AAVS1-BE* conditions by one-way ANOVA. **p**. Lineage-negative CD34+ progenitors are depleted by CSL362 CAR-T and protected by CD123^S59 BE. Left, representative flow cytometry plots of lineage-negative cells (mNeonGreen-CD3-19-14-11c-56-) with gating of CD34+ progenitors. Right, representative flow cytometry plots of lin-CD34+38-10- cells with gating of HSC (CD45RA-90+), MPP (CD45RA-90-), LMPP (CD45RA+90-) subsets. **q**. and **r**. Absolute counts of progenitors (**q**) and myeloid and lymphoid lineage subsets (**r**) in the BM. Untreated mice are pooled together (grey bars), while CSL362-treated CD123^S59 and *AAVS1*-BE mice are reported in pink and blue, respectively. The fold change in absolute counts (CD123^S59/*AAVS1*) for CAR treated groups is reported above each population bar plot. Mean ± SD. One-way ANOVA with multiple comparison. FDR-adjusted p-values of the comparison between CD123^S59 *vs AAVS1 BE* treated with 4G8-CAR are reported (p < 0.05 in bold).

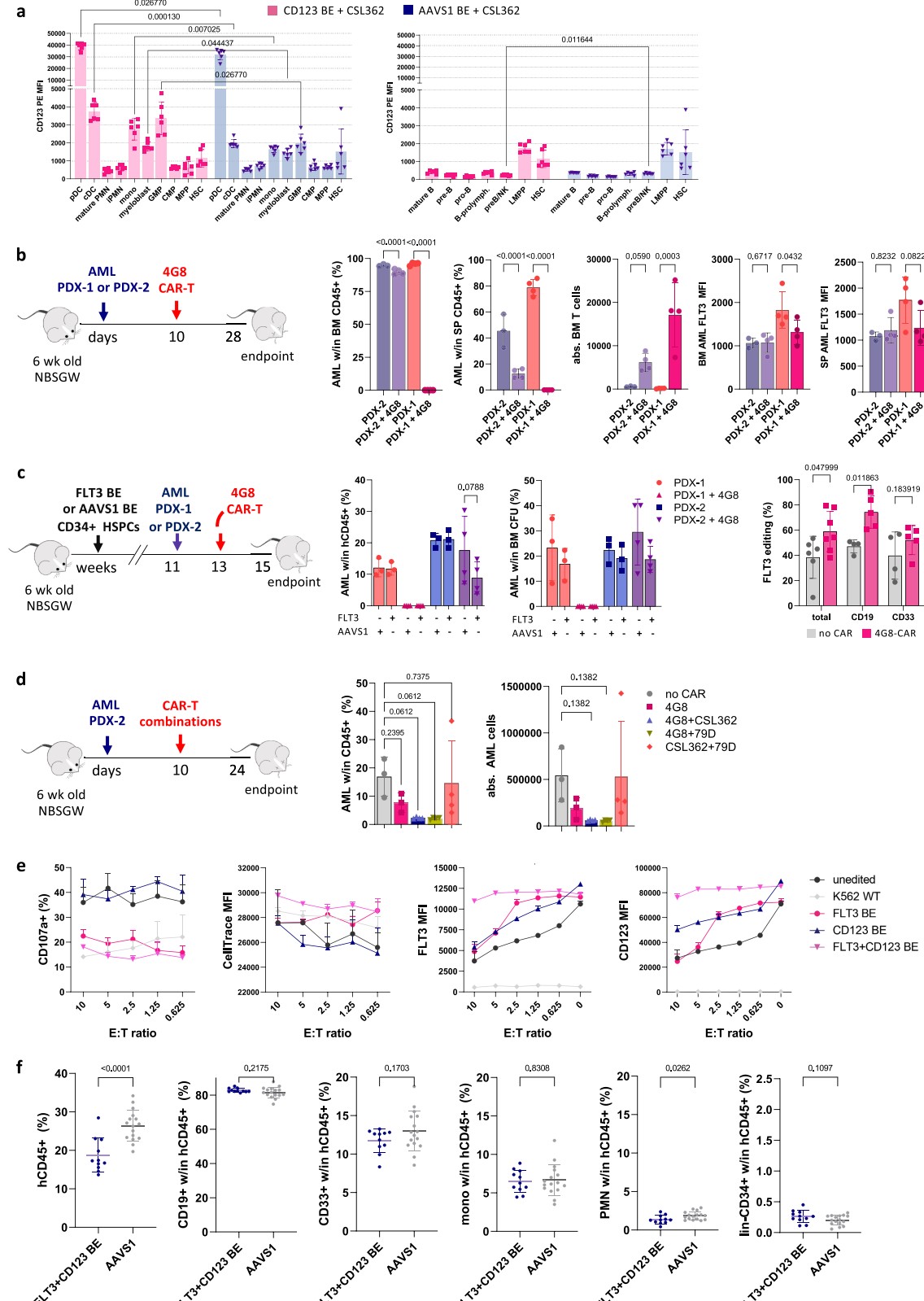

**Extended Data Fig. 11** | See next page for caption.

**Extended Data Fig. 11 | Susceptibility of AML PDX to targeted immunotherapies and improved efficacy of dual-specific CAR-T cells.**
**a.** CD123 expression (MFI) on myeloid (left) and lymphoid (right) BM subsets at the endpoint (experiment reported in Extended Data Fig. 10). Mean ± SD. Statistical differences by multiple unpaired t-test are reported. **b.** 4G8-CAR deplete PDX-1 *in vivo* but fail to eradicate PDX-2. NSBGW female mice were xeno-transplanted with AML PDX cells and, after 10 days, treated with 4G8 CAR-T cells. Experimental endpoint was 18 days after CAR-T administration. From left to right, bar plots report the % of AML PDX cells within total CD45+ cells in the bone marrow (BM), % of AML PDX cells within total CD45+ cells in the spleen (SP), absolute counts of BM T cells, the FLT3 MFI on surviving AML cells in the BM and the FLT3 MFI on surviving AML cells in the SP. Mean ± SD. One-way ANOVA with multiple comparison. **c.** 4G8-CAR T cells deplete PDX-1 *in vivo* but fail to eradicate PDX-2 in mice pre-engrafted with $AAVS1^{BE}$ and FLT3$^{N399}$ HSPCs. NBSGW mice were xeno-transplanted with edited HSPCs and, after 11 weeks, injected with PDX-1 or PDX-2 cells. After 10 days, mice were treated with 2.5 M 4G8 CAR-T cells, and the outcome was evaluated after 2 weeks. Left, bar plots reporting the % of AML cells within total BM hCD45+ cells and within CFU assays plated with total BM cells. Mean ± SD. One-way ANOVA with multiple comparison.

Right, FLT3 editing efficiency within total BM and FACS-sorted CD33+ and CD19+ subsets. 4G8-CAR deplete non-edited hematopoietic cells, resulting in higher editing efficiencies in the CAR-treated condition. Mean ± SD. Comparisons by t-test. **d.** Combinations of CAR-T cells have improved efficacy against PDX-2 *in vivo*. NBSGW mice were xeno-transplanted with 0.75 M PDX-2 cells and, after 10 days, treated with 2.5 M 4G8 CAR or combinations of 4G8 + CSL362, 4G8 + Fab79D or CSL362 + Fab79D CAR T cells (2.5 M each). The outcome was evaluated after 2 weeks. The bar plots show the % and absolute counts of AML cells within total BM hCD45+ cells. Mean ± SD. One-way ANOVA with multiple comparison (vs untreated condition). **e.** Dual FLT3/CD123 specific CAR-T cell in vitro co-culture assay with single and dual FLT3/CD123 epitope-edited HSPCs (same experiment reported in Fig. 5c). From Left to Right, T cell degranulation (CD107a+, top-right), median fluorescent intensity (MFI) of CellTrace (bottom-left), and FLT3 and CD123 expression (MFI) on target cells (bottom-centre and right). Mean ± SD (N = 4). Statistical comparisons are reported in Supplementary Table 24. **f.** Peripheral blood lineage composition of NBSGW mice xeno-transplanted with $AAVS1^{BE}$ and dual-edited FLT3$^{N399}$/CD123$^{S59}$ HSPCs at 9 weeks (same experiment as Fig. 5D). Mean ± SD. Unpaired t-tests.

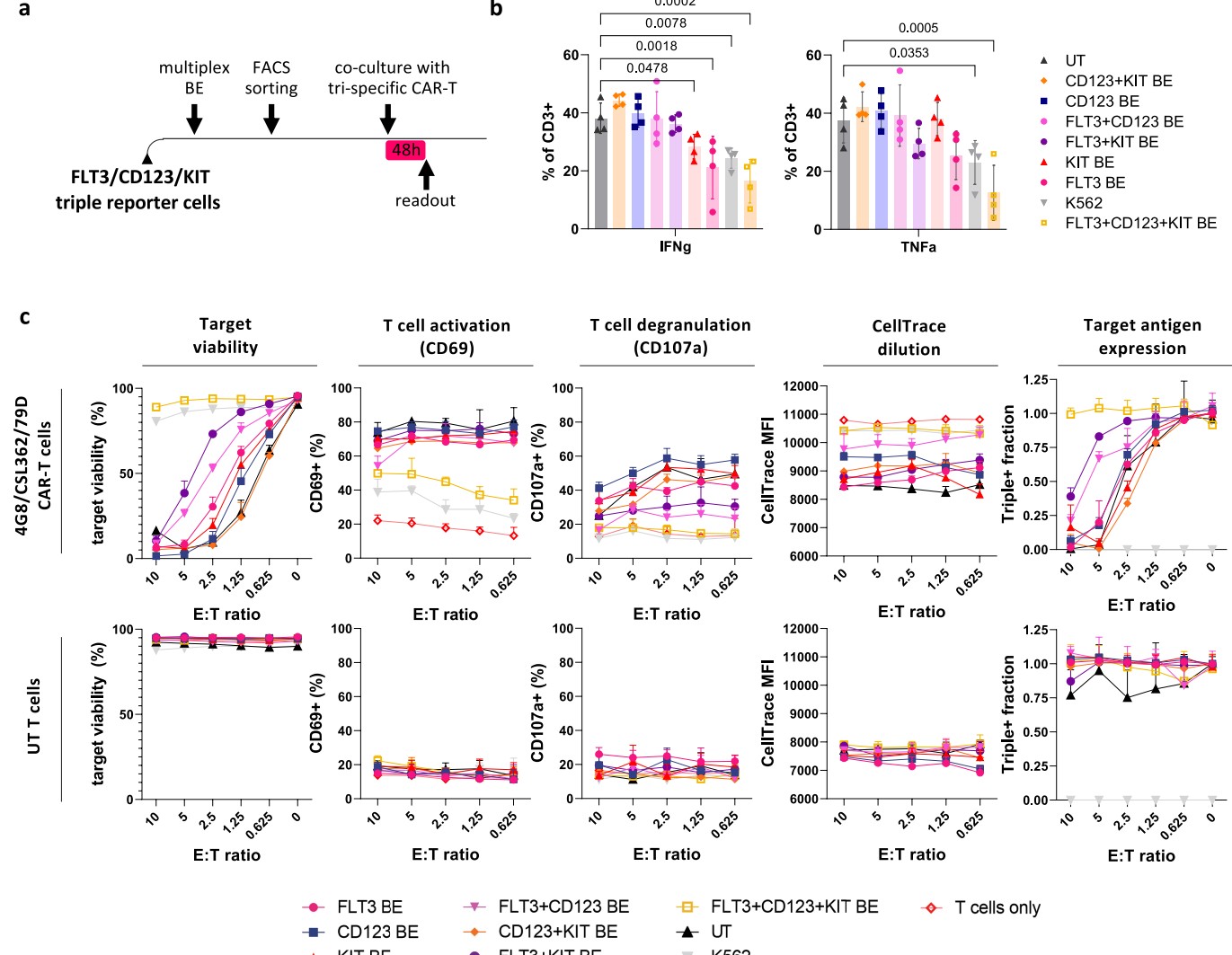

**Extended Data Fig. 12 | Triple epitope-edited cells are resistant to multi-specific FLT3/CD123/KIT CAR-T cells. a**. Triple epitope-edited cells are resistant to FLT3/CD123/KIT triple specific CAR-T cells. Triple FLT3/CD123/KIT reporter K562 were edited as previously described for all 3 targets and then FACS-sorted to isolate single-, double- and triple-edited cells, along with unedited K562. Purified populations were then co-cultured at different effector: target ratios with FLT3/CD123/KIT triple specific CAR-T cells obtained by co-transducing healthy-donor derived with LV vectors encoding the 4G8, CSL362 and 79D CAR constructs. Untransduced T cells were also included as a control. Co-cultures were incubated for 48h and then analysed by flow. N = 4 technical replicates. **b**. Interferon-γ and TNFα production by T cells assessed by flow cytometry (% of IFNγ+ and TNFα+ T cells). One-way ANOVA with multiple comparison. **c**. From left to right, target cell viability (AnnexinV-7AAD-), T cell activation (CD69+), T cell degranulation (surface CD107a+), T cell proliferation (CellTrace violet dilution), Target antigen expression (% of triple FLT3+/CD123+/KIT+ cells). Top row reports conditions cultured with FLT3/CD123/KIT CAR-T cells, bottom row conditions cultured with untransduced T cells.

# Reporting Summary

## Statistics

For all statistical analyses, confirm that the following items are present in the figure legend, table legend, main text, or Methods section.

| n/a | Confirmed | |
|---|---|---|
| ☐ | ☒ | The exact sample size (*n*) for each experimental group/condition, given as a discrete number and unit of measurement |
| ☐ | ☒ | A statement on whether measurements were taken from distinct samples or whether the same sample was measured repeatedly |
| ☐ | ☒ | The statistical test(s) used AND whether they are one- or two-sided<br>*Only common tests should be described solely by name; describe more complex techniques in the Methods section.* |
| ☐ | ☒ | A description of all covariates tested |
| ☐ | ☒ | A description of any assumptions or corrections, such as tests of normality and adjustment for multiple comparisons |
| ☐ | ☒ | A full description of the statistical parameters including central tendency (e.g. means) or other basic estimates (e.g. regression coefficient) AND variation (e.g. standard deviation) or associated estimates of uncertainty (e.g. confidence intervals) |
| ☐ | ☒ | For null hypothesis testing, the test statistic (e.g. *F*, *t*, *r*) with confidence intervals, effect sizes, degrees of freedom and *P* value noted<br>*Give P values as exact values whenever suitable.* |
| ☒ | ☐ | For Bayesian analysis, information on the choice of priors and Markov chain Monte Carlo settings |
| ☒ | ☐ | For hierarchical and complex designs, identification of the appropriate level for tests and full reporting of outcomes |
| ☐ | ☒ | Estimates of effect sizes (e.g. Cohen's *d*, Pearson's *r*), indicating how they were calculated |

*Our web collection on statistics for biologists contains articles on many of the points above.*

## Software and code

Policy information about availability of computer code

| | |
|---|---|
| Data collection | Immunophenotypic analyses were performed on FACS Fortessa, Fortessa X-20 (BD Pharmingen) or using BDFACS Diva software. Cell sorting was performed on a BD FACS Melody or BD FACS Aria Fusion (BD Biosciences) using BD FACS Diva software v6 or BD FACS Chorus. |
| Data analysis | EditR v1.0.8 (Kluesner M, Nedveck D, Lahr W, Moriarity B. EditR: A method to quantify base editing via Sanger sequencing. The CRISPR Journal. 2018.); drc R package v3.0; ggplot2 R package v3.4; brglm2 R package v0.9; Bioconductor DESeq2 v1.40.2; R Studio v2022.12.0; R version 4.2.2; FCS express v6 (DeNovo Software); GraphPad Prism v9.4; Microsoft Excel 365, v3201. R software v4.3, R Core Team (2021), R Foundation for Statistical Computing, Vienna, Austria. URL https://www.R-project.org/. STEMvision Software Version 2021.05.07.00. |

For manuscripts utilizing custom algorithms or software that are central to the research but not yet described in published literature, software must be made available to editors and reviewers. We strongly encourage code deposition in a community repository (e.g. GitHub). See the Nature Portfolio guidelines for submitting code & software for further information.

## Data

Policy information about availability of data

All manuscripts must include a data availability statement. This statement should provide the following information, where applicable:

- Accession codes, unique identifiers, or web links for publicly available datasets
- A description of any restrictions on data availability
- For clinical datasets or third party data, please ensure that the statement adheres to our policy

All relevant data are included in the manuscript, including source data for the figures, RNAseq, GuideSEQ, Off-target analyses,. RNAseq datasets have been deposited with links to BioProject accession number PRJNA986596 in the NCBI BioProject database (https://www.ncbi.nlm.nih.gov/bioproject/). Targeted amplicon sequencing datasets of off-target sites are deposited as BioProject accession number PRJNA986845. All mass spectrometry data files are available for download at: ftp://massive.ucsd.edu/MSV000092272. All remaining data, including source data, are available in this Article and its Supplementary Information.

## Human research participants

Policy information about studies involving human research participants and Sex and Gender in Research.

| | |
|---|---|
| Reporting on sex and gender | *Use the terms sex (biological attribute) and gender (shaped by social and cultural circumstances) carefully in order to avoid confusing both terms. Indicate if findings apply to only one sex or gender; describe whether sex and gender were considered in study design whether sex and/or gender was determined based on self-reporting or assigned and methods used. Provide in the source data disaggregated sex and gender data where this information has been collected, and consent has been obtained for sharing of individual-level data; provide overall numbers in this Reporting Summary. Please state if this information has not been collected. Report sex- and gender-based analyses where performed, justify reasons for lack of sex- and gender-based analysis.* |
| Population characteristics | *Describe the covariate-relevant population characteristics of the human research participants (e.g. age, genotypic information, past and current diagnosis and treatment categories). If you filled out the behavioural & social sciences study design questions and have nothing to add here, write "See above."* |
| Recruitment | *Describe how participants were recruited. Outline any potential self-selection bias or other biases that may be present and how these are likely to impact results.* |
| Ethics oversight | *Identify the organization(s) that approved the study protocol.* |

Note that full information on the approval of the study protocol must also be provided in the manuscript.

# Field-specific reporting

Please select the one below that is the best fit for your research. If you are not sure, read the appropriate sections before making your selection.

☒ Life sciences ☐ Behavioural & social sciences ☐ Ecological, evolutionary & environmental sciences

For a reference copy of the document with all sections, see nature.com/documents/nr-reporting-summary-flat.pdf

# Life sciences study design

All studies must disclose on these points even when the disclosure is negative.

| | |
|---|---|
| Sample size | Sample size for each experiment was determined by the total number of available treated cells, which is constrained by the human source of the material, to be split among each experimental conditions. Whenever possible we aimed to reach at least 5 replicates per group, thus reaching a minimum and sensible operational criterial for carrying out statistics. In some in vivo experiments, such as secondary transplantation, the total number of available cells was more constrained and limited to what could be harvested from the primary recipients. |
| Data exclusions | For in vivo analyses of multiplex edited HSPCs, one cage of mice (4 mice) from control group was excluded from further analysis because the immunodeficient animals developed evident sign of an opportunistic infection. No other data or sample were excluded from analysis, unless the data point was missing due to technical reasons (sample availability or instrumentation acquisition errors). All these criteria were pre-established. |
| Replication | Number of biological replicates is specified for each experiment in figure legends. All attempts at replication were successful. Inferential techniques were applied in presence of adequate sample sizes (n ≥ 5), otherwise only descriptive statistics are reported. |
| Randomization | Mice were randomly distributed to each experimental group. All in vitro experiments on cell-lines did not require randomization. Choice of healthy donor for experiments with human T cells or CD34+ HSPCs was random. |
| Blinding | All reported outcomes are based on measurable variables assessed by user-independent methods/instruments (eg. absolute cell counts by |

| Blinding | automated counting or CountingBeads, cell immunophenotype by flow cytometry, editing outcomes by genomic Sanger or NGS sequencing). The acquisition of these experimental variables is not affected by blinding. |

# Reporting for specific materials, systems and methods

We require information from authors about some types of materials, experimental systems and methods used in many studies. Here, indicate whether each material, system or method listed is relevant to your study. If you are not sure if a list item applies to your research, read the appropriate section before selecting a response.

## Materials & experimental systems

| n/a | Involved in the study |
|---|---|
| ☐ | ☒ Antibodies |
| ☐ | ☒ Eukaryotic cell lines |
| ☒ | ☐ Palaeontology and archaeology |
| ☐ | ☒ Animals and other organisms |
| ☒ | ☐ Clinical data |
| ☒ | ☐ Dual use research of concern |

## Methods

| n/a | Involved in the study |
|---|---|
| ☒ | ☐ ChIP-seq |
| ☐ | ☒ Flow cytometry |
| ☒ | ☐ MRI-based neuroimaging |

## Antibodies

| Antibodies used | See attached Supplementary Table 2 for the complete list of antibodies |
|---|---|
| Validation | Antibodies targeting FLT3 (clones BV10A4, 4G8), KIT (Fab-79D, 104D2, SR-1, Ab55), CD123 (9F5, 6H6, 7G3, S18016E), which are central to this manuscript were validated on human or murine cell lines overexpressing the target gene by Sleeping Beauty transposase. In particular, K562 cells transduced with the cDNAs of WT human FLT3, CD123 and KIT were used to assess specific binding of clones 4G8, BV10A4, 104D2, 79D, Ab55, 6H6, 9F5, 7G3, S18016F to the intended surface antigen. All antibodies used for the analysis of in vivo samples are commercially available and are validated by the manufacturing companies (Biolegend, BD, R&D) on either human or murine peripheral blood mononuclear cells or cell lines transfected with the appropriate target; antibodies were tested on human peripheral blood mononuclear cells or human bone marrow from healthy donors and titrated to identify optimal staining concentrations. |

## Eukaryotic cell lines

Policy information about cell lines and Sex and Gender in Research

| Cell line source(s) | K562, HEK-293T cells were a kind gift from the Biffi lab (Dana Farber Cancer Institute, Boston, US) and originally obtained from ATCC (CCL-243). NIH-3T3 were a kind gift from the Brendel lab (Boston Children's Hospital, Boston, US) and originally obtained from ATCC (CRL-1658). BaF3 were a kind gift from the Armstrong lab (Dana Farber Cancer Institute, Boston, US) and originally obtained from ATCC (HB-283). |
|---|---|
| Authentication | None of the cell lines used were authenticated (in this work cell lines were used only to express and study transgenes). |
| Mycoplasma contamination | All cell lines were tested periodically for mycoplasma contamination and found negative. |
| Commonly misidentified lines (See ICLAC register) | No commonly misidentified cell lines were used. |

## Animals and other research organisms

Policy information about studies involving animals; ARRIVE guidelines recommended for reporting animal research, and Sex and Gender in Research

| Laboratory animals | NOD.Cg-KitW-41J Tyr + Prkdcscid Il2rgtm1Wjl/ThomJ female mice (aka. NBSGW), 6-8 week old at the time of xenotransplantation with CD34+ HSPCs or AML PDX. Mice were housed in sterile individually ventilated cages and fed autoclaved food and water, with standard 12h day/night light cycle. |
|---|---|
| Wild animals | The study did not involve wild animals. |
| Reporting on sex | Only female mice were included for in vivo experiments due to their superior ability to support the engraftment of human CD34+ hematopoietic stem and progenitor cells (Notta F, Doulatov S, Dick JE. Engraftment of human hematopoietic stem cells is more efficient in female NOD/SCID/IL-2Rgc-null recipients. Blood. 2010;115(18):3704–3707) |
| Field-collected samples | The study did not involve samples collected from the field. |
| Ethics oversight | All animal experiments were performed in accordance to regulations set by the American Association for Laboratory Animal Science |

| Ethics oversight | and Dana Farber Cancer Institute (Boston, MA) Institutional Animal Care and Use Committee (IACUC). Approved protocol number: DFCI#21-002. |

Note that full information on the approval of the study protocol must also be provided in the manuscript.

# Flow Cytometry

## Plots

Confirm that:

☒ The axis labels state the marker and fluorochrome used (e.g. CD4-FITC).

☒ The axis scales are clearly visible. Include numbers along axes only for bottom left plot of group (a 'group' is an analysis of identical markers).

☒ All plots are contour plots with outliers or pseudocolor plots.

☒ A numerical value for number of cells or percentage (with statistics) is provided.

## Methodology

| Sample preparation | Cell lines: harvested, washed, incubated with Fc-block reagent and stained at 4C for 15-30 min, then washed. Mouse peripheral blood: collected, lysed with ACK reagent for 10 minutes at room temperature, then washed (repeat 2x lysis and wash); incubated with human and murine Fc-block, stained at room temperature for 15 minutes, then washed. Bone marrow, spleen: collected, washed, lysed with ACK reagent for 5 minutes at room temperature and washed; incubated with human and murine Fc-block, stained on ice for 35 minutes, then washed. BD Brilliant Stain buffer was included when multiple Brilliant Violet or UV dyes were co-stained. Viability staining was performed using Live/Dead yellow, 7-AAD, Propidium Iodide and AnnexinV staining depending on the experiment. All wash steps were performed with PBS + 2% FBS or PBS + 2% FBS + 2mM EDTA. |

| Instrument | BD Fortessa flow cytometer, BD Fortessa X-20, BD FACS Melody sorter |

| Software | BD Diva (acquisition, sorting), BD Chorus (sorting), FCS Express v6 (analysis). |

| Cell population abundance | BM samples were acquired with a target of 2-4 million total events. The median composition of the total human CD45+ engraftment is as follows: 69% CD19+ 27% CD33/66b+, 2.5% monocytes, 0.42% cDC, 13% immature PMN, 6.7% mature PMN, 0.37% mast cells, 8.9% pro-B, 59% pre-B, 0.57% mature B cells, 0.11% NK, 0.42% B-prolymphocytes, 1.4% myeloblasts, 1.1% lin-CD34+, 0.006% MLP, 0.38% preB/NK, 0.37% GMP, 0.13% CMP, 0.04% MEP, 0.08% LMPP, 0.04% MPP, 0.01% HSC. See Supplementary Data FIG.4 for relative abundance and gating of each populations. We also report absolute counts of all hematopoietic populations from in vivo experiments in the Main Figures and Source Data. |

| Gating strategy | See Methods, Supplementary Data Table 1 and Supplementary Data FIG.6 for Gating strategies. In vivo BM gating: pre-gating singlets (FSC-A/FSC-H)> live (PI-) > cells (FSC-A/SSC-A)<br>total hematopoietic cells hCD45+ or mCD45+<br>human cells hCD45+<br>murine cells mCD45+<br>AML PDX hCD45+mNeonGreen+<br>T cells (CAR) hCD45+CD3+<br>healthy hematopoiesis (AML and T cells excluded) hCD45+CD3-mNeonGreen-<br>myeloid cells hCD45+CD3-mNeonGreen-CD33/66b+19-<br>total granulocytes hCD45+CD3-mNeonGreen-CD33/66b+19-14-SSChigh<br>immature granulocytes hCD45+CD3-mNeonGreen-CD33/66b+19-14-10-SSChigh<br>mature granulocytes hCD45+CD3-mNeonGreen-CD33/66b+19-14-10+SSChigh<br>monocytes hCD45+CD3-mNeonGreen-CD33/66b+19-14+SSClow<br>cDC hCD45+CD3-mNeonGreen-CD33/66b+19-14-11c+SSClow<br>pDC hCD45+CD3-mNeonGreen-CD33/66b+19-14-11c-123high SSClow<br>myeloblasts hCD45+CD3-mNeonGreen-CD33/66b+19-14-11c-123-34-SSClow<br>CD19+ lymphoid cells hCD45+CD3-mNeonGreen-CD19+CD33/66b-<br>pro-B cells hCD45+CD3-mNeonGreen-CD19+CD33/66b-10+34+<br>pre-B cells hCD45+CD3-mNeonGreen-CD19+CD33/66b-10+34-<br>mature B cells hCD45+CD3-mNeonGreen-CD19+CD33/66b-10-34-<br>NK cells hCD45+CD3-mNeonGreen-CD33/66b-19-56+<br>B-prolymphocytes hCD45+CD3-mNeonGreen-CD33/66b-19-56-34-10+<br>T-prolymphocytes hCD45+CD3-mNeonGreen-CD33/66b-19-56-34-7+<br>lineage-CD34+ hCD45+CD3-mNeonGreen-CD33/66b-19-56-34+ or CD33/66b+19-14-11c-34+SSClow<br>lineage-CD34+38+ lineage-CD34+38+<br>lineage-CD34+38- lineage-CD34+38-<br>pre-B/NK lineage-CD34+38+10+<br>GMP lineage-CD34+38+10-45RA+FLT3+<br>CMP lineage-CD34+38+10-45RA-FLT3+<br>MEP lineage-CD34+38+10-45RA-FLT3-<br>MLP lineage-CD34+38-10+45RA+<br>LMPP lineage-CD34+38-10-45RA+90-<br>MPP lineage-CD34+38-10-45RA-90- |

HSC lineage-CD34+38-10-45RA-90+

☒ Tick this box to confirm that a figure exemplifying the gating strategy is provided in the Supplementary Information.

