## [Peer Review File · Nature]

Manuscript Title: Epitope Editing Enables Targeted Immunotherapies for Acute Myeloid Leukemia

Editorial Notes: Redactions – unpublished data

Reviewer Comments & Author Rebuttals

Reviewer Reports on the Initial Version:

Referees' comments:

Referee #1 (Remarks to the Author):

Casirati et al. describe an approach to edit the epitope of hematopoietic stem and progenitor cells (HSPCs) using base-editors to enable selective resistance to targeted immunotherapy against FLT3, CD123 and KIT in acute myeloid leukemia (AML). This approach would allow targeting of these receptors in AML cells, while shielding and preserving the function of HSPCs. First, the authors use targeted approaches to change the epitope of FLT3, CD123 and KIT in order to identify variants that preserve the binding of the ligand, while diminishing the binding of therapeutic antibodies. The authors elegantly showed that this can be done using base editors. Specifically, the authors showed that these epitopes retained functionality in terms of ligand binding and ligand-induced proliferation. Further, they showed that these variants when introduced into leukemic K562 cells are resistant to CAR-T cells in vitro. Consequently, the authors were able to introduce these variants with high efficiency into HSPCs and were able to observe ligand-dependent proliferation in vitro and long-term engraftment in primary and secondary xenotransplantations with multilineage differentiation. In a very thorough in vivo experiment with co-culture of an AML patient xenograft and CAR-T cells, the authors were able to show that base edited FLT3 HSPCs were resistant, while control edited HPSCs showed lower percentages of normal myeloid and lymphoid progenitor subpopulations. Next, the authors were able to obtain similar results for base-edited CD123 HSPCs. Finally, the authors carried out multiplex editing with combined base-edited FLT3 and CD123 HSPCs in a proof-of-concept experiment showing that dual targeted immunotherapy could have synergistic benefits. Overall, the manuscript is well written as this can be a difficult task since the authors are describing three different targets. The number of in vivo experiments involving human normal and leukemic primary cells are impressive and the quality and stringency of the experimental approaches are very high. However, there are several major and minor comments that need to be addressed.

Major comments:

- Extended Data Figure 2a-d: Can the authors show similar activation of downstream signaling upon ligand binding in a more global approach, for example through mass spectrometry which could include multiple downstream pathways and phosphorylation states.
- Off target effects: It has been reported that there are off target effects of base editors with certain development contexts (PMID: 30819928, 30995674 and 31181567). Although the authors mention possible off target effects in the discussion, can the authors check the extent of these off-target effects experimentally?
- Figure 3c: The authors say that the base-editing efficiencies were equal between stem and progenitor subpopulations, which is remarkable, compared to previous HDR-directed approaches. Can the authors show the experiment with CD90-CD45RA- and CD90-CD45RA+. I am not sure why

the authors chose to include CD45RA+ cells as these are multi-lymphoid progenitors and are represented as a large percentage of the overall population (~40-50%). Finally, have the authors thought of including the marker CD49f (CD90-CD45RA+CD49f) as these represent long-term hematopoietic stem cells (PMID: 21737740). Similarly, can the authors show multiplex editing efficiencies for FLT3 and CD123 in Figure 6a for these subpopulations.

- Figure 3d: Can the authors show that there is no skewing of the lineage hierarchy in vitro when CD34+ bulk HSPCs are used and cultured, for example showing flow cytometry plots at day 0 before electroporation and then at the end of the experiment. I was under the assumption that there is considerable shift of these stem cells markers during in vitro culture. Additionally, can the authors isolate CD90-CD45RA+ HSCs at the end of the experiment and assess editing efficiency in that subpopulation, rather than just determining bulk editing efficiency of the whole population at the end of the experiment.

- Figure 3g: What are the results with KIT directed CAR-T cells? The authors mention that they would like to circumvent on target toxicities by using mAb instead of CAR-T cells, but why is that valid for KIT only and not FLT3 and CD123?

- Figure 3i - k: The authors nicely show primary and secondary xenotransplantations of FLT3 base-edited HSPCs. In the secondary xenotransplantations, the variation in editing efficiencies between different mice became much greater than in primary xenotransplantations. Can the authors speculate on this result? Also, as there is evidence that conventional knock-out of FLT3 does not have an effect in human HSCs upon transplantation, could the authors also incorporate secondary xenotransplantation results of KIT and CD123 variants?

- Figure 4m and p: The authors beautifully showed reduced percentages of myeloid and lymphoid progenitors, but less of an effect in HSCs upon CAR-T treatment in vivo. If they were to secondarily xenotransplant the HSPCs, would they be able to obtain comparable engraftments from base edited FLT3 vs. control cells, meaning there might be or might not be an advantage in the long-term repopulation upon base editing of FLT3 and short-term CAR-T exposure. It would be great to see that functional effect in long-term HSCs. Similar question to Figure 5 for CD123.

- There are no in vivo co-culture experiments with base-edited KIT. If the authors have data that would show that this approach is not feasible (negative data), it would be great to show these as well.

Minor comments:

- Line 52: The authors of this manuscript engineer a truncated version of CD33 (PMID: 30291334) in order to eliminate CD33-targeted immunotherapy on HSPCs. The authors should include this publication.

- Line 56: The statement that long-term studies on the functionality of CD33-KO myeloid cells in humans are lacking is correct, but could be a bit misleading. Experiments in humans have not been carried out, but previous studies in rhesus macaques showed that CD33 KO HSPC transplantation have long-term multilineage engraftment (PMID: 29856956).

- Figure 1d: Can the authors clarify the complexity of the combinatorial library. In total, how many different sequences were screened for FLT3?

- Extended Figure 2b: The western blot for pKIT and KIT look similar. Just to clarify, was the same membrane used for anti-pKIT and KIT detection and this is the reason why the blots are the same?

- Extended Data Figure 4f: As the authors have to electroporate HSPCs twice, what is the overall effect on cell numbers pre- and post-electroporation?

Referee #2 (Remarks to the Author):

Relapse is a common scenario in AML patients. Immunotherapies with CAR T or antibody for AML are still elusive, in part owing to the absence of AML-specific surface antigens, making AML difficult to be targeted. Since AML shares most of its surface markers with normal HSPCs or differentiated myeloid cells, immunotherapies would result in treatment-related myeloid aplasia and impairment of hematopoietic reconstitution. This article used the gene-edited hematopoietic stem cell transplant in order to allow safe and sustain the function of anti-AML immunotherapies, and retain long-term engraftment and multilineage differentiation capacity of HSPCs. There are some problems that need to be addressed.

Major

1.The article proves that epitope editing did not affect the engraftment and differentiation of HSPCs, whether this epitope editing has impact on the function of differentiated mature cells needs to be determined by cell function assays.

2.How about the gene expression profile of the differentiated untreated, AAVS1, FLT3 BE, CD123 BE and KIT BE HPSC cells? The RNA seq of these cells need to be added, which can assist in proving that epitope editing has no effect on HPSC cells.

3.The mice models with AML PDX cells were suggested to be measured by in vivo imaging to observe the AML burden in mice directly. If the fluorescence in vivo imaging of mNeonGreen is not well, the luciferase can be used to transduced to PDX for bioluminescent imaging.

4.Please add the result of in vitro Fab79D CAR killing assay on KITH378 epitope edited HSPCs.

5.Figure 3i-k, please add an untreated group as a blank control, which is helpful to prove this gene editing technology does not affect the stemness and differentiation of HSPCs in vivo.

6.Figure 3j, please add the flow cytometry analysis results of human engraftment (hCD45+ cells) in the secondary transplanted mice.

7.Line 321 and in Extended Data Fig. 6f and g, the results showed the phenotype of CAR T cells had no difference at the end of the experiment, except the PD-1 expression. Please add the results of the activation and degranulation of CAR T cells including CD69 and CD107a.

8.Figure 6, dual-specific FLT3/CD123 CAR-T cells were used in in vitro experiments, while 1:1 pool of 4G8 and CSL362 CAR-T cells were used in in vivo experiments, why? How about the anti-AML effect of dual-specific FLT3/CD123 CAR-T cells in vivo?

9.Extended Data Fig.7, the proportion of AML within BM CD45+ were more than 80% in b, while the proportion of AML within BM CD45+ were less than 20% in d, please explain. How about combination of 4G8, CSL362 and 79D CAR T cells?

10. FLT3 (CD135) was expressed more than 80% in both PDX-1 and PDX-2 cells, but the anti-leukemia effect of 4G8 was less effective in killing PDX-2 than PDX-1, please explain.

11. The authors performed double editing of FLT3 and CD123, how about editing of FLT3, CD123 and KIT simultaneously on K562 or HSPCs? Will the effect be better?

12. Why choose FLT3, CD123 and KIT? CD33 and CD38 were expressed higher than these markers expressed in PDX cells showed in Extended Data Fig. 5g.

Minor

1. Figure 1c, the western blot image of FLT3 was cropped. Please change with an original image of western blot.

2. Figure 1, whether the compensation of some flow cytometry plots was adjusted well, please confirm.

3. Line 231, "After optimization of mRNA in vitro transcription, culture and electroporation conditions and editing time point after HSPCs stimulation (Extended Data Fig. 4D,E,F), we achieved up to 86.6%, 78.6% and 67.9% of target A>G conversion for FLT3, KIT and CD123, respectively. We were able to efficiently edit the target adenines within the windows of FLT3-18, KIT-Y and CD123-R sgRNAs (Fig. 3B)". Which figure is corresponding to "86.6%, 78.6% and 67.9%"?

4. For in vitro and in vivo assays, please indicate whether the HSPCs were treated with autologous CAR T cells.

5. Figure 4e, please align the "+ -" with the figure.

6. Figure 4b, 5b and 6g, why was the proportion of FLT3 base editing different in the CD33+ and CD19+ BM cell from mice treated and untreated with CAR-T cells? Moreover, the proportion of CD123 editing had no difference. Please explain.

7. Figure 6f, why was the proportion of AML within hCD45+ cells higher in FLT3-CD123 than in AAVS1? Please explain.

Referee #3 (Remarks to the Author):

manuscript number 2022-09-14750

"Epitope Editing of Hematopoietic Stem Cells Enables Adoptive Immunotherapies for Acute Myeloid Leukemia"

The manuscript by Casirati and colleagues uses base editing techniques to engineer HSPCs for bone marrow transplantation. The engineering allows hematopoietic lineages with selective resistance to CAR-T or monoclonal antibody (mAb) therapy without affecting HSPC function. This is very important in the AML field because the targets for CAR-T or monoclonal antibody (mAb) therapy are also expressed on HSPCs, and thereby, these therapies usually have on-target/off-tumor toxicity. Compared to the work in the literature regarding gene editing to knock out the entire stem cell marker, the point mutation of the current work to edit the genome of stem cells is not conceptually novel. The authors' data on stem cell resistance to CAR T cell treatment do not match their hypothesis (see detailed comments below). The long-term safety issue for changing stem cells is

unclear. The authors also need to discuss in detail how to practice this in future clinical applications.

Below are specific comments of this reviewer:

1. It is unclear whether the reported gene editing has a safety issue or not. For example, it may change the malignancy potential of HSPCs. This at least should be discussed.
2. As the authors mentioned, several studies of gene-based editing of HSPCs targeting cancer and other diseases have been reported, including PMC7869435 and PMC4709030. Some similar studies are already tested in the early phase clinical stage (e.g., NCT04849910). This makes the current study incremental but not conceptually novel in the field.
3. Figure 4 (M and P): The main issue is the number of HSCs in the FLT3 BE/ + 4G8 group was obviously decreased compared to the no-CAR group, suggesting engineered FLT3 BE cannot completely resist CAR T therapy or the survival of edited HSPCs was decreased. Importantly, there is no significant difference between the editing group and the non-editing control group in the presence of CAR T cells. The same issue exists in the CD123 BE + CSL362 group vs. the no-CAR group in Figure 5, and in the FLT3/CD123 BE + 4G8/CSL362 group vs. the no-CAR group in Figure 6. This does not match the author's hypothesis.
4. Figure 2 (C-E): Is there a change in the production of cytokines by CAR T cells cocultured with HSPCs? The statistical data showed three groups (at the right), which is not consistent with the two groups of representative flow data (at the left). The testing groups for all p value calculations are not clear.
5. Figure 3 (J): The absolute cell numbers of each cell type should be provided. Data for various tissues such as blood and liver should be added.
6. The authors only showed one signaling pathway was not impaired in the edited HSPCs. This is not sufficient for the authors to draw the conclusion that the FLT3, KIT, and CD123 variants expressed in HSPCs preserved their signaling. The analysis of more signaling pathways downstream of FLT3, KIT, and CD123 should be provided.
7. It is unclear how this will be practiced in the clinic in the future. For example, is this mainly for preventing relapse post-transplantation? The cost and the time to engineer stem cells to meet clinical applications also need to be considered.
8. For CAR-T cell generation in Figure 2, the authors used different MOIs to infect T cells, but the infection ratio and expansion were very similar among different MOIs. The results should be confirmed.
9. The potential gRNA off-targeting in stem cells may lead to a safety issue.
10. The authors claim "with no differences within the myeloid and lymphoid lineages (FACS-sorted CD33+ and CD19+ cells, respectively; Fig.4B)". However, the p-values show a difference. Also, the p values in the CD33+ group and the p value in the CD19+ group are identical (0.028582). This seems to be unlikely.
11. Fig. 5p and q, it looks like the count of HSCs in CD132BE and AAC1 BE groups are significantly lower than no CAR group (5.3-fold difference), but the p-value is not significant (P=0.21). Also, these numbers are identical in Figure 5q.

Minor comments:

1. The extended data J and K are missing.
2. More detailed information on the statistical analyses should be provided.

Point-by-point reply to the Editor and Reviewers

Referees' comments:

Referee #1 (Remarks to the Author):

Casirati et al. describe an approach to edit the epitope of hematopoietic stem and progenitor cells (HSCPs) using base-editors to enable selective resistance to targeted immunotherapy against FLT3, CD123 and KIT in acute myeloid leukemia (AML). This approach would allow targeting of these receptors in AML cells, while shielding and preserving the function of HSPCs. First, the authors use targeted approaches to change the epitope of FLT3, CD123 and KIT in order to identify variants that preserve the binding of the ligand, while diminishing the binding of therapeutic antibodies. The authors elegantly showed that this can be done using base editors. Specifically, the authors showed that these epitopes retained functionality in terms of ligand binding and ligand-induced proliferation. Further, they showed that these variants when introduced into leukemic K562 cells are resistant to CAR-T cells in vitro. Consequently, the authors were able to introduce these variants with high efficiency into HSPCs and were able to observe ligand-dependent proliferation in vitro and long-term engraftment in primary and secondary xenotransplantations with multilineage differentiation. In a very thorough in vivo experiment with co-culture of an AML patient xenograft and CAR-T cells, the authors were able to show that base edited FLT3 HSPCs were resistant, while control edited HPSCs showed lower percentages of normal myeloid and lymphoid progenitor subpopulations. Next, the authors were able to obtain similar results for base-edited CD123 HSPCs. Finally, the authors carried out multiplex editing with combined base-edited FLT3 and CD123 HSPCs in a proof-of-concept experiment showing that dual targeted immunotherapy could have synergistic benefits. Overall, the manuscript is well written as this can be a difficult task since the authors are describing three different targets. The number of in vivo experiments involving human normal and leukemic primary cells are impressive and the quality and stringency of the experimental approaches are very high. However, there are several major and minor comments that need to be addressed.

We thank the Reviewer for the positive comments on our work and the careful evaluation of our data and analyses.

Major comments:

- Extended Data Figure 2a-d: Can the authors show similar activation of downstream signaling upon ligand binding in a more global approach, for example through mass spectrometry which could include multiple downstream pathways and phosphorylation states.

We agree that the point raised by Referee #1 is of great importance in validating the functionality of epitope-edited cytokine/growth factor receptors and edited HSPCs. To this end, we have included a comprehensive and unbiased analysis of receptor activation and downstream signaling by (1) performing global RNAseq of CD34+ HSPCs edited for FLT3, CD123, KIT, AAVS1 or unedited (electroporation only), either unstimulated or stimulated with each respective ligand, in biological triplicate; (2) investigating the phospho-proteome profile of stimulated or unstimulated edited HSPCs by mass-spectrometry (MS), in biological duplicate. Additionally, we have

interrogated a wider panel of intracellular pathways by phospho-flow on *in vitro* differentiated myeloid cells stimulated with several mediators.

The RNAseq analyses identified 78, 2667 and 7944 differentially expressed genes (DEGs) associated with FLT3L, SCF and IL-3 stimulation, respectively (Extended Data FIG.5b,c). The number of measured DEGs correlates well with the overall impact of each cytokine on the heterogeneous CD34 stem/progenitor cells and Gene Set Enrichment Analysis (GSEA) confirmed that these genes are associated with the receptor-mediated proliferation pathways. By comparing FLT3, CD123, KIT edited conditions with the respective AAVS1 control conditions, we observed no significant transcriptional differences both at baseline and upon FLT3L, IL-3 and SCF stimulation (FIG.3h), thus confirming full preservation of the functionality and downstream effects of the epitope engineered receptors.

These results are reported in Fig.3 and Extended Data Fig.5 and described in text lines 269-276: *“In order to comprehensively evaluate any transcriptional changes associated with the epitope editing procedure and the response of FLT3, CD123 and KIT modified cells upon stimulation with their respective ligands, we performed RNAseq of CD34+ HSPCs edited for FLT3, CD123, KIT, AAVS1 or unedited cells (electroporation only), either stimulated or unstimulated with the respective ligand. We found 78, 2667 and 7944 differentially expressed genes (DEGs) associated with FLT3L, SCF and IL-3 stimulation, respectively (Extended Data Fig.5b,c). By comparing FLT3, CD123, KIT edited conditions with AAVS1 control, we confirmed the absence of transcriptional differences both at baseline and upon FLT3L, IL-3 and SCF stimulation (Fig.3h).”*

In order to complement the transcriptional analysis with a global proteomic assay, we established a collaboration with Dr. Jarrod A. Marto, director of the Blais Proteomics Center at the Dana-Farber Cancer Institute. Thanks to his expertise, we were able to perform a phospho-proteome profiling of a large number of edited HSPCs, either at baseline or stimulated with their respective receptor ligands. Despite the technical challenges of this complex analysis, caused by the low protein content obtained from primary HSPCs and the high signal-to-noise ratio of this methodology, which did not allow us to perform in-depth inferential analyses, we were able to observe concordant effects of the stimulated samples across editing conditions and AAVS1 controls.

These new data are now included in the new Extended Data FIG.5 and Supplementary Data Fig.3 and described in lines 275-279: *“Phospho-proteomic profiling by mass-spectrometry of FLT3, CD123, KIT or AAVS1 edited CD34+ HSPCs showed concordant changes of differentially phosphorylated sites upon ligand stimulation between receptor-edited and AAVS1 conditions (Extended Data Fig.5e), again confirming in an unbiased manner that activation of downstream signaling by epitope-modified receptors is preserved.”*

- Off target effects: It has been reported that there are off target effects of base editors with certain development contexts (PMID: 30819928, 30995674 and 31181567). Although the authors mention possible off target effects in the discussion, can the authors check the extent of these off-target effects experimentally?

Off-target effects (OT) constitute a critical aspect of any genome editing technique. While CRISPR-Adenine Base Editing (ABE) has demonstrated considerable advantages compared to other gene-editing technologies (eg. nuclease mediated homology-directed repair or even Cytidine Base Editors) due to the lack of DNA double-strand breaks, minimal non-gRNA-mediated genomic deamination, no risk of introducing STOP codons and the inherent safety associated with the redundancy of the genomic code, a comprehensive evaluation of potential genomic OT and their impact on cell functionality and fitness must be addressed before moving any stem cell-based therapeutic strategy toward clinical translation.

While such an extensive and in-depth investigation is beyond the scope of this work, we have now included a new full paragraph in which we provide promising pieces of evidence of the prospective safety of our epitope engineering approach. The potential gRNA-dependent off-target (OT) effects of the utilized base editors were evaluated by combining i) a genome-wide, unbiased off-target identification assay (GUIDE-seq), performed in collaboration with Dr. Daniel Bauer (Boston Children's Hospital), and ii) an in silico prediction analysis. To adapt the GUIDE-seq assay to our SpRY-based enzyme, we generated a SpRY-nuclease mRNA and used it to perform the dsDNA oligo-bait trapping in a permissive cell line (293T), to possibly increase the sensitivity of the screening. Despite this, the analysis led to the identification of few potential OT sites for all the tested gRNAs (N = 12 intronic and 9 intergenic for FLT3-18, N = 1 intergenic for KIT-Y and N = 1 intergenic for CD123-R, Supplementary Data FIG.4b and Table 16). Since none of the identified OTs fell into a coding or canonical splicing sequence, even if unspecific deaminations occur in these regions they will likely have no functional consequences. We then decided to extend this analysis and characterize the top off-target genomic loci for FLT3, CD123, and KIT sgRNAs predicted by the CRISPOR tool on the basis of sequence homologies with the gRNAs. For these sites, we assess the actual OT deamination occurring on CD34+ HSPCs treated for base editing by performing targeted next-generation sequencing. This analysis showed no significant off-target deamination over background for any of the analyzed CD123 or KIT sites (Supplementary Data Figure 4c). However, for the FLT3-18 sgRNA, we observed comparatively higher deamination in four loci: three intronic (ERN1, NFX1, RP11-242J7) and one exonic (SNTG1). While the unspecific base editing of these intronic sites (distant from canonical splicing sequence) likely has no functional consequences, the low but detectable edit (~5%) of the SNTG1 exon might possibly result in amino acid substitution of the affected protein. Luckily, however, the protein encoded by the *SNTG1* gene (Syntrophin γ 1) is a brain-specific protein of the syntrophin family that is not expressed and does not have any known functional role in hematopoietic tissue¹⁻³. By reanalyzing our RNAseq data, we also confirmed that the SNTG1 transcript is not detectable in CD34+ HSPCs (Supplementary Data Table 22).

Overall, these analyses suggest a promising specificity profile of our selected gRNAs and base editor enzyme. Moreover, while we used a PAM-relaxed Cas variant to facilitate the development of our epitope engineering for this proof-of-concept work, after identification of the desired base change for epitope engineering it will be possible to test several Cas variants with specific PAM restrictions that better fit our target genes and concomitantly provide an overall higher specificity profile for a prospective clinical application. As a representative example, we tested an alternative gRNA for FLT3 (FLT3-16) with an AGA PAM (binding 2-nt upstream of FLT3-18) in combination with a more restrictive Cas variant (SpG). Targeted deep sequencing of the top intronic and exonic FLT3-16 predicted OT sites showed significant deamination over background while this base

editor preserved ~90% of the on-target activity compared to the FLT3-18 SpRY (70.6% vs 81% median editing in CD34+ HSPCs, respectively; Supplementary Data FigG.4c and Table 20).

Since base editing enzymes can also have non-gRNA-dependent OT effects, we interrogated our new RNAseq dataset generated on CD34+ HSPCs (from Fig.3h and Extended Data Fig.5) to assess the occurrence of major OT activity on some of the expressed genes. Despite the limitations of this analysis, which had to be confined to the transcripts with a high sequencing coverage (top 5%) in order to guarantee proper statistical power, we observed a promising profile with no significant A>G conversions in edited samples compared to controls (Supplementary Data Fig.5b). This analysis confirmed that any OT activity on RNA, if occurring, is highly transient and not detectable after 4 days from base editing treatment (the time at which HSPCs were analyzed for RNAseq). Moreover, it also provided a low sensitivity but unbiased assessment of unspecific genomic deamination of genes that are highly expressed in the edited cells, which are more prone to unspecific editing due to the presence of ssDNA denatured by the passage of the transcriptional machinery.

Finally, we also evaluated the occurrence of undesired on-target mutations by performing deep sequencing of the edited genes. This analysis showed a low rate of indel formation, which was on average 0.66% for all target loci (0.6, 1.2 and 0.2% for FLT3, CD123 and KIT, respectively; Supplementary Fig. 5c), in line with previously reported results utilizing adenine base editors^{4,5}. As highlighted in the discussion, while we cannot formally exclude that base editing could induce additional unpredicted and unwanted on-target mutations, the region affected by epitope editing (extracellular domain 4 for FLT3 and KIT and the N-terminal domain for CD123) is distant from mutational hotspots involved in cancer-associated variants (ie. tyrosine kinase domain for KIT-D816 or FLT3-D835 or the FLT3 juxta-membrane auto-inhibitory domain involved in FLT3 internal tandem duplications – see also Response to the Referees - Figure 5). Thus, the main concern for these indels would be a possible loss of function in a fraction of the treated cells, which will be spontaneously counter-selected due to the reduced fitness of the resulting cells within the stem cell pool.

These results are reported in Supplementary Data Fig.4 and 5 and described in the new paragraph “Off-target effects of epitope editing” in lines 315-342: *“Since the use of SpRY-Cas9 might lead to potential gRNA-dependent off-target (OT) effects, we performed a specificity analysis by combining genome-wide, unbiased identification of OT sites (GUIDE-Seq) and in silico OT prediction analysis (CRISPOR). To identify potential OTs of our selected FLT3, KIT and CD123 gRNAs, we first performed a GUIDE-seq screening using the SpRY-nuclease (Supplementary Data Fig.4a). By mapping the identified OT sites with mismatches or bulge ≤ 6 , we found that all of them were located in non-coding genomic regions (12 intronic and 11 intergenic, Supplementary Data FIG.4b and Table 16). Thus, we characterized the top exonic and intronic in silico predicted OT sites for FLT3 (N=12), CD123 (N=9), KIT (N=12) sgRNAs and assessed the levels of undesired deamination on BE CD34+ HSPCs by targeted deep sequencing (Supplementary data FIG.5a). In this analysis, no significant off-target deamination over background was observed for any of the analyzed CD123 or KIT sites (Supplementary Data FIG.4c), while 4 loci showed comparatively higher deamination for the FLT3-18 sgRNA. Among these FLT3 OTs, only one was located in an exonic sequence but the affected gene, SNTG1 (Syntrophin γ 1), is a brain-specific syntrophin*

family protein with no expression (Supplementary Data Table 22) nor known functional role in hematopoietic tissue¹⁻³. Despite this generally safe profile, we found that the use of an alternative gRNA (binding 2-nt upstream of FLT3-18 with an AGA PAM) in combination with a more restricted Cas variant (SpG) allows avoiding OT deamination at the predicted OT sites while preserving ~90% of on-target activity compared to the FLT3-18 sgRNA (70.6% vs 81% median editing in CD34+ HSPCs, respectively; Supplementary Data FIG.4c and Table 20). To assess the occurrence of major non-gRNA-dependent deaminations, we interrogated our RNAseq dataset generated on CD34+ HSPCs (from Fig.3h and Extended Data Fig.5) and observed no significant A>G conversions on transcripts with high sequencing coverage (top 5%) in edited samples compared to controls (Supplementary Data FIG.5b). Finally, we evaluated the rate of on-target indels, which were below 1.5% for all target loci (0.6, 1.2 and 0.2% for FLT3, CD123 and KIT, respectively; Supplementary FIG. 5c), in line with previously reported data for ABE^{4,5}. Overall, these data support a generally safe genotoxicity profile of FLT3, CD123 and KIT epitope editing in CD34+ HSPCs.”

While overall these encouraging results suggest a promising safety profile of our selected base editing enzymes, we continue to highlight in the discussion the importance of performing a more comprehensive characterization of the specificity of gRNA and non-gRNA dependent off-target activity as well as a careful evaluation of risk/benefit ratio before clinical implementation.

- Figure 3c: The authors say that the base-editing efficiencies were equal between stem and progenitor subpopulations, which is remarkable, compared to previous HDR-directed approaches. Can the authors show the experiment with CD90-CD45RA- and CD90-CD45RA+. I am not sure why the authors chose to include CD45RA+ cells as these are multi-lymphoid progenitors and are represented as a large percentage of the overall population (~40-50%). Finally, have the authors thought of including the marker CD49f (CD90-CD45RA+CD49f) as these represent long-term hematopoietic stem cells (PMID: 21737740). Similarly, can the authors show multiplex editing efficiencies for FLT3 and CD123 in Figure 6a for these subpopulations.

Our previous analysis showed that the more primitive HSPC population (CD90+CD45RA-) has similar editing efficiencies compared to the more differentiated CD90- fractions (CD45RA positive and negative). However, we agree with the Reviewer's suggestion to exploit a more stringent combination of markers, including CD49f, to identify hematopoietic stem cells with long-term repopulating capacity from primary samples (lin-CD34+38-10-90+45RA-49f+)⁶. While there have been reports showing that expression of CD49f alone does not identify cells with enhanced repopulating potential during *in vitro* culture of CD34+ HSPCs⁷, stringent gating of CD34+133+45RA-90+49f^{mid} instead of just CD49f^{high} allows isolation of cells overlapping with the LT-HSCs-enriched subpopulation (Fares et al., Supplementary Figure 4⁷). To stringently assess *in vitro* editing efficiencies in subsets with different repopulating potentials, we performed editing experiments on mPB CD34+ HSPCs and sorted CD90-45RA-, CD90-45RA+, CD90+133+45RA-49f-, CD90+133+45RA-49f^{mid} cells and compared the editing efficiencies of the bulk population with sorted fractions (biological triplicate, new Fig.3c). We observed similar editing % across all subsets, with only minor differences in the case of KIT (for which CD45RA-90- cells displayed relatively higher editing efficiency) indicating that BE is relatively devoid of cell type or cell cycle bias as in the case of HDR. We are now also reporting the multiplex editing efficiencies performed

in the same sorted populations of FLT3 and CD123 dual-edited HSPCs, which again showed no difference across the analyzed subpopulations (new Fig.6a).

These results are reported in the new Fig.3a and Fig.6a and described in lines 239-241: “*Contrary to what we previously observed with HDR-mediated editing⁸, base editing efficiencies were similar in bulk cells and in more primitive, HSC-enriched subsets (CD34+133+45RA-90+49f- and CD34+133+45RA-90+49f^{mid}; Fig.3b).*”⁸

- Figure 3d: Can the authors show that there is no skewing of the lineage hierarchy in vitro when CD34+ bulk HSPCs are used and cultured, for example showing flow cytometry plots at day 0 before electroporation and then at the end of the experiment. I was under the assumption that there is considerable shift of these stem cell markers during in vitro culture. Additionally, can the authors isolate CD90-CD45RA+ HSCs at the end of the experiment and assess editing efficiency in that subpopulation, rather than just determining bulk editing efficiency of the whole population at the end of the experiment.

We apologize to the Reviewer for the lack of clarity. Skewing of the surface immune-phenotype and progressive loss of repopulating potential are inevitable consequences of *in vitro* hematopoietic stem cell culture. Despite the use of stem-preserving (SR-1, UM171) compounds, in our experiments we consistently observed considerable shifts of stem cell markers during culture. In our *in vitro* analyses, we aimed to compare the culture composition of FLT3, CD123, and KIT epitope-edited HSPCs vs controls (*AAVS1* safe harbor edited or electroporation-only, UT) at the same time point after initial stimulation. By showing the absence of phenotypic differences between epitope-edited and control HSPCs, we conclude that our edited receptors do not accelerate or delay HSPCs differentiation, thus supporting the notion that target epitope engineering has a negligible impact on HSPCs functionality. We now better clarify this point in the text (lines 243-244).

Regarding the second point, as described above, Fig.3c and Fig.6a have been updated with a more granular sorting strategy to assess editing efficiencies in stem-enriched and stem-depleted subsets at the end of the *in vitro* culture, including CD45RA-90-, CD45RA+90-, CD90+49f- and CD90+49f^{mid} subsets.

- Figure 3g: What are the results with KIT directed CAR-T cells? The authors mention that they would like to circumvent on target toxicities by using mAb instead of CAR-T cells, but why is that valid for KIT only and not FLT3 and CD123?

Contrary to FLT3 and CD123, which are expressed almost exclusively in hematopoietic cells, the extra-hematopoietic expression of KIT raises serious concerns about the safety of prolonged or highly aggressive KIT-directed adoptive immunotherapies. Sadly, this issue has been recently highlighted by the occurrence of severe adverse events in an early-phase clinical trial employing a KIT ADC for the treatment of patients affected by AML and myelodysplastic syndrome⁹. In this trial, after two cases of dose-limiting toxicities — both Grade 4 respiratory serious adverse events — in the cohort receiving the highest dose (0.13 mg/kg), one of the patients treated with a lower dose (0.08 mg/kg) experienced a Grade 5 Serious Adverse Event (SAE) (respiratory failure and cardiac arrest resulting in death) deemed to be related to the anti-KIT ADC. While the actual

causative mechanism of this toxicity has still to be identified, the extra-hematopoietic KIT expression has been listed among the possible causes, along with the toxicity of the conjugated toxin and the contribution of the treated hematopoietic malignancy. For this reason, we thought that exploiting the dimerization-blocking activity and antibody-mediated cellular cytotoxicity (ADCC) of the naked Fab79D mAb would provide a less stringent, more controllable, and therefore safer way to deplete AML cells and/or healthy HPSCs compared to a cellular therapy with CAR T cells (Fig.3g). An ongoing clinical trial using a similar ligand blocking, naked mAb against KIT (Briquilimab) has been indeed shown to be well tolerated in all the treated patients (Jasper Therapeutics).

In a parallel work, we are currently exploring the KIT BE strategy to achieve (1) *in vivo* selection of genome-modified HSPCs and (2) enable safer immune-based non-genotoxic conditioning for chemotherapy-free replacement of the human hematopoiesis. Whereas these experiments will be described in a future follow-up manuscript, we have already obtained preliminary evidence of the efficacy of Fab79D mAb in co-selecting epitope-edited HSPCs *in vivo* (Response to Referee FIG.1), which further supports our decision of prioritizing mAb for this target.

Redacted

Regarding the KIT-directed CAR, we have exploited KIT CAR-T cells as a tool for obtaining stringent *in vitro* validation of the role of H378R epitope editing to protect KIT-overexpressing K562 cells (Fig.2d). As previous work by Myburgh et al. (PMID: 32358567), KIT CAR-T cells generated with a similar version of the same Fab79D domain have already been shown to effectively deplete both healthy and malignant human hematopoietic cells both *in vitro* and in xeno-transplanted mice. While we have not extensively optimized our Fab79D CAR construct, we are now providing new experimental data supporting the protection of KIT^{H378R} BE HSPCs co-cultured with KIT CAR-T cells. As HSPCs cultured *in vitro* with SCF express very low surface levels of KIT, the on-target killing observed in our experiments is not on par with what was observed with FLT3 or CD123 CAR-T cells. Nonetheless KIT BE CD34⁺ HSPCs are relatively preserved when compared with *AAVS1* control at high effector:target ratios (Response to the Referee FIG.2). However, since these results have been obtained with an underdeveloped version of the Fab79D CAR, we have reported these results in this point-by-point reply (which, in case of publication of this manuscript, will be available as online material), but we have not currently described them in the main text.

[Response to Referees] Figure 2. *KIT^{H378R} BE HSPCs are relatively resistant to Fab79D CAR-T cells in vitro. KIT epitope-edited HSPCs were co-cultured with Fab79D CAR-T cells or untransduced T cells (UT) at different E:T ratios. Outcome was evaluated at D7 by flow cytometry. Normalized absolute counts are reported in the plots.*

- Figure 3i – k: The authors nicely show primary and secondary xenotransplantations of FLT3 base-edited HSPCs. In the secondary xenotransplantations, the variation in editing efficiencies between different mice became much greater than in primary xenotransplantations. Can the authors speculate on this result? Also, as there is evidence that conventional knock-out of FLT3 does not have an effect in human HSCs upon transplantation, could the authors also incorporate secondary xenotransplantation results of KIT and CD123 variants?

We thank the Reviewer for spotting this very interesting aspect of this experimental hematology model. In all our previous xenotransplantation studies performed with HDR-edited human HSPCs, we always noticed a greater variation in the percentage of edited cells when moving from primary to secondary transplants. It appears that the lower the percentage of edited cells in primary transplant, the higher the variation in secondary recipients (e.g., compare results from PMID: 24870228 vs PMID: 32601433). In a recent work, we exploited a barcoding approach (BAR-seq)¹⁰ to perform clonal tracking of edited cells in both primary and secondary transplants. In this study, we found that the number of dominant edited clones contributing to hematopoietic output in primary transplants is relatively small (~10), because of the well-known limitation of the NSG xenogeneic model, and it contracts further in secondary recipients (PMID: 32601433). These results indicate a ‘bottleneck’ effect during the engraftment of human HSPCs in secondary recipients, which randomly distributes the few edited HSCs in different mice and thus results in higher variability in their percentage contribution with respect to the unedited HSCs.

In the experiment reported in FIG3i-k, HSPC base editing efficiency was relatively low (~40%). Moreover, the bone marrow cells isolated from primary recipients underwent a freeze-and-thaw cycle before injection in secondary recipients, which likely further reduced the overall clonality of the graft and increased the “bottleneck” effect, which in turn resulted in the observed higher variation in the fraction of edited cells.

We also agree with the Reviewer that serial transplantation is one important assay to assess long-term repopulation capacity and we strive to provide evidence of secondary engraftment for all our in vivo experiments. Since starting new primary and secondary transplant experiments would require several months of work, which could jeopardize the timely re-submission of this manuscript focused on a highly innovative but very competitive topic, we used as cell source the frozen bone marrow of the primary mice described in the original Extended Data Fig.5 and Fig.6. Cells from these samples were thawed and, in the case of CD123, human HSPCs were FACS-

sorted to remove AML PDX cells. Despite the low amount of starting material, these experiments confirmed the long-term engraftment capacity and persistence of both KIT and CD123 edited HSPCs without counter-selection effects.

These experiments are now reported in *Extended Data Fig. 7e-j* and described in lines 307-312: *”Similar results were also observed for secondary transplantation of BM KIT^{H378} cells, which generated grafts with comparable composition as AAVS1^{BE} controls (Extended Data Fig. 7e-f), with no change in editing levels, both measured by molecular or flow cytometry analyses (Extended Data Fig. 7g,h). As done for FLT3 and KIT, we confirmed the engraftment and the stability of CD123^{S59} BE levels in both primary and secondary TXs (Extended Data Fig. 7i,j).”*

Finally, we would like to respectfully highlight that, despite Flt3 knock-out mice are viable, there are compelling evidences that Flt3 KO in murine hematopoiesis has significant effects on repopulating capacity and lineage differentiation, including reduced HSC and defects in B cell and myeloid development¹¹⁻¹⁵. While we have not performed xenotransplantation of human FLT3-KO HSPCs, we assessed the expansion and colony-forming capacity of FLT3-KO CD34+ cells *in vitro* and compared it to FLT3 epitope editing (Response to the Referee Fig.3). These data, coupled with the observation that murine Flt3l is cross-reactive with human FLT3, further corroborate the notion that the epitope editing of FLT3 is not affecting the fitness and functionality of treated cells.

[Response to Referees] Figure 3. CRISPR-Cas9 mediated knock-out of FLT3 results in reduced CD34+ expansion and colony-forming potential. Left, representative photomicrograph of CFU assays plated with CD34+ HSPCs, either FLT3 epitope edited or knocked out by CRISPR-Cas9 nuclease. CFUs were imaged 14 days after plating. Top Right, Absolute counts of total CD34+ HSPCs and CD90+45RA- stem enriched subset in untreated (UT), AAVS1 BE, FLT3 BE and FLT3 KO during *in vitro* expansion culture. Bottom Right, CFU counts for the same conditions at 14 days.

- Figure 4m and p: The authors beautifully showed reduced percentages of myeloid and lymphoid progenitors, but less of an effect in HSCs upon CAR-T treatment *in vivo*. If they were to secondarily xenotransplant the HSPCs, would they be able to obtain comparable engraftments from base edited FLT3 vs. control cells, meaning there might be or might not be an advantage in the long-term repopulation upon base editing of FLT3 and short-term CAR-T exposure. It would be great to see that functional effect in long-term HSCs. Similar question to Figure 5 for CD123.

We thank the Reviewer for the appreciation of our results and for kindly suggesting a possible strategy to further highlight the protective role of epitope editing in HSC. While FLT3 and CD123 are expressed on early progenitors, as reported in the flow analysis of human grafts (Extended

Data Fig.8e and 9a) their expression on phenotypically defined HSCs (lineage-CD34+38-10-45RA-90+) *in vivo* is low and CAR-mediated killing of HSCs may play less a significant role compared to other progenitor subsets. Nonetheless, our data showed some degree of protection of epitope-edited HSCs that, albeit not reaching statistical significance, was consistent across all the described *in vivo* CAR experiments (3.1x, 5.3x and 7.1x more HSCs in the edited group with FLT3, CD123 and combined FLT3+CD123 editing, respectively, FIG,4m, 5p, 6h). As described in detail in the reply to Reviewer #3, the on-target effects of CAR-T on HSC are also further confounded by an unspecific effect of transplanted T cells on HSC graft (new Extended Data Fig.7n). We have reorganized the plots in Fig.4, Fig.5 and Fig.6, by separating the progenitor populations and changing the scale axes, to better highlight the protective effect on edited HSC.

While we agree with the Reviewer's advice that secondary transplantation might in theory underline the protective effect of epitope editing on repopulating HSCs, the actual experiments are technically challenging because, as described in the reply to the previous point, they would require extensive cell manipulation to remove CAR-T and PDX and would require a large number of both CD34+ and animals to reach significant results (because of the aforementioned unspecific toxicity of T cells). As the main goal of the FLT3 and CD123 immunotherapies described in this manuscript is not the depletion of LT-HSC but the elimination of AML leukemia stem cells (LSCs) which display higher expression levels of our targets, we do not consider the lack of statistical differences within the HSC subset an issue of the epitope editing approach but a reflection of the underlying biology. While we concentrated on reducing the toxicity on the most severely affected hematopoietic lineages, the hypotheses and the experimental procedures suggested by the referee would lay the foundation for an exciting follow-up study more focused on the effects of targeted immunotherapies on HSCs, which may be used - for example - to develop a new biological conditioning strategy for HSCT.

- There are no *in vivo* co-culture experiments with base-edited KIT. If the authors have data that would show that this approach is not feasible (negative data), it would be great to show these as well.

As discussed above, we agree that the development of our anti-AML immunotherapy strategies targeting KIT is currently less advanced compared to FLT3 and CD123. Yet, while the potency of our 79D CAR-T on *in vitro* expanded CD34+ cells is currently not on par with our FLT3 and CD123 CARs, we were able to observe significant protection of KIT^{H378R} BE cells compared to *AAVS1* control (Response to Referee Fig.2). Moreover, we are currently evaluating the effects of KIT 79D mAb *in vivo* (see Response to Referee Fig.1) for a follow-up work. Thus, we believe that there is great value in reporting in this manuscript our current findings on KIT epitope engineering.

Minor comments:

- Line 52: The authors of this manuscript engineer a truncated version of CD33 (PMID: 30291334) in order to eliminate CD33-targeted immunotherapy on HSPCs. The authors should include this publication.

We thank the Reviewer for spotting this mistake. In our original manuscript, we were already mentioning the exon-skipping approach described by the authors of PMID: 30291334, but this

reference was erroneously missing. We have now updated the references list to amend this omission and include the suggested publication (line 57).¹⁶

- Line 56: The statement that long-term studies on the functionality of CD33-KO myeloid cells in humans are lacking is correct, but could be a bit misleading. Experiments in humans have not been carried out, but previous studies in rhesus macaques showed that CD33 KO HSPC transplantation have long-term multilineage engraftment (PMID: 29856956).

We apologize for the lack of clarity. With our statement, we were referring to function of the CD33 protein on myeloid cells. While previous studies in rhesus macaques showed very promising results on hematopoietic reconstitution and myeloid differentiation, it is not known if CD33 has any functional role in myeloid cell differentiation or long-term engraftment. Instead, CD33 appears to play a role in the regulation of the phagocytic function of myeloid cells¹⁷⁻²⁰ and some CD33 polymorphisms have been linked to Alzheimer disease risk²⁰⁻³⁰. We have now modified Line 61 to acknowledge the existing studies of CD33-KO in rhesus macaques and to better indicate the current lack of knowledge on the CD33 function in human cells.

- Figure 1d: Can the authors clarify the complexity of the combinatorial library. In total, how many different sequences were screened for FLT3?

The combinatorial library for FLT3 was designed to randomly contain the human or the murine codons at each of the 16 positions (N354, S356, D358, Q363, E366, Q378, T384, R387, K388, K395, D398, N399, N408, H411, Q412, H419) of the extracellular domain, and cloned to accommodate at least 10 times the theoretical library complexity of 65,536. We have now updated the Methods section to include a more comprehensive description of the FLT3 combinatorial library design and complexity.

“We designed a combinatorial plasmid library where the human or the murine codons were randomly selected at each of 16 positions (N354, S356, D358, Q363, E366, Q378, T384, R387, K388, K395, D398, N399, N408, H411, Q412, H419) within FLT3 ECD4 (GenScript). The library was cloned to allow a total complexity of 65,536 (covered at least 10 times in the library). Single colony sequencing, generated at GenScript, showed a rate of $N = 4$ mutations per variant of 68.75%, $N = 5$ mutations per variant = 18.75%. NGS sequencing of K562 cells transduced with the library, among 9166 filtered reads, detected 4375 unique amino acid sequences with a frequency > 0.0001 .”

- Extended Figure 2b: The western blot for pKIT and KIT look similar. Just to clarify, was the same membrane used for anti-pKIT and KIT detection and this is the reason why the blots are the same?

Yes, the same membrane was probed first for pKIT, then stripped by using Thermo Scientific Restore stripping buffer (cat.no. 21059) according to the manufacturer’s instructions, and then re-probed with anti-KIT antibody. Actin was probed on the same membrane, which was cut at the ~75 kDa mark to separate KIT (~130-135 kDa) and Actin (~42 kDa), before antibody incubation. Full description of western blot methods has been updated in the Methods section (lines 1464-

1470) and, accordingly to editorial policies, full-size uncropped pictures of the western blot membranes have been provided (Supplementary Data Fig.1 and 2).

- Extended Data Figure 4f: As the authors have to electroporate HSPCs twice, what is the overall effect on cell numbers pre- and post-electroporation?

We apologize for the lack of clarity. During the editing procedure, HSPCs undergo only one round of electroporation. The experiment reported in Extended Data Fig.4f was aimed at exploring the effects of electroporation at different time points after CD34+ cell thawing and identifying the best-performing time for maximizing efficiency while reducing cellular toxicity. Briefly, mobilized peripheral blood-derived CD34+ HSPCs were thawed and cultured in StemSpan SFEMII supplemented with FLT3L, SCF, TPO, SR-1, UM171 and divided into three separate wells. At 24, 48, or 72 hours after thawing, the cells in one of three wells were collected, further divided into 3 conditions and electroporated with SpRY-ABE8e mRNA and combinations of 2 sgRNAs (FLT3+CD123, FLT3+KIT and CD123+KIT). We found that the 48h timepoint provided the best compromise between editing efficiency, stem cell phenotype and absolute counts of cultured cells.

Referee #2 (Remarks to the Author):

Relapse is a common scenario in AML patients. Immunotherapies with CAR T or antibody for AML are still elusive, in part owing to the absence of AML-specific surface antigens, making AML difficult to be targeted. Since AML shares most of its surface markers with normal HSPCs or differentiated myeloid cells, immunotherapies would result in treatment-related myeloid aplasia and impairment of hematopoietic reconstitution. This article used the gene-edited hematopoietic stem cell transplant in order to allow safe and sustain the function of anti-AML immunotherapies, and retain long-term engraftment and multilineage differentiation capacity of HSPCs. There are some problems that need to be addressed.

We thank the Reviewer for acknowledging the significance and relevance of our work in the development of potential new treatments for AML patients.

Major

1.The article proves that epitope editing did not affect the engraftment and differentiation of HSPCs, whether this epitope editing has impact on the function of differentiated mature cells needs to be determined by cell function assays.

We thank the Reviewer for raising this question. While our molecular characterization of receptor functionality should attenuate any concern of unexpected downstream effects on mature hematopoietic cells, we acknowledge that a direct functional validation of differentiated cells would provide even stronger evidence of the full functionality of the receptor-edited hematopoiesis.

In addition to the extensive validation of epitope-edited FLT3, KIT and CD123 receptor functionality (including ligand affinity, phosphorylation, signal transduction, cellular responses to stimulation, in vivo SCID-repopulating and multilineage differentiation capacity), we have now

generated new data to support the functionality of differentiated mature lineages derived from CD34+ HSPCs.

We performed *in vitro* differentiation of edited HSPCs towards the myeloid, macrophage, neutrophil, dendritic and megakaryocyte lineages. After confirmation of successful differentiation by evaluating cell-type specific markers expression, we assessed: (1) reactive oxygen species (CellROX) production of myeloid cells upon stimulation; (2) phospho-flow profiling of myeloid cells stimulated with several mediators (IL-4, PMA/ionomycin, GM-CSF, type-I interferon, IL-6, LPS); (3) phagocytosis of E.coli bacteria by macrophages; (4) M1 and M2 polarization of LPS or IL-4 stimulated macrophages; (5) expression of co-stimulatory molecules involved in antigen-presentation (HLA-DR, CD86) by classical DCs upon stimulation; (6) NETosis formation by granulocytes stimulated with PMA; (7) surface profiling and ploidy of megakaryocytes.

The results of these experiments are reported in a full new Extended Data Fig.6 and described in lines 283-291:” *In vitro differentiation of CD34+ HSPCs toward myeloid, macrophage, classical dendritic, granulocytic and megakaryocytic lineages was similar irrespective of editing condition and did not result in counterselection of edited cells (Extended Data Fig.6a). Functional validation assays of HSPC-derived differentiated cells showed similar results across all conditions, including: reactive oxygen species production by myeloid cells, E.coli phagocytosis by macrophages, M1/M2-like macrophage polarization, phospho-flow profiling of IL4-, PMA/ionomycin-, GM-CSF-, IFN type-I-, IL-6- and LPS-stimulated myeloid cells, HLA class-II/CD86 upregulation by DCs, induction of granulocyte NETosis and generation of hyperdiploid megakaryocytes (Extended Data Fig.6b,c,d,e,f,g,h,i). ”*

2.How about the gene expression profile of the differentiated untreated, AAVS1, FLT3 BE, CD123 BE and KIT BE HPSC cells? The RNA seq of these cells need to be added, which can assist in proving that epitope editing has no effect on HPSC cells.

We agree that an unbiased evaluation of the transcriptome-wide impact of epitope editing on stimulated and non-stimulated HSPCs would strongly reinforce our claims.

To address this issue, we have now performed RNAseq of FLT3, CD123, KIT, AAVS1 edited or mock-treated HSPCs, either unstimulated or stimulated with their respective ligand (AAVS1 and electroporation-only were stimulated with all three FLT3L, IL-3, SCF). The RNAseq analyses identified 78, 2667 and 7944 differentially expressed genes (DEGs) associated with FLT3L, SCF and IL-3 stimulation, respectively (Extended Data Fig.5b,c). The number of measured DEGs correlates well with the overall impact of each cytokine on the heterogenous CD34 stem/progenitor cells and Gene Set Enrichment Analysis (GSEA) confirmed that these genes are associated with the receptor-mediated proliferation pathways. By comparing FLT3, CD123, KIT edited conditions with the respective AAVS1 control conditions, we observed no significant transcriptional differences both at baseline and upon FLT3L, IL-3 and SCF stimulation (Fig.3h), thus confirming full preservation of the functionality and downstream effects of the epitope engineered receptors. When comparing base-edited HSPCs with mock controls, we observed only a few DEGs associated with the cell cycle and the protein processing in endoplasmic reticulum gene ontology pathways (Supplementary Data Fig.3a and Tables 6-9). While this late time point of analysis may

not detect short-term acute effects of the mRNA electroporation, these data confirm that the editing procedure has minimal impact on the treated cells.

These results are reported in Fig.3 and Extended Data Fig.5 and described in text lines 269-276: *“In order to comprehensively evaluate any transcriptional changes associated with the epitope editing procedure and the response of FLT3, CD123 and KIT modified cells upon stimulation with their respective ligands, we performed RNAseq of CD34+ HSPCs edited for FLT3, CD123, KIT, AAVSI or unedited cells (electroporation only), either stimulated or unstimulated with the respective ligand. We found 78, 2667 and 7944 differentially expressed genes (DEGs) associated with FLT3L, SCF and IL-3 stimulation, respectively (Extended Data Fig.5b,c). By comparing FLT3, CD123, KIT edited conditions with AAVSI control, we confirmed the absence of transcriptional differences both at baseline and upon FLT3L, IL-3 and SCF stimulation (Fig.3h).”*

As described in one reply to Reviewer #1, we also complemented the transcriptional analysis with a global proteomic assay by establishing a collaboration with Dr. Jarrod A. Marto, director of the Blais Proteomics Center at the Dana-Farber Cancer Institute. Thanks to his expertise, we were able to perform a phospho-proteome profiling of a large number of edited HSPCs, either at baseline or stimulated with their respective receptor ligands. Despite the technical challenges of this complex analysis, caused by the low protein content obtained from primary HSPCs and the high signal-to-noise ratio of this methodology, which did not allow us to perform in-depth inferential analyses, we were able to observe concordant effects of the stimulated samples across editing conditions and AAVSI controls.

These new data are now included in the new Extended Data Fig.5 and Supplementary Data Fig.3 and described in lines 275-279: *“Phospho-proteomic profiling by mass-spectrometry of FLT3, CD123, KIT or AAVSI edited CD34+ HSPCs showed concordant changes of differentially phosphorylated sites upon ligand stimulation between receptor-edited and AAVSI conditions (Extended Data Fig.5e), again confirming in an unbiased manner that activation of downstream signaling by epitope-modified receptors is preserved.”*

3.The mice models with AML PDX cells were suggested to be measured by in vivo imaging to observe the AML burden in mice directly. If the fluorescence in vivo imaging of mNeonGreen is not well, the luciferase can be used to transduced to PDX for bioluminescent imaging.

While we agree that transduction and *in vivo* monitoring of tumor models with Luciferase are possible and typically show excellent sensitivity, we chose to mark patient-derived AML xenografts only with the fluorescent marker mNeonGreen for several reasons:

- To improve transduction efficiency of difficult-to-transduce primary AML xenograft samples by minimizing transgene payload. Our PDX can only be expanded in immunodeficient mice and display significant mortality when cultured *in vitro*. For these reasons, we optimized a brief *ex vivo* culture and transduction protocol that takes advantage of stem-preserving compounds (SR-1, UM171), transduction enhancers (PGE2) and overnight exposure to high-titer 3rd generation LV particles ($>10^{10}$ TU/mL). mNeonGreen is relatively small (711 nt) compared to Firefly luciferase (fluc, 1653 nt), which contributes to the generation of high-quality (both high titer and infectivity) LV preps.

- To ensure 100% transduction of PDX cells by serial FACS-sorting. Clear detection of a fluorescent protein allowed us to FACS-purify transduced cells, serially passage them in secondary recipients and re-sort until we obtained a uniform population. The use of Firefly Luciferase would have required the inclusion of a fluorescent protein with a ribosomal-skipping peptide or an IRES, which would have increased vector size, lowered LV transduction efficiency, and penalized the expression of one of the two transgenes.
- To optimize detection by flow cytometry, which is the primary read-out of our experiments. To comprehensively assess the impact of CAR-T treatment on the different hematopoietic lineages of the hematochimeric mice we largely used multiparametric flow cytometry analyses, often associated with counting beads for absolute quantifications. mNeonGreen has one of the highest molecular brightness among green-fluorescent proteins (92.8)³¹ and was always easily discernible in our *in vivo* experiments. This allows high detection sensitivity and enables discrimination of the PDX from the healthy hematopoietic cells.

In the Response to the Referee Fig.4 we show representative flow cytometry plots that highlight the excellent discrimination of transduced AML PDX cells from healthy CD34⁺ derived hematopoiesis.

[Response to Referees] Figure 4. Detection of mNeonGreen transduced PDX cells in mice co-engrafted with healthy CD34⁺ HSPCs by flow cytometry. Representative FACS plots of BM samples from mice co-engrafted with CD34⁺ HSPCs and PDX-1, which shows a leukemia-associated immune-phenotype with co-expression of CD33⁺ and CD56^{bright}. (Top) Mice with only CD34⁺ derived hematopoiesis, (bottom) co-occurrence of healthy human hematopoiesis and AML PDX engraftment

4. Please add the result of *in vitro* Fab79D CAR killing assay on KITH378 epitope edited HSPCs.

As discussed above, we agree that the development of our anti-AML immunotherapy strategies targeting KIT is currently less advanced compared to FLT3 and CD123. We have exploited KIT CAR-T cells mainly as a tool for obtaining stringent *in vitro* validation of the role of H378R epitope editing to protect KIT-overexpressing K562 cells (Fig.2d). As previous work by Myburgh et al. (PMID: 32358567), KIT CAR-T cells generated with a similar version of the same Fab79D domain have already been shown to effectively deplete both healthy and malignant human hematopoietic cells both *in vitro* and in xeno-transplanted mice. While we have not extensively optimized our Fab79D CAR construct, we are now providing new experimental data supporting the protection of KIT^{H378R} BE HSPCs co-cultured with KIT CAR-T cells. As HSPCs cultured *in vitro* with SCF

express very low surface levels of KIT, the on-target killing observed in our experiments is not on par with what was observed with FLT3 or CD123 CAR-T cells. Nonetheless KIT BE CD34+ HSPCs are relatively preserved when compared with *AAVS1* control at high effector:target ratios (Response to the Referee Fig.2). However, since these results have been obtained with an underdeveloped version of the Fab79D CAR, we have reported these results in this point-by-point reply (which, in case of publication of this manuscript, will be available as online material), but we have not currently described them in the main text.

[Response to Referees] Figure 3. *KIT^{H378R} BE HSPCs are relatively resistant to Fab79D CAR-T cells in vitro. KIT epitope-edited HSPCs were co-cultured with Fab79D CAR-T cells or untransduced T cells (UT) at different E:T ratios. Outcome was evaluated at D7 by flow cytometry. Normalized absolute counts are reported in the plots.*

Moreover, we are currently evaluating the effects of KIT 79D mAb *in vivo* (see Response to Referee Fig.1) for a follow-up work. Thus, we believe that there is great value in reporting in this manuscript our current findings on KIT epitope engineering.

5. Figure 3i-k, please add an untreated group as a blank control, which is helpful to prove this gene editing technology does not affect the stemness and differentiation of HSPCs *in vivo*.

While we agree that an untreated group would provide additional information on the effects of electroporation and CRISPR-based editing on stem cell phenotype and functionality, the main goal of the experiment described in the original Fig.3i-k (Fig.3e-g in the revised manuscript) was the comparison of HSPCs epitope-edited and HSPCs with unmodified receptors both at steady state and after exposure to CAR-T cells or monoclonal antibodies. Editing of the AAVS1 safe harbor locus does not induce relevant on-target effects and is the preferred control for these types of experiments.

A recent work fully dedicated to assessing the impact of base editing on human HSPC and comparing it with conventional nuclease-based gene editing was presented at the last meeting of the American Society of Cell and Gene Therapy (ASGCT 2022, Fiumara et al., Molecular Therapy Vol 30 No 4S1, April 2022). In this work, the Authors showed that Adenine base-editing enzymes are less toxic compared to Cas9-editing and have no impact on the long-term, multilineage repopulating potential of the treated cells. The only detectable effect induced by ABE mRNA electroporation was a transient upregulation of interferon-stimulated genes, which could indicate innate cellular sensing of the long-encoding mRNAs, as we previously reported in a similar setting (PMID: 29021165). We are aiming to confirm these results also in our experimental setting in a follow-up study, in which we will perform a comprehensive panel of pre-clinical assays and experiments specifically directed to assess the overall safety of the proposed procedure in view of the future preparation of an IND dossier. As mentioned in a previous point, we have now included

in this manuscript a comparison of base-edited HSPCs with mock controls from our RNAseq analysis, where we observed only a few DEGs partially associated with the cell cycle and the protein processing in endoplasmic reticulum gene ontology pathways (Supplementary Data Fig.3a and Tables 6-9). While this late time point of analysis may not detect short-term acute effects of the mRNA electroporation (such as interferon responses), these data confirm that the editing procedure has minimal impact on the treated cells. Moreover, we included an untreated condition (UT) also when assessing i) HSPCs phenotypic composition during *in vitro* culture (Fig.3d), ii) colony forming potential of treated HSPCs (Fig.3i), iii) *in vitro* differentiation of edited HSPCs towards the myeloid, macrophage, neutrophil, dendritic and megakaryocyte lineages (Extended Data Fig.6), and iv) functionality of the differentiated cells. In all these experimental settings, the epitope-edited cells were undistinguishable with respect to UT conditions.

Finally, it has to be mentioned that the genome editing technologies employed in this manuscript are already being evaluated in clinical trials, as in the case of BEACON study for Sickle Cell disease (NCT05456880) or CD7/CD52/TRAC triple base edited CAR-T cells for the treatment of T-lymphoblastic leukemia (ISRCTN15323014³²). Publication of the results of these first-in-human trials will provide invaluable information on the effects and the safety of the base editing procedure on cell function.

6. Figure 3j, please add the flow cytometry analysis results of human engraftment (hCD45+ cells) in the secondary transplanted mice.

We apologize for the lack of clarity. The flow cytometry analysis results of human engraftment (hCD45+ cells) for the secondary transplantation of the experiment described in Fig.3j (FLT3 edited HSPCs) is reported in Extended Data Fig.7a-b.

In addition to the secondary transplantation of FLT3-edited HSPCs, we have now performed secondary transplants of CD123 and KIT epitope-edited HSPCs. As described above in response to one of the Reviewer #1 questions, we used as cell source the frozen bone marrow cells of the primary mice described in the original Extended Data Fig.5. Cells from these samples were thawed and, in the case of CD123, human HSPCs were sorted for removing the AML PDX. Despite the low amount of starting material, these experiments confirmed the long-term engraftment capacity and persistence of both KIT and CD123 edited HSPCs without counterselection effects.

These experiments are now reported in *Extended Data Fig.7e-j* and described in lines 307-312: *”Similar results were also observed for secondary transplantation of BM KIT^{H378} cells, which generated grafts with comparable composition as AAVS1^{BE} controls (Extended Data Fig.7e-f), with no change in editing levels, both measured by molecular or flow cytometry analyses (Extended Data Fig.7g,h). As done for FLT3 and KIT, we confirmed the engraftment and the stability of CD123^{S59} BE levels in both primary and secondary TXs (Extended Data Fig.7i,j).”*

7. Line 321 and in Extended Data Fig. 6f and g, the results showed the phenotype of CAR T cells had no difference at the end of the experiment, except the PD-1 expression. Please add the results of the activation and degranulation of CAR T cells including CD69 and CD107a.

We did not observe differences in CD69 expression on CD4+ or CD8+ CAR T cells *in vivo* at the evaluated time point (14 days after CAR infusion). The plots reporting CD69 expression were

added to Extended Data Fig. 8g). Degranulation was not measured on BM/SP samples due to poor specific signal of the CD107a staining on cells isolated from mouse tissues.

8. Figure 6, dual-specific FLT3/CD123 CAR-T cells were used in *in vitro* experiments, while 1:1 pool of 4G8 and CSL362 CAR-T cells were used in *in vivo* experiments, why? How about the anti-AML effect of dual-specific FLT3/CD123 CAR-T cells *in vivo*?

To provide proof of concept that targeting more than one antigen at the same time could result in improved anti-AML efficacy and that multiplex epitope-editing could protect the healthy hematopoiesis in this setting, we used pooled FLT3 and CD123 CAR-T cells *in vivo*. This was done for two reasons: (1) to rely on previously *in-vivo* validated single CAR-expressing T cells (2) to reduce the risk of excessive T cell activation and exhaustion by co-expressing complete 2nd generation CARs in the same cell, which might result in overstimulation. Indeed double or triple CAR expression in the same T cell might increase the chances of exhaustion and/or cause excessive activation and cytokine secretion, leading to loss of on-target killing and/or higher risks of systemic toxicity.

For our *in vitro*, where the exposure to the antigen is limited to 48h and the main goal was to stringently validate the protection conferred by epitope-editing, we decided to test T cells co-expressing multiple CARs. We are currently testing different configurations of bi-/tri-specific CARs in which we are shuffling the intracellular activation and second stimulation domains, as reported in PMID: 34746799, to identify the optimal configuration for co-stimulation and metabolic fitness and to compare them vs a double/triple pool CAR-T cell treatment. If successful, these studies will be part of a follow-up work.

9. Extended Data Fig. 7, the proportion of AML within BM CD45⁺ were more than 80% in b, while the proportion of AML within BM CD45⁺ were less than 20% in d, please explain. How about combination of 4G8, CSL362 and 79D CAR T cells?

We thank the Reviewer for spotting this detail, which allowed us to identify and correct an error in the experimental timeline depicted in Extended Data Fig. 9b and d. Mice reported in Fig. 9b received the AML PDX cells by tail vein injection at day 0, the CAR-T cells at day 10 and were sacrificed 18 days later (28 days since day 0). At day 28, several mice exhibited hunched posture and partial paralysis of hind limbs. To avoid animal suffering (as per IACUC protocol) and to better appreciate differences in CAR-mediated killing of AML cells, all subsequent experiments, including the one described in Extended Data Fig. 9d, were performed with a 14 days interval between CAR treatment and euthanasia (a total of 24 days between AML PDX injection and endpoint). Most likely, this was the main reason why the AML burden in the bone marrow (expressed as % hCD45⁺ w/in total CD45⁺) was lower compared to the pilot experiment in Extended Data FIG. 9b. We apologize for the mistake and we are now reporting the correct sacrifice time points in the experimental drawings.

While we think that multiple-target immunotherapies are key to AML eradication, we did not evaluate combinations of all three CAR-T cells *in vivo*. As described above, we will test testing different configurations of bi-/tri-specific CARs in a follow-up work. However, we have performed and included in this manuscript an *in vitro* killing assay in which triple-specific

FLT3/CD123/KIT CAR-T cells were co-culture with triple epitope-edited K562 reporter cells to demonstrate the possibility of further multiplex the epitope-engineering approach.

This experiment is described in Extended Data Fig.10 and described in lines 442-447: *”To demonstrate the potential of multiplex epitope-editing, we repeated this experiment with all 3 targets (FLT3, CD123, KIT) on triple-reporter K562 cells, FACS sorted all editing combinations and co-cultured them with triple-specific CAR-T cells (Extended Data Fig.10a). Only triple edited cells were able to survive CAR-mediated killing, prevent T cell activation, degranulation and cytokine secretion (IFN- γ and TNF- α , Extended Data Fig.10b,c), while still expressing FLT3, CD123 and KIT on their surface (Extended Data Fig.10c).”*

10.FLT3 (CD135) was expressed more than 80% in both PDX-1 and PDX-2 cells, but the anti-leukemia effect of 4G8 was less effective in killing PDX-2 than PDX-1, please explain.

While both PDX-1 and PDX-2 had a high proportion of FLT3+ cells (>80%, Extended Data Fig.7m), the levels of FLT3 expression (MFI) in the BM and SP samples of PDX-2 are lower than PDX-1 (Extended Data FIG.9b). More in detail, the FLT3 MFI of PDX-2 in untreated mice is on-par with the FLT3 MFI of the few surviving cells of PDX-1 after exposure to 4G8 CAR-T cells. While we cannot formally exclude other escape mechanisms (e.g., expression of immune blockage molecules), it is well known that target expression levels play a fundamental role in determining the susceptibility to CAR-mediated killing, as previously reported with a similar 4G8 CAR³³. This observation further underlines the importance of targeting multiple surface molecules and escape mechanisms in the treatment of AML.

11.The authors performed double editing of FLT3 and CD123, how about editing of FLT3, CD123 and KIT simultaneously on K562 or HSPCs? Will the effect be better?

We thank the Reviewer for his/her interest in the versatility of epitope engineering with base-editing tools. As described above, to provide proof of concept of triple editing/targeting, we performed multiplex editing of the FLT3, CD123 and KIT epitopes in triple-positive K562 cells, FACS-sorted all editing combinations (N = 8, ie. non-edited, single edited, dual edited or triple edited) and co-cultured them with triple-specific FLT3/CD123/KIT CAR-T cells (Extended Data Fig.10). We observed intermediated levels of protection for single and dual-gene edited cells, while only triple edited cells were fully spared from CAR-mediated killing and were on-par with K562 cells that do not express any of the three targets. While the possibility of multiplex editing and CAR treatment could facilitate the treatment of heterogeneous malignancies such as AML, as discussed above, further studies have to be conducted in order to optimize the design of the multi-specific CAR and confirm the overall safety of multiplex HSPCs editing.

The results of this new experiment are reported in Extended Data Fig.10a and described in lines 441-446.

12.Why choose FLT3, CD123 and KIT? CD33 and CD38 were expressed higher than these markers expressed in PDX cells showed in Extended Data Fig. 5g.

We selected our targets based on their well-characterized functional role in AML biology, to maximize the chances of complete disease eradication and reduce the risk of tumor immune escape.

CD33 is expressed with variable levels in the majority of AML cases (85-90%) but, with the possible exception of its inhibitory effects dependent on Syk expression³⁴, it does not seem to play a significant role in AML biology. As we mentioned in the introduction, while anti-CD33 immunotherapies could provide therapeutic efficacy in AML patients, the immune pressure against a gene non-essential for AML survival could facilitate the occurrence of tumor escape mechanisms through Ag loss or downregulation, as observed in CD19-negative relapses after CD19 CAR-T therapies or mismatched HLA loss after haploidentical HSCT of AML patients.

CD38 is heterogeneously expressed on total AML cells³⁵⁻³⁷, while leukemia stem cells (LSCs), characterized by increased leukemia repopulating capacity and resistance to treatment, have been found to be enriched within the CD34+CD38- subset.³⁸⁻⁴⁷

For these reasons, while CD33 and CD38 could serve as excellent targets to debulk AML blasts, they are more likely to spare LSCs-enriched subsets or to be downregulated compared to cytokine receptors with a well-characterized function and expression on LSCs^{45,48-51}.

Minor

1. Figure 1c, the western blot image of FLT3 was cropped. Please change with an original image of western blot.

The western blot images were cropped due to space constraints (some blots have empty lanes between each sample to avoid cross-contamination). Uncropped, original images of the western blot membranes have been added (Supplementary Data Fig.1 and 2).

2. Figure 1, whether the compensation of some flow cytometry plots was adjusted well, please confirm.

Appropriate control plots (fluorescence-minus-one, FMO) have been added (Supplementary Data Fig.1a) to show that compensation was correctly applied to Fig.1.

3. Line 231, “After optimization of mRNA *in vitro* transcription, culture and electroporation conditions and editing time point after HSPCs stimulation (Extended Data Fig.4D,E,F), we achieved up to 86.6%, 78.6% and 67.9% of target A>G conversion for FLT3, KIT and CD123, respectively. We were able to efficiently edit the target adenines within the windows of FLT3-18, KIT-Y and CD123-R sgRNAs (Fig.3B)”. Which figure is corresponding to “86.6%, 78.6% and 67.9%”?

We thank the Reviewer for spotting this point. The reported editing efficiencies for FLT3, KIT, CD123) refer to samples used in *in vivo* experiments. The sentence has been updated to: “After optimization of mRNA *in vitro* transcription, culture and electroporation conditions and editing time point after HSPCs stimulation (Extended Data Fig.4d,e,f), we achieved up to 86.6%, 78.6% and 78.9% of target A>G conversion for FLT3, KIT and CD123, respectively (Fig.4b, Fig.5b and Extended Data Fig.7g).”

The source of the editing efficiencies is now clearly indicated and updated with our optimized results.

4. For *in vitro* and *in vivo* assays, please indicate whether the HSPCs were treated with autologous CAR T cells.

The presented experiments were performed with non-matched healthy donor-derived CAR-T cells. This information has been clarified in line 1420 within the Methods section.

In vitro assays with cell lines (Fig.2 and 6; Extended Data Fig.3 and 10) were performed by co-culturing K562 cells, either over-expressing the target Ag by EF1 α promoter integration or Sleeping Beauty gene addition, with healthy-donor derived CAR-T cells. As K562 cells do not express surface HLA, alloreactivity effects can be excluded. Appropriate controls, including cultures with antigen-negative K562 and T cells not expressing the CAR (untransduced, UT) were included to support this conclusion.

Similarly, CD34⁺ HSPCs were co-cultured with non-matched healthy-donor derived CAR-T cells (Fig.3). Non-CAR specific effects can be excluded due to (1) the short duration of the co-culture (48h), (2) inclusion of the untransduced T cells controls, which do not induce a decrease of CD34⁺ cells absolute counts compared to T0.

To exclude the influence of alloreactivity during *in vivo* experiments, we limited the exposure to CAR-T cells to 14 days before euthanasia and performed a dedicated experiment with transplantation of untransduced T cells to evaluate their effect on the human graft *in vivo* (now included in Extended Data FIG.7n). We did not observe any effect on hematopoietic composition except for (1) a significant decrease in HSC frequency (proportional to CD3⁺ expansion) and (2) a slight increase in mature monocytes, possibly due to T cell mediated cytokine release. These observations support the reliability of our results and the target specificity of our findings with FLT3 and CD123-targeted CAR-T cells *in vivo*.

5. Figure 4e, please align the “+ -” with the figure.

We thank the Reviewer for spotting this graphical mistake. Figure 4e was updated with aligned +/- references.

6. Figure 4b, 5b and 6g, why was the proportion of FLT3 base editing different in the CD33⁺ and CD19⁺ BM cell from mice treated and untreated with CAR-T cells? Moreover, the proportion of CD123 editing had no difference. Please explain.

Fig.4b shows the FLT3 editing efficiency in CD34⁺ liquid culture, mouse peripheral blood and sorted BM CD33⁺ myeloid and CD19⁺ lymphoid subsets at the end of the experiment. There are no significant differences between BM CD33⁺ and CD19⁺ in untreated mice (light pink), supporting the lack of any lineage skewing induced by the FLT3 editing procedure. The significant p values refer to the difference between untreated and 4G8-CAR treated conditions, within the CD33 and CD19 subsets. This is expected due to the negative selection of non-epitope-edited cells and provides further evidence for the protection of edited hematopoietic lineages. Furthermore,

the editing increase is more pronounced within CD19+ cells, consistent with the pattern of FLT3 expression on lymphoid cells that are thus better targeted by CAR-T. While the same comparison did not reach statistical significance in the CD123 experiment ($p=0.06$), there is a clear trend toward editing increase within the CD33+ subset (Fig.5b), again consistent with the myeloid depleting effect of CD123-targeted CAR-T cells. As mentioned in the reply to Reviewer #1 and noted in the discussion of this manuscript, this strategy can be exploited to confer a selective advantage to genetically engineered cells in the context of non-genotoxic pre-transplant conditioning, thus possibly broadening the applicability of epitope engineering also to non-malignant diseases.

7. Figure 6f, why was the proportion of AML within hCD45+ cells higher in FLT3-CD123 than in AAVS1? Please explain.

The reported differences in AML % within hCD45+ are due to experimental inter-mouse variability, which is expected when working with primary patient-derived xenografts. Additionally, our model of AML co-engraftment in humanized NBSGW mice is affected by the interaction between pre-engrafted HSPCs, murine hematopoiesis and the transplanted AML cells, which contributes to the observed variability. Nevertheless, the observed variability in AML content does not affect the results, given that dual CAR-T cell treatment resulted in the complete eradication of AML cells.

Referee #3 (Remarks to the Author):

The manuscript by Casirati and colleagues uses base editing techniques to engineer HSPCs for bone marrow transplantation. The engineering allows hematopoietic lineages with selective resistance to CAR-T or monoclonal antibody (mAb) therapy without affecting HSPC function. This is very important in the AML field because the targets for CAR-T or monoclonal antibody (mAb) therapy are also expressed on HSPCs, and thereby, these therapies usually have on-target/off-tumor toxicity. Compared to the work in the literature regarding gene editing to knock out the entire stem cell marker, the point mutation of the current work to edit the genome of stem cells is not conceptually novel. The authors' data on stem cell resistance to CAR T cell treatment do not match their hypothesis (see detailed comments below). The long-term safety issue for changing stem cells is unclear. The authors also need to discuss in detail how to practice this in future clinical applications.

We thank the Reviewer for acknowledging the significance and relevance of our work for the AML field and for his/her interest in the possible clinical setting for potential future clinical applications.

Below are specific comments of this reviewer:

1. It is unclear whether the reported gene editing has a safety issue or not. For example, it may change the malignancy potential of HSPCs. This at least should be discussed.

We agree that any genome modification performed on stem cells intrinsically bears the possibility to perturb the safety of the reconstituted hematopoiesis in terms of functionality or malignancy potential, which is often difficult to model *in vitro* and in animal models. We have put considerable effort into the validation of: (1) the variant receptors and their function; (2) the safety of the genome-editing procedure; (3) the phenotype and functionality of epitope-edited CD34+ HSPCs. As described above (please, see also reply to Reviewer #1), the revised manuscript provides new experimental data to support the safety of our approach (RNAseq, mass-spectrometry, off-target analysis, validation of differentiated hematopoietic lineages, additional secondary transplantation experiments).

For FLT3 and KIT, two genes commonly mutated in myeloid malignancies, we compared the position of our epitope amino-acid mutations (N399 and H378, respectively) to annotated mutations reported in the COSMIC database and found no variants associated with hematological malignancies (Response to the Referee FIG.5).

[Response to the Referees] Figure 5 - Distribution of COSMIC annotated mutations in the FLT3 and KIT genes. FLT3 (left) and KIT (right) amino-acid positions commonly mutated in human cancers. The position of the N399 FLT3 mutation and the H378 KIT mutation are indicated by the red arrows. Modified from Kazi and Rönstrand, <https://doi.org/10.1152/physrev.00029.2018> (Left); Lennartsson, and Rönstrand <https://doi.org/10.1152/physrev.00046.2011> (right).

While overall our encouraging results suggest a promising safety profile of our selected base editing enzymes, we continue to highlight in the discussion the importance of performing a more comprehensive characterization of the specificity of gRNA and non-gRNA dependent off-target activity as well as a careful evaluation of risk/benefit ratio before clinical implementation (lines 531-557).

2. As the authors mentioned, several studies of gene-based editing of HSPCs targeting cancer and other diseases have been reported, including PMC7869435 and PMC4709030. Some similar studies are already tested in the early phase clinical stage (e.g., NCT04849910). This makes the current study incremental but not conceptually novel in the field.

The publication by Jing et al. (PMC7869435) reports the use of CRISPR cytidine base editing RNP to disrupt BCL11A erythroid enhancer in human CD34+ HSPCs to induce hemoglobin F re-expression and provide a therapeutic option for sickle cell disease patients without the need for

nuclease-mediated knock-out. While this paper describes an application of CRISPR base editing to correct hematological conditions, it is not related to the application of immunotherapies for hematological cancers nor to the modification of surface epitopes. The publication by Hendriks et al. (PMC4709030) describes the use of designer nucleases to edit human pluripotent stem cells.

NCT04849910 is a phase 1/2 clinical trial of evaluating the safety of VOR33, an allogeneic transplant product with knock-out of CD33, coupled with post-transplant administration of the CD33-targeted ADC Mylotarg to reduce the risk of AML relapse. Whereby this elegant work exploits a similar general strategy of removing the target epitope from transplanted cells as proposed in this manuscript, there are a few key differences: (1) the removal of the target antigen is achieved through CRISPR-Cas9 knock-out, which does not preserve the function of the selected target and exposes HSPCs to the genotoxic risks of DNA double-strand breaks (DSBs). Epitope editing by means of base editing preserves protein function and introduces desirable mutations without DSBs. (2) the chosen target antigen (CD33) is irrelevant for hematopoietic development and myeloid differentiation. Targeting a non-essential Ag bears the inherent risk of immune escape by AML cells, which can downregulate or lose target expression. FLT3, CD123 and KIT are cytokine receptors involved in cell proliferation and frequently mutated or overexpressed in AML and therefore less likely to be downregulated or lost, lowering the risk of Ag-negative relapse. (3) The therapeutic agent is an antibody-drug conjugate, gemtuzumab ozogamicin (GO), which has recently been re-approved for AML treatment. While hematopoietic toxicity is a concern, GO shows dose-limiting hepatic toxicity, including veno-occlusive disease, reported as an FDA Boxed Warning. CD33-KO HSPCs do not address the risk of severe and potentially fatal hepatic toxicity.

In our manuscript, we propose the use of chimeric antigen receptors or naked monoclonal Abs, which can exploit potent on-target killing without the additional side effects associated with the toxin payload.

As mentioned in the Introduction and the Discussion sections, we believe that our strategy shows significant novelty compared to alternative approaches, as demonstrated by a great deal of interest received at the 2022 ASH meeting oral presentation. Furthermore, analogous approaches targeting CD45 are currently under development by independent investigators at the University of Pennsylvania⁵².

3. Figure 4 (M and P): The main issue is the number of HSCs in the FLT3 BE/ + 4G8 group was obviously decreased compared to the no-CAR group, suggesting engineered FLT3 BE cannot completely resist CAR T therapy or the survival of edited HSPCs was decreased. Importantly, there is no significant difference between the editing group and the non-editing control group in the presence of CAR T cells. The same issue exists in the CD123 BE + CSL362 group vs. the no-CAR group in Figure 5, and in the FLT3/CD123 BE + 4G8/CSL362 group vs. the no-CAR group in Figure 6. This does not match the author's hypothesis.

We thank the reviewer for highlighting this important aspect of our results, which we agree deserves additional clarification. As discussed above in the reply to Reviewer #1, the reduction of HSC abundance with CAR-T cell treatment appears to be a non-specific effect associated with any T cell treatment in our humanized NBSGW model instead of a CAR-mediated on-target effect. To confirm this, we performed an experiment in which NBSGW mice engrafted with CD34+ HSPCs

received 2.5 M of untransduced T cells (cultured and expanded *in vitro* similarly to CAR-transduced T cells). Mice were then monitored for 2 weeks and then euthanized to assess bone marrow lineage composition. We did not observe any effect induced by untransduced T cells on hematopoietic lineage composition, except for a dramatic reduction of HSC frequency (0.08x, i.e. ~12.5-fold reduction), a mild increase in monocytes (2.57x), and a trend towards increased mature B cells (1.97x; new Extended Data Fig.7n). This most likely reflects a cytokine-induced mechanism associated with T cells engraftment and expansion in mice. We updated Extended Data Fig.7n to include the results of this important control experiment.

While FLT3 and CD123 are expressed on early progenitors, as reported in the flow analysis of human grafts (Extended Data 8e and 9a) their expression on phenotypically defined HSCs (lineage-CD34+38-10-45RA-90+) *in vivo* is low, thus the actual specific CAR-T-mediated killing of HSCs may play less a significant role compared to other progenitor subsets and is further masked by the aforementioned unspecific bystander T cell effect. Nonetheless, our data showed some degree of protection of epitope-edited HSCs that, albeit not reaching statistical significance, was consistent across all the described *in vivo* CAR experiments (3.1x, 5.3x and 7.1x more HSCs in the edited group with FLT3, CD123, and FLT3+CD123 combined editing, respectively, FIG,4m, 5p, 6h). We have now reorganized the plots in Fig.5 and Fig.6, by separating the progenitor populations (which have the lower absolute numbers) and changing the scale axes, to better highlight the protective effect on edited HSC.

4. Figure 2 (C-E): Is there a change in the production of cytokines by CAR T cells cocultured with HSPCs? The statistical data showed three groups (at the right), which is not consistent with the two groups of representative flow data (at the left). The testing groups for all p value calculations are not clear.

We thank the Reviewer for raising these points. We have evaluated multiple read-outs for our *in vitro* co-culture experiments, including: stringent cell viability (by 7AAD and AnnexinV staining), absolute cell counts (CountBeads enumeration), expression of recognized molecule on target cells (expressed as % positive cells or MFI), T cell activation (CD69+ fraction), T cell degranulation (CD107a surface staining), T cell proliferation (CellTrace dilution) and included both untransduced T cells conditions and cells not expressing the target molecule as negative controls. While we did not routinely include evaluation of cytokine secretion as a standard readout, we have performed a new experiment with FLT3/CD123/KIT triple-specific CAR-T cells co-cultured with target cells edited for all combinations of the three epitopes (FLT3, CD123, KIT). In addition to the aforementioned readouts, we included IFN γ and TNF α secretion by T cells (Extended Data Fig.10), confirming that edited cells do not induce cytokine secretion on CAR-T cells.

5. Figure 3 (J): The absolute cell numbers of each cell type should be provided. Data for various tissues such as blood and liver should be added.

The absolute cell numbers of the FLT3 edited HSPCs (Fig.3J) in the secondary recipients are reported in Extended Data Fig. 7b. Absolute numbers of primary transplants are not available because flow cytometry analyses of this experiment were performed without the cell counting beads, however absolute counts of different lineages of primary FLT3 edited transplants are

reported in Fig.4m,p. Moreover, we have now included full tables reporting the absolute cell counts of hematopoietic populations from *in vivo* experiments as Source Data and Supplementary Data Table 23.

As standard read-out of *in vivo* experiments on humanized immunodeficient mice, we analyze hematopoietic organs, including peripheral blood, bone marrow and spleen. Liver is not a standard hematopoietic organ in adult mice and was not collected.

6. The authors only showed one signaling pathway was not impaired in the edited HSPCs. This is not sufficient for the authors to draw the conclusion that the FLT3, KIT, and CD123 variants expressed in HSPCs preserved their signaling. The analysis of more signaling pathways downstream of FLT3, KIT, and CD123 should be provided.

We agree that the point raised by the Referee is of great importance in validating the functionality of epitope-edited cytokine/growth factor receptors and edited HSPCs. To this end, we have now included a comprehensive and unbiased analysis of receptor activation and downstream signaling by (1) performing global RNAseq of CD34+ HSPCs edited for FLT3, CD123, KIT, AAVS1 or unedited (electroporation only), either unstimulated or stimulated with each respective ligand, in biological triplicate; (2) investigating the phospho-proteome profile of stimulated or unstimulated edited HSPCs by mass-spectrometry (MS), in biological duplicate. Additionally, we have interrogated a wider panel of intracellular pathways by phospho-flow on *in vitro* differentiated myeloid cells stimulated with several mediators.

The RNAseq analyses identified 78, 2667 and 7944 differentially expressed genes (DEGs) associated with FLT3L, SCF and IL-3 stimulation, respectively (Extended Data Fig.5b,c). The number of measured DEGs correlates well with the overall impact of each cytokine on the heterogeneous CD34 stem/progenitor cells and Gene Set Enrichment Analysis (GSEA) confirmed that these genes are associated with the receptor-mediated proliferation pathways. By comparing FLT3, CD123, KIT edited conditions with the respective AAVS1 control conditions, we observed no significant transcriptional differences both at baseline and upon FLT3L, IL-3 and SCF stimulation (FIG.3h), thus confirming full preservation of the functionality and downstream effects of the epitope engineered receptors.

These results are reported in Fig.3 and Extended Data Fig.5 and described in text lines 269-276: *“In order to comprehensively evaluate any transcriptional changes associated with the epitope editing procedure and the response of FLT3, CD123 and KIT modified cells upon stimulation with their respective ligands, we performed RNAseq of CD34+ HSPCs edited for FLT3, CD123, KIT, AAVS1 or unedited cells (electroporation only), either stimulated or unstimulated with the respective ligand. We found 78, 2667 and 7944 differentially expressed genes (DEGs) associated with FLT3L, SCF and IL-3 stimulation, respectively (Extended Data Fig.5b,c). By comparing FLT3, CD123, KIT edited conditions with AAVS1 control, we confirmed the absence of transcriptional differences both at baseline and upon FLT3L, IL-3 and SCF stimulation (Fig.3h).”*

In order to complement the transcriptional analysis with a global proteomic assay, we established a collaboration with Dr. Jarrod A. Marto, director of the Blais Proteomics Center at the Dana-Farber Cancer Institute. Thanks to his expertise, we were able to perform a phospho-proteome profiling of a large number of edited HSPCs, either at baseline or stimulated with their respective

receptor ligands. Despite the technical challenges of this complex analysis, caused by the low protein content obtained from primary HSPCs and the high signal-to-noise ratio of this methodology, which did not allow us to perform in-depth inferential analyses, we were able to observe concordant effects of the stimulated samples across editing conditions and AAVS1 controls.

These new data are now included in the new Extended Data Fig.5 and Supplementary Data Fig.3 and described in lines 275-279: *“Phospho-proteomic profiling by mass-spectrometry of FLT3, CD123, KIT or AAVS1 edited CD34+ HSPCs showed concordant changes of differentially phosphorylated sites upon ligand stimulation between receptor-edited and AAVS1 conditions (Extended Data Fig.5e), again confirming in an unbiased manner that activation of downstream signaling by epitope-modified receptors is preserved.”*

7. It is unclear how this will be practiced in the clinic in the future. For example, is this mainly for preventing relapse post-transplantation? The cost and the time to engineer stem cells to meet clinical applications also need to be considered.

We thank the Reviewer for his/her interest in the possible clinical setting for potential future clinical applications. In our opinion, there are several potential routes to exploit epitope editing in the clinical setting, as also confirmed by the fact that an early-phase clinical trial using a similar strategy is currently ongoing (NCT04849910). One unmet clinical need is the effective treatment of high-risk AML with poor response or upfront refractoriness to induction chemotherapy. Patients who cannot achieve a meaningful remission before allogeneic stem cell transplantation often show measurable minimal residual disease (MRD) and are likely to relapse early after HSCT, which leaves the patient with few therapeutic options and ultimately leads to unfavorable outcomes.

One possible clinical setting entails the administration of an epitope-edited graft for fit patients diagnosed with high-risk disease (eg. According to ELN2022 classification) for which allo-HSCT in first remission (or at least best response) constitutes standard clinical practice. The patient would receive one or two induction regimens while a search for a suitable donor and HSPC engineering is underway. The same apheresis used to harvest mobilized CD34+ HSPCs would provide T cells for the production of the CAR-T cell product, which can be frozen for future use. The patient would then receive a standard myeloablative conditioning regimen and the epitope-modified graft (including non-edited T cells). FLT3-targeted CAR-T cells would be administered prophylactically at the time of hematopoietic reconstitution to provide an adoptive anti-leukemia effect to eradicate residual disease. A safety switch (eg. Truncated EGFR or inducible Caspase 9) embedded in the CAR design would provide a means to moderate potential toxicity. Alternatively, post-transplant anti-leukemia maintenance with anti-FLT3, CD123 or KIT antibodies could be administered periodically up to 24 months to decrease the chances of MRD emergence without inducing on-target hematopoietic toxicity.

Patients with intermediate-risk disease may receive an epitope-edited allograft in the first remission and closely monitor molecular or flow-cytometric MRD to guide the use of adoptive immunotherapies. In this scenario, CAR-T cells or monoclonal antibodies would be administered only for patients with sustained MRD positivity or imminent relapse.

Finally, after the safety and efficacy of this approach are confirmed in clinical settings with a highly favorable risk/benefit ratio, CAR or HSPC-targeted antibodies could also be used to replace HSCT myeloablative conditioning altogether, by exerting both anti-AML and myeloablative effects to free the BM niche and lower the risk of relapse (pre-transplant T cell depleting agents would still be required to ensure sufficient host immunosuppression to avoid graft rejection and enable CAR expansion). While this setting is particularly appealing to reduce conditioning-related toxicities, it would require careful design and titration of each therapeutic agent to ensure efficient engraftment.

We have updated our discussion to include more details about the possible first in human testing of this strategy and its cost-related issues: *“Overall, we envisage a straightforward path to clinical translation of this strategy, given the growing clinical experience with HSPCs genome editing strategies and the fact that the immunotherapies based on our selected Abs have already reached clinical testing (NCT02789254; NCT02642016; talacotuzumab). We expect that our approach could benefit several AML patients and disease subtypes, and in particular cases that display high-risk features at diagnosis or that struggle to achieve a deep remission with standard treatment. HSCT is largely used for the treatment of high-risk AML patients, but their long-term survival hinges entirely on the remission status before HSCT, the disease biology and the delicate balance between graft vs leukemia (GvL) and GvHD effects. In the presence of post-transplant minimal residual disease (MRD) or, worse still, relapse, these patients are left with little to no treatment options. Our epitope engineering strategy could be rapidly implemented in currently used allogeneic-HSCT protocols to enable additional therapeutic options for MRD eradication or anti-leukemia maintenance regimens after HSCT. Moreover, the use of HSPCs from healthy donors will avoid the risk of inadvertently editing residual AML cells from the host. **A paradigmatic therapeutic setting for first-in-human testing of this strategy could be the administration of CAR-T cells at the time of hematopoietic reconstitution after HSCT of relapsed/refractory or high-risk patients transplanted in presence of MRD, to prevent the occurrence of early relapse. Another intriguing option, also proposed by other groups⁶⁷, would entail the use of CAR-T therapy itself as myeloablating conditioning before HSCT to concomitantly kill leukemic cells and free the bone marrow niche for the engraftment of the epitope-engineered HSPC graft. The elimination of chemo- or radio-therapy mediated myeloablation would also likely minimize the risk of GvHD due to lower tissue damage and reduced release of pro-inflammatory mediators. Whereas gene editing is a personalized medicine entailing costly and complex procedures, it has the potential to establish an enduring benefit with substantial savings on the cost of repeated patient hospitalizations and the administration of conventional therapies.”***

8. For CAR-T cell generation in Figure 2, the authors used different MOIs to infect T cells, but the infection ratio and expansion were very similar among different MOIs. The results should be confirmed.

For our CAR-T cell production, we used third-generation lentiviral vectors produced at high titer and infectivity. In Fig.2 we optimized our T cell lentiviral transduction (TDX) protocol by titrating the MOI used (MOIs 2, 5, 10, 15). This resulted in high transduction efficiency already at MOI = 2, and reached saturation at MOI = 10. As we did not observe negative impact on T cell expansion, all subsequent transduction experiments were performed with MOI = 10, which resulted in reproducible TDX levels.

9. The potential gRNA off-targeting in stem cells may lead to a safety issue.

Off-target effects (OT) constitute a critical aspect of any genome editing technique. While CRISPR-Adenine Base Editing (ABE) has demonstrated considerable advantages compared to other gene-editing technologies (eg. nuclease mediated homology-directed repair or even Cytidine Base Editors) due to the lack of DNA double-strand breaks, minimal non-gRNA-mediated genomic deamination, no risk of introducing STOP codons and the inherent safety associated with the redundancy of the genomic code, a comprehensive evaluation of potential genomic OT and their impact on cell functionality and fitness must be addressed before moving any therapeutic strategy toward clinical translation.

While such an extensive and in-depth investigation is beyond the scope of this work, we have now included a new full paragraph in which we provide promising pieces of evidence of the prospective safety of our epitope engineering approach. The potential gRNA-dependent off-target (OT) effects of the utilized base editors were evaluated by combining i) a genome-wide, unbiased off-target identification assay (GUIDEseq), performed in collaboration with Dr. Daniel Bauer (Boston Children's Hospital), and ii) an in silico prediction analysis. To adapt the GUIDEseq assay to our SpRY-based enzyme, we generated a SpRY-nuclease mRNA and used it to perform the dsDNA oligo-bait trapping in a permissive cell line (293T), to possibly increase the sensitivity of the screening. Despite this, the analysis led to the identification of few potential OT sites for all the tested gRNAs (N = 12 intronic and 9 intergenic for FLT3-18, N = 1 intergenic for KIT-Y and N = 1 intergenic for CD123-R, Supplementary Data FIG.4b and Table 16). Since none of the identified OTs fell into a coding or canonical splicing sequence, even if unspecific deaminations occur in these regions they will likely have no functional consequences. We then decided to extend this analysis and characterize the top off-target genomic loci for FLT3, CD123, and KIT sgRNAs predicted by the CRISPOR tool on the basis of sequence homologies with the gRNAs. For these sites, we assess the actual OT deamination occurring on CD34+ HSPCs treated for base editing by performing targeted next-generation sequencing. This analysis showed no significant off-target deamination over background for any of the analyzed CD123 or KIT sites (Supplementary Data Figure 4c). However, for the FLT3-18 sgRNA, we observed comparatively higher deamination in four loci: three intronic (ERN1, NFX1, RP11-242J7) and one exonic (SNTG1). While the unspecific base editing of these intronic sites (distant from canonical splicing sequence) likely has no functional consequences, the low but detectable edit (~5%) of the SNTG1 exon might possibly result in amino acid substitution of the affected protein. Luckily, however, the protein encoded by the *SNTG1* gene (Syntrophin γ 1) is a brain-specific protein of the syntrophin family that is not expressed and does not have any known functional role in hematopoietic tissue¹⁻³. By reanalyzing our RNAseq data, we also confirmed that the SNTG1 transcript is not detectable in CD34+ HSPCs. Overall, these analyses suggest a promising specificity profile of our selected gRNAs and base editor enzyme. Moreover, while we used a PAM relaxed Cas variant to facilitate the development of our epitope engineering for this proof-of-concept work, after identification of the desired base change for epitope engineering it will be possible to test several Cas variants with specific PAM restrictions that better fit our target genes and concomitantly provide an overall higher specificity profile for a prospective clinical application. As a representative example, we tested an alternative gRNA for FLT3 (FLT3-16) with an AGA PAM (binding 2-nt upstream of FLT3-18) in combination with a more restrictive Cas variant (SpG). Targeted deep sequencing of the top intronic and exonic

FLT3-16 predicted OT sites showed significant deamination over background while this base editor preserved ~90% of the on-target activity compared to the FLT3-18 SpRY (70.6% vs 81% median editing in CD34+ HSPCs, respectively; Supplementary Data FigG.4c and Table 20).

Since base editing enzymes can also have non-gRNA-dependent OT effects, we interrogated our new RNAseq dataset generated on CD34+ HSPCs (from Fig.3h and Extended Data Fig.5) to assess the occurrence of major OT activity on some of the expressed genes. Despite the limitations of this analysis, which had to be confined to the transcripts with a high sequencing coverage (top 5%) in order to guarantee proper statistical power, we observed a promising profile with no significant A>G conversions in edited samples compared to controls (Supplementary Data Fig.5b). This analysis confirmed that any OT activity on RNA, if occurring, is highly transient and not detectable after 4 days from base editing treatment (the time at which HSPCs were analyzed for RNAseq). Moreover, it also provided a low-sensitivity but unbiased assessment of unspecific genomic deamination of genes that are highly expressed in the edited cells, which are more prone to unspecific editing due to the presence of ssDNA denatured by the passage of the transcriptional machinery.

Finally, we also evaluated the occurrence of undesired on-target mutations by performing deep sequencing of the edited genes. This analysis showed a low rate of indel formation, which was on average 0.66% for all target loci (0.6, 1.2 and 0.2% for FLT3, CD123 and KIT, respectively; Supplementary Fig. 5c), in line with previously reported results utilizing adenine base editors^{4,5}. As highlighted in the discussion and mentioned above, while we cannot formally exclude that base editing could induce additional unpredicted and unwanted on-target mutations, the region affected by epitope editing (extracellular domain 4 for FLT3 and KIT and the N-terminal domain for CD123) is distant from mutational hotspots involved in cancer-associated variants (ie. tyrosine kinase domain for KIT-D816 or FLT3-D835 or the FLT3 juxta-membrane auto-inhibitory domain involved in FLT3 internal tandem duplications – see also Response to the Referees - Figure 5). Thus, the main concern for these indels would be a possible loss of function in a fraction of the treated cells, which will be spontaneously counter-selected due to the reduced fitness of the resulting cells within the stem cell pool.

These results are reported in Supplementary Data Fig.4 and 5 and described in the new paragraph “Off-target effects of epitope editing” on lines 315-342: *“Since the use of SpRY-Cas9 might lead to potential gRNA-dependent off-target (OT) effects, we performed a specificity analysis by combining genome-wide, unbiased identification of OT sites (GUIDE-Seq) and in silico OT prediction analysis (CRISPOR). To identify potential OTs of our selected FLT3, KIT and CD123 gRNAs, we first performed a GUIDE-seq screening using the SpRY-nuclease (Supplementary Data Fig.4a). By mapping the identified OT sites with mismatches or bulge ≤ 6 , we found that all of them were located in non-coding genomic regions (12 intronic and 11 intergenic, Supplementary Data FIG.4b and Table 16). Thus, we characterized the top exonic and intronic in silico predicted OT sites for FLT3 (N=12), CD123 (N=9), KIT (N=12) sgRNAs and assessed the levels of undesired deamination on BE CD34+ HSPCs by targeted deep sequencing (Supplementary data FIG.5a). In this analysis, no significant off-target deamination over background was observed for any of the analyzed CD123 or KIT sites (Supplementary Data FIG.4c), while 4 loci showed comparatively higher deamination for the FLT3-18 sgRNA. Among these FLT3 OTs, only one was located in an exonic sequence but the affected gene, SNTG1 (Syntrophin γ 1), is a brain-specific syntrophin*

family protein with no expression (Supplementary Data Table 22) nor known functional role in hematopoietic tissue¹⁻³. Despite this generally safe profile, we found that the use of an alternative gRNA (binding 2-nt upstream of FLT3-18 with an AGA PAM) in combination with a more restricted Cas variant (SpG) allows avoiding OT deamination at the predicted OT sites while preserving ~90% of on-target activity compared to the FLT3-18 sgRNA (70.6% vs 81% median editing in CD34+ HSPCs, respectively; Supplementary Data FIG.4c and Table 20). To assess the occurrence of major non-gRNA-dependent deaminations, we interrogated our RNAseq dataset generated on CD34+ HSPCs (from Fig.3h and Extended Data Fig.5) and observed no significant A>G conversions on transcripts with high sequencing coverage (top 5%) in edited samples compared to controls (Supplementary Data FIG.5b). Finally, we evaluated the rate of on-target indels, which were below 1.5% for all target loci (0.6, 1.2 and 0.2% for FLT3, CD123 and KIT, respectively; Supplementary FIG. 5c), in line with previously reported data for ABE^{4,5}. Overall, these data support a generally safe genotoxicity profile of FLT3, CD123 and KIT epitope editing in CD34+ HSPCs.”

While overall these encouraging results suggest a promising safety profile of our selected base editing enzymes, we continue to highlight in the discussion the importance of performing a more comprehensive characterization of the specificity of gRNA and non-gRNA dependent off-target activity as well as a careful evaluation of risk/benefit ratio before clinical implementation.

10. The authors claim “with no differences within the myeloid and lymphoid lineages (FACS-sorted CD33+ and CD19+ cells, respectively; Fig.4B)”. However, the p-values show a difference. Also, the p values in the CD33+ group and the p value in the CD19+ group are identical (0.028582). This seems to be unlikely.

Fig.4b shows the FLT3 editing efficiency in CD34+ liquid culture, mouse peripheral blood and sorted BM CD33+ myeloid and CD19+ lymphoid subsets at the end of the experiment. There are no significant differences between BM CD33+ and CD19+ in untreated mice (light pink), supporting the lack of any lineage skewing induced by the FLT3 editing procedure. The significant p values refer to the difference between untreated and 4G8-CAR treated conditions, within the CD33 and CD19 subsets. This is expected due to the negative selection of non-epitope-edited cells and provides further evidence for the protection of edited hematopoietic lineages. Furthermore, the editing increase is more pronounced within CD19+ cells, consistent with the pattern of FLT3 expression on lymphoid cells that are thus better targeted by CAR-T. While the same comparison did not reach statistical significance in the CD123 experiment (p=0.06), there is a clear trend toward editing increase within the CD33+ subset (Fig.5b), again consistent with the myeloid depleting effect of CD123-targeted CAR-T cells. As mentioned in the reply to Reviewer #1 and noted in the discussion of this manuscript, this strategy can be exploited to confer a selective advantage to genetically engineered cells in the context of non-genotoxic pre-transplant conditioning, thus possibly broadening the applicability of epitope engineering also to non-malignant diseases.

The p values in the CD33+ and CD19+ subsets are identical due to the effects of the correction for multiple comparisons. While this is mathematically correct and recurs in other comparisons in our manuscript, we reviewed all the statistical analyses of the manuscript with a professional

statistician (Dr. Danilo Pellin, now included in the author list) and updated the statistical analysis of Fig.4b with non-corrected multiple t tests, as the specific experimental setting (repeated measures across timepoints) does not necessitate FDR correction.

11. Fig. 5p and q, it looks like the count of HSCs in CD132BE and AACS1 BE groups are significantly lower than no CAR group (5.3-fold difference), but the p-value is not significant (P=0.21). Also, these numbers are identical in Figure 5q.

As discussed above, the reduction of HSC abundance with CAR-T cell treatment appears to be a non-specific effect associated with any T cell treatment in our humanized NBSGW model instead of a CAR-mediated on-target effect (new Extended Data Fig.7n). The statistical analysis refers to the comparison between the AAVS1 and CD123 groups treated with CAR T cells, which is the focus of the presented experiment. While FLT3 and CD123 are expressed at low levels in the primitive HSCs, our data showed some degree of protection of epitope-edited HSCs that, albeit not reaching statistical significance, was consistent across all the described *in vivo* CAR experiments (3.1x, 5.3x, and 7.1x more HSCs in the edited group with FLT3, CD123, and FLT3+CD123 combined editing, respectively, Fig.4m, 5p, 6h). We have now reorganized the plots in Fig.4, Fig.5, and Fig.6, by separating the progenitor populations (which have the lower absolute numbers) and changing the scale axes, to better highlight the protective effect on edited HSC and we have updated the legend of these figures to better clarify the represented statistical comparisons.

Minor comments:

1. The extended data J and K are missing.

We thank the Reviewer for spotting this mistake. We have amended the error in Extended Data Figures.

2. More detailed information on the statistical analyses should be provided.

We apologize for the lack of clarity. We strived to provide inferential analyses for most of our experimental data, as appropriate for the number and type of observations. We have now reviewed all the statistical analyses of the manuscript with a professional statistician (Dr. Danilo Pellin, now included in the author list) and updated the Methods section to provide more detailed information on the utilized statistical analyses. Furthermore, we provided Source Data for all the main figures, 23 Supplementary Data Tables reporting additional raw data and statistical tests for RNAseq, phospho-proteomic and off-target analyses.

Throughout our manuscript, we specified the number of biologically independent samples, animals or technical replicates in the figure legend of each plot (N) or within the plot itself. We reported each observation whenever the graphical appearance of the plot made it possible and plotted the Mean \pm Standard Deviation (SD) as error bars.

Comparing one variable between two groups was performed by unpaired t-tests. Multiple t tests were FDR corrected using the Benjamini, Krieger and Yekutieli method and the corrected p values are reported as appropriate.

When we compared one variable among >2 groups, we used one-way ANOVA, as in the case of hematopoietic populations of *in vivo* samples or editing efficiencies in different stem cell subsets. Multiple comparisons with two-stage step-up procedure of Benjamini, Krieger and Yekutieli to control the false discovery rate (FDR, $Q = 0.05$) were performed to identify differences between the individual conditions. When evaluating the effect of the epitope-edited vs control-edited conditions, only the comparison of edited vs AAVS1 or unedited is reported (as shown by the brackets on the plotted conditions).

When we compared multiple variables among >2 groups, we used two-way ANOVA, as in the case of CAR-T co-cultures or in stimulation experiments of *in vitro* differentiated hematopoietic lineages. We reported the p-value of the row or column effect (according to the comparison of interest) as appropriate (eg. editing effect). Multiple comparisons between groups were performed and used the two-stage step-up procedure of Benjamini, Krieger and Yekutieli to control FDR ($Q = 0.05$).

In all the analyses, the significance threshold was set at 0.05, but in some instances we show p values <0.1 to highlight a trend towards significance.

Analyses were performed using GraphPad Prism v.9.4 (GraphPad) and R software.

The details for the statistical analyses of genomic off targets, RNAseq, phosphoproteomic profiling by MS, RNA editing and antibody/ligand affinity curves (an updated analysis compared to the previous version of the manuscript) are reported in the Method section describing these experiments.

References

1. Piluso, G. *et al.* γ 1- and γ 2-syntrophins, two novel dystrophin-binding proteins localized in neuronal cells. *Journal of Biological Chemistry* **275**, 15851–15860 (2000).
2. Bashiardes, S. *et al.* SNTG1, the gene encoding gammal-syntrophin: a candidate gene for idiopathic scoliosis. *Hum Genet* **115**, 81–89 (2004).
3. Hafner, A., Obermajer, N. & Kos, J. γ -1-syntrophin mediates trafficking of γ -enolase towards the plasma membrane and enhances its neurotrophic activity. *Neurosignals* **18**, 246–258 (2010).
4. Li, J. *et al.* Structure-guided engineering of adenine base editor with minimized RNA off-targeting activity. *Nature Communications* **2021 12:1** **12**, 1–8 (2021).
5. Xue, N. *et al.* Improving adenine and dual base editors through introduction of TadA-8e and Rad51DBD. *Nature Communications* **2023 14:1** **14**, 1–12 (2023).
6. Notta, F. *et al.* Isolation of single human hematopoietic stem cells capable of long-term multilineage engraftment. *Science (1979)* **333**, 218–221 (2011).
7. Fares, I. *et al.* EPCR expression marks UM171-expanded CD34+ cord blood stem cells. *Blood* **129**, 3344–3351 (2017).
8. Genovese, P. *et al.* Targeted genome editing in human repopulating haematopoietic stem cells. *Nature* **510**, 235–240 (2014).
9. Magenta Therapeutics Voluntarily Pauses the MGTA-117 Phase 1/2 Dose-Escalation Clinical Trial to Investigate Drug Safety – Magenta Therapeutics. <https://investor.magentatx.com/news-releases/news-release-details/magenta-therapeutics-voluntarily-pauses-mgta-117-phase-12-dose>.
10. Ferrari, S. *et al.* Efficient gene editing of human long-term hematopoietic stem cells validated by clonal tracking. *Nat Biotechnol* **38**, 1298–1308 (2020).
11. Mackarehtschian, K. *et al.* Targeted disruption of the flk2/flt3 gene leads to deficiencies in primitive hematopoietic progenitors. *Immunity* **3**, 147–161 (1995).
12. Zriwil, A. *et al.* Direct role of FLT3 in regulation of early lymphoid progenitors. *Br J Haematol* **183**, 588–600 (2018).
13. McKenna, H. J. *et al.* Mice lacking flt3 ligand have deficient hematopoiesis affecting hematopoietic progenitor cells, dendritic cells, and natural killer cells. *Blood* **95**, 3489–3497 (2000).
14. Tsapogas, P. *et al.* In vivo evidence for an instructive role of fms-like tyrosine kinase-3 (FLT3) ligand in hematopoietic development. *Haematologica* **99**, 638–646 (2014).
15. Kikushige, Y. *et al.* Human Flt3 Is Expressed at the Hematopoietic Stem Cell and the Granulocyte/Macrophage Progenitor Stages to Maintain Cell Survival. *The Journal of Immunology* **180**, 7358–7367 (2008).
16. Humbert, O. *et al.* Engineering resistance to CD33-targeted immunotherapy in normal hematopoiesis by CRISPR/Cas9-deletion of CD33 exon 2. *Leukemia* **33**, 762–808 (2019).

17. Taylor, V. C. *et al.* The myeloid-specific sialic acid-binding receptor, CD33, associates with the protein-tyrosine phosphatases, SHP-1 and SHP-2. *J. Biol. Chem.* **274**, 11505–11512 (1999).
18. Bhattacharjee, A. *et al.* Repression of phagocytosis by human CD33 is not conserved with mouse CD33. *Communications Biology* **2019 2:1 2**, 1–13 (2019).
19. Ulyanova, T., Blasioli, J., Woodford-Thomas, T. A. & Thomas, M. L. The sialoadhesin CD33 is a myeloid-specific inhibitory receptor. *Eur. J. Immunol.* **29**, 3440–3449 (1999).
20. Bhattacharjee, A. *et al.* The CD33 short isoform is a gain-of-function variant that enhances A β 1–42 phagocytosis in microglia. *Mol Neurodegener* **16**, 1–22 (2021).
21. Naj, A. C. *et al.* Common variants at MS4A4/MS4A6E, CD2AP, CD33 and EPHA1 are associated with late-onset Alzheimer’s disease. *Nat Genet* **43**, 436–443 (2011).
22. Hollingworth, P. *et al.* Common variants at ABCA7, MS4A6A/MS4A4E, EPHA1, CD33 and CD2AP are associated with Alzheimer’s disease. *Nat Genet* **43**, 429–436 (2011).
23. Bertram, L. *et al.* Genome-wide Association Analysis Reveals Putative Alzheimer’s Disease Susceptibility Loci in Addition to APOE. *Am J Hum Genet* **83**, 623–632 (2008).
24. Griciuc, A. *et al.* Alzheimer’s disease risk gene cd33 inhibits microglial uptake of amyloid beta. *Neuron* **78**, 631–643 (2013).
25. Raj, T. *et al.* CD33: increased inclusion of exon 2 implicates the Ig V-set domain in Alzheimer’s disease susceptibility. *Hum Mol Genet* **23**, 2729–2736 (2014).
26. Malik, M. *et al.* CD33 Alzheimer’s Risk-Altering Polymorphism, CD33 Expression, and Exon 2 Splicing. *Journal of Neuroscience* **33**, 13320–13325 (2013).
27. Jiang, T. *et al.* CD33 in Alzheimer’s disease. *Mol Neurobiol* **49**, 529–535 (2014).
28. Griciuc, A. *et al.* Gene therapy for Alzheimer’s disease targeting CD33 reduces amyloid beta accumulation and neuroinflammation. *Hum Mol Genet* **29**, 2920–2935 (2020).
29. Gu, X., Dou, M., Cao, B., Jiang, Z. & Chen, Y. Peripheral level of CD33 and Alzheimer’s disease: a bidirectional two-sample Mendelian randomization study. *Translational Psychiatry* **2022 12:1 12**, 1–6 (2022).
30. Wißfeld, J. *et al.* Deletion of Alzheimer’s disease-associated CD33 results in an inflammatory human microglia phenotype. *Glia* **69**, 1393–1412 (2021).
31. Shaner, N. C. *et al.* A bright monomeric green fluorescent protein derived from Branchiostoma lanceolatum. *Nature Methods* **2013 10:5 10**, 407–409 (2013).
32. ISRCTN - ISRCTN15323014: CAR T cells to fight T cell leukaemia. <https://www.isrctn.com/ISRCTN15323014>.
33. Jetani, H. *et al.* CAR T-cells targeting FLT3 have potent activity against FLT3 – ITD + AML and act synergistically with the FLT3-inhibitor crenolanib. *Leukemia* **32**, 1168–1179 (2018).

34. Balaian, L., Zhong, R. K. & Ball, E. D. The inhibitory effect of anti-CD33 monoclonal antibodies on AML cell growth correlates with Syk and/or ZAP-70 expression. *Exp Hematol* **31**, 363–371 (2003).
35. Zhong, X. & Ma, H. Targeting CD38 for acute leukemia. *Front Oncol* **12**, 5619 (2022).
36. Keyhani, A. *et al.* Increased CD38 expression is associated with favorable prognosis in adult acute leukemia. *Leuk Res* **24**, 153–159 (2000).
37. Naik, J. *et al.* CD38 as a therapeutic target for adult acute myeloid leukemia and T-cell acute lymphoblastic leukemia. *Haematologica* **104**, e100 (2019).
38. Mohamed, M. M. I., Aref, S., Agdar, M. al, Mabed, M. & El-Sokkary, A. M. A. Leukemic Stem Cell (CD34+/CD38-/TIM3+) Frequency in Patients with Acute Myeloid Leukemia: Clinical Implications. *Clin Lymphoma Myeloma Leuk* **21**, 508–513 (2021).
39. Zeijlemaker, W. *et al.* CD34+CD38- leukemic stem cell frequency to predict outcome in acute myeloid leukemia. *Leukemia* *2018* **33**:5, 1102–1112 (2018).
40. Human Acute Myeloid Leukemia CD34+/CD38- Progenitor Cells Have Decreased Sensitivity to Chemotherapy and Fas-induced Apoptosis, Reduced Immunogenicity, and Impaired Dendritic Cell Transformation Capacities | Cancer Research | American Association for Cancer Research. <https://aacrjournals.org/cancerres/article/60/16/4403/506708/Human-Acute-Myeloid-Leukemia-CD34-CD38-Progenitor>.
41. Thomas, D. & Majeti, R. Biology and relevance of human acute myeloid leukemia stem cells. *Blood* **129**, 1577 (2017).
42. Ishikawa, F. *et al.* Chemotherapy-resistant human AML stem cells home to and engraft within the bone-marrow endosteal region. *Nature Biotechnology* *2007* **25**:11, 1315–1321 (2007).
43. Zeijlemaker, W. *et al.* CD34+CD38- leukemic stem cell frequency to predict outcome in acute myeloid leukemia. *Leukemia* *2018* **33**:5, 1102–1112 (2018).
44. Plesa, A. *et al.* High frequency of CD34+CD38-/low immature leukemia cells is correlated with unfavorable prognosis in acute myeloid leukemia. *World J Stem Cells* **9**, 227 (2017).
45. Herrmann, H. *et al.* Delineation of target expression profiles in CD34+/CD38- and CD34+/CD38+ stem and progenitor cells in AML and CML. *Blood Adv* **4**, 5118–5132 (2020).
46. Hope, K. J., Jin, L. & Dick, J. E. Acute myeloid leukemia originates from a hierarchy of leukemic stem cell classes that differ in self-renewal capacity. *Nat Immunol* **5**, 738–743 (2004).
47. Bonnet, D. & Dick, J. E. Human acute myeloid leukemia is organized as a hierarchy that originates from a primitive hematopoietic cell. *Nat Med* **3**, 730–737 (1997).
48. Haubner, S. *et al.* Coexpression profile of leukemic stem cell markers for combinatorial targeted therapy in AML. *Leukemia* **33**, 64–74 (2019).
49. Grafone, T., Palmisano, M., Nicci, C. & Storti, S. An overview on the role of FLT3-tyrosine kinase receptor in acute myeloid leukemia: Biology and

- treatment. *Oncology Reviews* vol. 6 64–74 Preprint at <https://doi.org/10.4081/oncol.2012.e8> (2012).
50. Testa, U., Pelosi, E. & Castelli, G. CD123 as a Therapeutic Target in the Treatment of Hematological Malignancies. *Cancers (Basel)* **11**, (2019).
 51. Russkamp, N. F., Myburgh, R., Kiefer, J. D., Neri, D. & Manz, M. G. Anti-CD117 immunotherapy to eliminate hematopoietic and leukemia stem cells. *Exp Hematol* **95**, 31–45 (2021).
 52. Wellhausen, N. Epitope Editing in Hematopoietic Cells Enables CD45-Directed Immune Therapy. Preprint at (2022).
 53. Gill, S. *et al.* Preclinical targeting of human acute myeloid leukemia and myeloablation using chimeric antigen receptor-modified T cells. *Blood* **123**, 2343–2354 (2014).

Reviewer Reports on the First Revision:

Referees' comments:

Referee #1 (Remarks to the Author):

Casirati et al. present a comprehensive rebuttal. They have included thoughtful experiments to validate similar gene and phospho-protein expression of edited CD34+ HSPCs and also validated the normal function of differentiated edited cells using a number of functional readouts. Sufficient off-target validation is performed through GUIDE-seq in HEK 293T cells and deep sequencing of candidate off-target sites. The authors show that there is potential for some limited off-target editing and this limitation is stated in the discussion. I don't have any other concerns and congratulate the authors on this impressive and innovative study.

Referee #2 (Remarks to the Author):

The authors performed most experiments and addressed most inquiries. However, there are still some minor comments that need to be addressed.

1. The description of "the p-value of the editing effect is reported" in Figure legends and p-value in plots are not clear. When multiple variables are compared among >2 groups, multiple comparison is necessary in some figures to assess the statistical significance for the specified two groups, and the p-values of multiple comparison need to be indicated in plots or figure legends. E.g. Figure 1h and j, Figure 2 c-e, Figure 3 e and f, Figure 6 c, and Extended Data Figures.
2. Figure 1c, the western blot of FLT3 and pFLT3 of WT and FLT3 variant cells should be placed on one membrane without empty lanes for imaging as the western blot of pKit.
3. Figure 2e, surface expression of CD123 on residual live target cells was stained by 9F5. The x-axis of flow chart "FSC-A" should be changed to "CD123 9F5", similarly to c and d. The group "unedited" in plot of "abs. Liver fraction" should be changed to "CD123 WT".
4. In Extended Data Fig. 10 b and c, there are total of 10 groups, while only 9 groups are in some plots. Please check the results.
5. In the Results section of "Off-target effects of epitope editing", the important results in the supplementary data are suggested to be added to the main figure.

Referee #3 (Remarks to the Author):

Although the authors have made efforts, my major concerns still remain. The current study lacks conceptual novelty and significance. Engineering stem cells to avoid CAR T cell activity has been reported in the literature. Importantly, there are no consistent data in the literature showing that current anti-AML CAR T cells can clear CD34+ HSCs. Although the authors have addressed this concern in one paragraph on how to implement the strategy in future clinical studies, it would be difficult to do so due to its complexity and the associated high cost. The safety concern still has not been adequately addressed. For example, the argument that the forced mutation region is not in the spontaneous mutation region found in the database cannot convince that the forced mutations are

safe.

Additionally, the author claimed that the decrease of HSC abundance with CAR-T cell treatment appears to be a non-specific effect associated with any T cell treatment in the humanized NBSGW model instead of a CAR-mediated on-target effect and included the data in the Extended Data Fig. 7n. The data in Extended Data Fig. 7n show that untransduced T cells dramatically depleted HSCs (from 10% to 2%). However, the data in Figure 4m show that FLT3 BE+4G8 T cells (edited one) did not significantly protect HSC when compared to control ($P = 0.12$). These data make the reviewer believe that gene editing does not have a significantly protective role on HSCs, which is hard to explain.

To address the above issue, the authors argue that the late-stage cells show difference. For example, FLT3 BE+4G8 T cells and control show differences for Pro-B, pre-B, and B cells. However, it does not make sense that FLT3 BE+4G8 T cells show fewer B cells than control (AAVS1 BE + 4G8, $P = 0.003$).

Epitope Editing of Hematopoietic Stem Cells Enables Adoptive Immunotherapies for Acute Myeloid Leukemia

2022-09-14750

We thank the Editor and the Reviewers for their careful evaluation of our revised manuscript. We are glad to hear about the overall positive response by the Referees and we are happy to address any remaining concerns raised in the Referees' comments.

Point-by-point reply to the Editor and Reviewers

Referees' comments:

Referee #1 (Remarks to the Author):

Casirati et al. present a comprehensive rebuttal. They have included thoughtful experiments to validate similar gene and phospho-protein expression of edited CD34+ HSPCs and also validated the normal function of differentiated edited cells using a number of functional readouts. Sufficient off-target validation is performed through GUIDE-seq in HEK 293T cells and deep sequencing of candidate off-target sites. The authors show that there is potential for some limited off-target editing and this limitation is stated in the discussion. I don't have any other concerns and congratulate the authors on this impressive and innovative study.

We appreciated the Referee's constructive comments and insight that helped us improve our original manuscript.

Referee #2 (Remarks to the Author):

The authors performed most experiments and addressed most inquiries. However, there are still some minor comments that need to be addressed.

1. The description of "the p-value of the editing effect is reported" in Figure legends and p-value in plots are not clear. When multiple variables are compared among >2 groups, multiple comparison is necessary in some figures to assess the statistical significance for the specified two groups, and the p-values of multiple comparison need to be indicated in plots or figure legends. E.g. Figure 1h and j, Figure 2 c-e, Figure 3 e and f, Figure 6 c, and Extended Data Figures.

As suggested by the Referee, we have now included in the plots of Fig.1h-i, Fig.2c-e and Fig.3e-f the p values of the 2-way ANOVA multiple comparisons for the specified two groups to better describe the presented data. We agree that the new analyses further improve the interpretation of both affinity curves and in vitro killing assays. In Fig. 6c, due to space constraint and the high number of comparisons, we included these analyses in Supplementary Table 24.

2. Figure 1c, the western blot of FLT3 and pFLT3 of WT and FLT3 variant cells should be placed on one membrane without empty lanes for imaging as the western blot of pKit.

The western blot of FLT3 and pFLT3 of WT and FLT3 variants was performed on two membranes that were developed together and imaged with the same acquisition settings and exposure time. Despite this, we agree with the reviewer that this western blot can only be interpreted qualitatively (as such, we only comment on preserved phosphorylation of FLT3 upon ligand exposure). A more quantitative assessment (with normalization on actin levels from the same samples) of the pFLT3/FLT3 western blot on WT, N399D and N399G is reported in Extended Data Fig.2. According to editorial policies, full-size uncropped pictures of the western blot membranes are reported (Supplementary Data Fig.1 and 2).

3. Figure 2e, surface expression of CD123 on residual live target cells was stained by 9F5. The x-axis of flow chart “FSC-A” should be changed to “CD123 9F5”, similarly to c and d. The group “unedited” in plot of “abs. Live fraction” should be changed to “CD123 WT”.

As suggested by the Referee, we updated the representative flow cytometry plots to show CD123 expression with the control mAb on the x-axis. The group “unedited” in plot of “abs. Live fraction” has been changed to “CD123 WT”.

4. In Extended Data Fig. 10 b and c, there are a total of 10 groups, while only 9 groups are in some plots. Please check the results.

We thank the reviewer for noting this incongruence between legends and presented data. We updated the legends for Extended Data Fig. 10 b and c to correctly reflect the groups reported in each plot/experiment.

5. In the Results section of “Off-target effects of epitope editing”, the important results in the supplementary data are suggested to be added to the main figure.

As suggested, we moved the Off-target analysis of the epitope editing to Extended Data Fig.8, thanks to the Editor who kindly approved one additional Extended Data item.

Referee #3 (Remarks to the Author):

Although the authors have made efforts, my major concerns still remain. The current study lacks conceptual novelty and significance.

The epitope engineering approach described in this manuscript has never been reported in the literature. We believe that our findings will be of general interest in the field and provide significant advances compared to alternative approaches, as demonstrated by a great deal of attention received at our oral presentations delivered at the 2022 ASH meeting (Abstract Achievement award), the 2023 Keystone symposia (travel award), the tandem meeting of the

ASTCT & CIBMTR (Young Investigator Awards conferred to the first author) and the ASGCT annual meeting (excellence in research award). We were also invited to present these data at the plenary section of the 2023 ESGCT meeting, the ISSCR meeting (travel award), and the ASGCT spotlight on Immuno-Oncology Conference, again underlining the interest in this work for the fields of adoptive immunotherapy, gene editing, and transplantation medicine.

The work has a high significance because it could support the development of novel and more effective immunotherapy approaches for the treatment of AML and other malignancies. A few other academic and industry research groups are currently developing similar approaches exploiting different antigens and editing technologies, further supporting the relevance of this work in addressing this unmet clinical need in leukemia treatment.

Engineering stem cells to avoid CAR T cell activity has been reported in the literature.

We have cited relevant publications that report the knock-out of a hematopoietic-dispensable antigen (CD33) to avoid on-target toxicities by CAR-T cells. Compared to previously reported strategies, the advantages of the epitope-editing approach are several: (1) it is potentially applicable to every target Ag, regardless of its function in HSPC biology or hematopoietic cells, and we showed the feasibility on 3 antigens relevant to AML immunotherapy; (2) it does not eliminate or disrupt the expression and function of the target antigen in the normal hematopoiesis derived from the modified cells, which may be desirable even for non-indispensable genes; (3) it allows targeting genes with essential roles in tumor biology and survival, thus reducing the risk of immune escape by Ag downregulation or loss; (4) thanks to the minimal modification involved (single amino-acid changes) it is efficiently introduced into HSPCs by novel, less toxic genome editing technologies, such as base editing (BE). Compared to nuclease-mediated HDR, BE is better tolerated by HSPC, alleviates the risks associated with dsDNA breaks and is suitable for multiplexing, which represents another key goal for AML immunotherapy; (5) as mentioned in the discussion, the possibility of multiplexing can allow coupling epitope editing with other therapeutic base editing approaches, thus enabling the exploitation of this strategy in autologous gene therapies to achieve progressive in vivo selection and enrichment of genetically modified cells by multiple infusions of mAb, ADC or BiTE.

Importantly, there are no consistent data in the literature showing that current anti-AML CAR T cells can clear CD34+ HSCs.

While AML CAR-T cell therapies are commonly aimed at HSPC antigens, it is not necessarily expected that limiting toxicities would result from on-target elimination of bona fide HSCs: pancytopenia may also arise from prolonged elimination of more mature progenitors or differentiated cells, which expose the patients to cytopenia, increased risk of infections and may, in turn, promote HSC exhaustion in the long-term. Pre-clinical studies demonstrating the toxicity of AML CAR-T cells on HSPCs and strategies to minimize them are several¹⁻⁵. As a matter of fact, there are no currently approved anti-AML CAR-T cell products, while several are in the early clinical phase (85 results by searching for “Acute Myeloid Leukemia” and “CAR” on ClinicalTrials.gov as of 05/28/23). For the vast majority of these trials, the availability of an allogeneic donor is a condition required for patient enrollment. These eligibility criteria highlight the necessity – even in the best-case scenario of effective anti-tumoral response – of a backup strategy to resolve possible ongoing toxicity by replacing hematopoietic cells. While there is

limited available data from these early-phase trials, on-target/off-tumor effects of targeted immunotherapies in AML are difficult to assess in the clinical setting, as patients suffer from severe cytopenia due to the disruption of the BM niche by AML or previous anti-leukemia treatments. Furthermore, the majority of patients achieving measurable response would proceed to allo-HSCT, again making it impossible to evaluate the CAR-mediated on-target toxicity towards hematopoietic progenitors.

Although the authors have addressed this concern in one paragraph on how to implement the strategy in future clinical studies, it would be difficult to do so due to its complexity and the associated high cost.

VOR Biopharma is currently conducting a phase 1/2 trial (NCT04849910) of CD33-KO HSCT (Tremtelectogene Empogeditemcel or Trem-cel) with post-HSCT treatment with Mylotarg and plans to evaluate CD33 CAR-T cells in combination Trem-cel in the future (reported in the IND-enabling phase on VOR's website as of 05/28/23), thus demonstrating the feasibility to conduct clinical trials using a general strategy of removing the target from transplanted cells that is similar to the one proposed in this manuscript. While we recognize the substantial scientific and regulatory hurdles towards the clinical translation of these new and complex cellular therapies, the cost of these products is likely to decrease in the coming years, thanks to the diffusion of closed system GMP devices, lower costs of GMP-grade reagents and the improvement in cellular manufacturing technologies. Additionally, increased competition in the biotechnology sector and the growing adoption of these therapies by healthcare systems worldwide will contribute to cost reductions, paving the way for broader accessibility to these groundbreaking treatments.

The safety concern still has not been adequately addressed. For example, the argument that the forced mutation region is not in the spontaneous mutation region found in the database cannot convince that the forced mutations are safe.

We provided several new experimental data supporting the safety of the proposed mutations in terms of i) preservation of the receptor functionality (RNAseq, phospho-proteomic analysis), ii) preservation of HSPC phenotype, differentiation, and long-term persistence (editing in sorted subpopulations, functional assays on differentiated cells, secondary transplantation) and iii) minimal genotoxicity of the editing procedure (off-target analysis). The lack of cancer-associated mutations in online databases (COSMIC) at the intended codon for FLT3 and KIT serves as additional assurance of the low risk of targeted genome modification at these loci/protein domains. Furthermore, the FLT3 N399D and KIT H378R mutations are naturally found in several mammal orthologs (non-human primates for FLT3 and mice and rats for KIT), further suggesting that the forced mutations can exert physiologic functionality of these receptors.

Additionally, the author claimed that the decrease of HSC abundance with CAR-T cell treatment appears to be a non-specific effect associated with any T cell treatment in the humanized NBSGW model instead of a CAR-mediated on-target effect and included the data in the Extended Data Fig. 7n. The data in Extended Data Fig. 7n show that untransduced T cells dramatically depleted HSCs (from 10% to 2%). However, the data in Figure 4m show that FLT3 BE+4G8 T cells (edited one) did not significantly protect HSC when compared to control ($P = 0.12$). These data make the

reviewer believe that gene editing does not have a significantly protective role on HSCs, which is hard to explain.

We apologize for the lack of clarity on the expected outcomes of epitope-engineering when combined with anti-AML immunotherapies: primitive HSC subsets (lineage-CD34+38-10-45RA-90+) are not the main target of the on-target toxicity induced by FLT3 and CD123 immunotherapies, due to the low levels of antigen expression. Lineage-committed progenitors clearly expressing the target antigens (e.g., GMP, LMPP, and differentiated subsets) are more efficiently depleted by CAR-T cells and are the likely cause of hematopoietic toxicity in patients receiving FLT3 or CD123-targeted therapies and therefore their preservation is one of the aims of the proposed research. Importantly, we show that these progenitors are protected by epitope editing.

The vast majority of HSC depletion observed in our murine model is caused by a non-CAR-specific effect (induced by untransduced T cells as well), which is consistent with cytokine-mediated toxicity associated with T cell expansion. CAR-induced myelotoxicity has been observed in patients receiving CD19 CAR-T cells for B-ALL and DLBCL and is increasingly being recognized by the medical community⁶⁻²¹. According to the EBMT/EHA CAR-T cell handbook, hematologic toxicity is the most common adverse event after CAR-T cell therapy, with a cumulative 1-year incidence of 58% (CTCAE grade ≥ 3) in the real-world setting¹⁷. Studies report incidences up to 60% at 28-42 days and persisting grade 3-4 cytopenias in 27% of patients at 1 year^{18,19}. Our model seems to recapitulate these CAR-induced effects (Extended Data Fig. 7n).

Despite the confounding effect caused by the unspecific T cell effect, the epitope-engineered groups show a trend towards improved HSC counts compared to AAVS1 controls (Fig.4m, 5p, 6h), suggesting that editing can also protect this cell compartment from low-level CAR activation or from the cytokine-mediated effects (by reducing CAR-T cell expansion thanks to the reduced antigen burden). Notably, if the HSCs' absolute counts from CAR-treated conditions are extrapolated and analyzed independently (excluding the variability of the untreated condition), the degree of protection endowed by epitope-editing reaches statistical significance in all three presented examples (Response to the Referee's Fig.1). While this simplified analysis would further support the protective role of epitope-editing even on HSC expressing low levels of the antigen(s), we preferred to include the more comprehensive FDR-adjusted one-way ANOVA comparisons in the main figures.

Response to the Referee's Figure 1 – Absolute counts of HSCs from CAR-treated conditions reported in Fig.4, 5 and 6 (FLT3, CD123 and FLT3+CD123 experiments). Statistical comparison by t-test.

To address the above issue, the authors argue that the late-stage cells show difference. For example, FLT3 BE+4G8 T cells and control show differences for Pro-B, pre-B, and B cells. However, it does not make sense that FLT3 BE+4G8 T cells show fewer B cells than control (AAVS1 BE + 4G8, P = 003).

At the timepoint selected for in vivo experiments (2 weeks after CAR-T cell administration), we observe the effects of on-target CAR killing, which are proportional to antigen levels and absolute cell numbers (Extended Data Fig.8e and 9a), while it is more difficult to precisely quantify the effects on the mature progeny not expressing the target antigen (which have different in vivo half-lives and derive from the depleted progenitors with variable maturation times).

Regarding the example reported by the Referee, as reported in Fig.7n (ie. treatment with untransduced T cells), we observed an increased trend of mature B cell numbers in the conditions treated with T cells, again likely associated with released soluble mediators or immune crosstalk. As reported in Fig.4d, AAVS1 controls show 1.4x higher CAR-T cell engraftment than FLT3-BE mice, likely caused by the decreased antigen burden in the FLT3-BE group. These data suggest that the observed B cell increase may be associated with higher CAR-T activation and expansion. Additionally, the magnitude of the effect on depleted/increased cell subsets must be considered: AAVS1 controls have on average 14.9 M less pre-B (the largest population in humanized NBSGW mice) and 1.7 M less pro-B cells compared to FLT3-BE mice, while the increased number in mature B cells (CD19+10-34-) accounts for only 0.29 M cells. These minor changes, which can only be appreciated thanks to our comprehensive and granular gating strategy, further support the argument that the 1.4x higher CAR-T cell engraftment is the cause of mature B cell expansion in the FLT3-BE group. As LMPP, pre-B/NK, B-prolymphocytes, pro-B and pre-B cells are all significantly depleted by FLT3 CAR-T in AAVS1 controls, it is reasonable to expect that mature B cells may decrease over time, but this effect cannot be evaluated in our short-term murine experimental setting. Moreover, the strong B cell lineage bias of the NBSGW xeno-transplanted mouse model may further complicate this observation even in long-term studies. To more rigorously evaluate the long-term consequences of CAR treatment on differentiated subsets, future experiments will be performed on non-human primates, which will provide a more relevant model for human hematopoiesis.

References

1. Jetani, H. *et al.* CAR T-cells targeting FLT3 have potent activity against FLT3 – ITD + AML and act synergistically with the FLT3-inhibitor crenolanib. *Leukemia* **32**, 1168–1179 (2018).
2. Myburgh, R. *et al.* Anti-human CD117 CAR T-cells efficiently eliminate healthy and malignant CD117-expressing hematopoietic cells. *Leukemia* **34**, 2688–2703 (2020).
3. Sommer, C. *et al.* Allogeneic FLT3 CAR T Cells with an Off-Switch Exhibit Potent Activity against AML and Can Be Depleted to Expedite Bone Marrow Recovery. (2020) doi:10.1016/j.ymthe.2020.06.022.
4. Benmebarek, M. R. *et al.* A modular and controllable T cell therapy platform for acute myeloid leukemia. *Leukemia* *2021* **35**:8 **35**, 2243–2257 (2021).

5. Loff, S. *et al.* Rapidly Switchable Universal CAR-T Cells for Treatment of CD123-Positive Leukemia. *Mol Ther Oncolytics* **17**, 408 (2020).
6. Jain, T. *et al.* Hematopoietic recovery in patients receiving chimeric antigen receptor T-cell therapy for hematologic malignancies. *Blood Adv* **4**, 3776–3787 (2020).
7. Corona, M. *et al.* Management of prolonged cytopenia following CAR T-cell therapy. *Bone Marrow Transplantation* **2022** *57:12* **57**, 1839–1841 (2022).
8. Wang, L. *et al.* New-Onset Severe Cytopenia After CAR-T Cell Therapy: Analysis of 76 Patients With Relapsed or Refractory Acute Lymphoblastic Leukemia. *Front Oncol* **11**, 2433 (2021).
9. Penack, O. *et al.* Severe cytopenia after CD19 CAR T-cell therapy: a retrospective study from the EBMT Transplant Complications Working Party. *J Immunother Cancer* **11**, e006406 (2023).
10. Zhou, J. *et al.* Cytopenia after chimeric antigen receptor T cell immunotherapy in relapsed or refractory lymphoma. *Front Immunol* **13**, (2022).
11. Taneja, A., Jain, T., Correspondence, T. & Jain, S. CAR-T-OPENIA: Chimeric antigen receptor T-cell therapy-associated cytopenias. *EJHaem* **3**, 32–38 (2022).
12. Dowling, M. R. & Dickinson, M. Post CAR-T cytopenia: poorly understood and clinically challenging. <https://doi.org/10.1080/10428194.2022.2095631> **63**, 1774–1776 (2022).
13. Zhou, J. *et al.* Cytopenia after chimeric antigen receptor T cell immunotherapy in relapsed or refractory lymphoma. *Front Immunol* **13**, (2022).
14. Jain, T., Olson, T. S. & Locke, F. L. How I treat cytopenias after CAR T-cell therapy. *Blood* **141**, 2460–2469 (2023).
15. Fried, S. *et al.* Early and late hematologic toxicity following CD19 CAR-T cells. *Bone Marrow Transplant* (2019) doi:10.1038/s41409-019-0487-3.
16. Cordeiro, A. *et al.* Late Events after Treatment with CD19-Targeted Chimeric Antigen Receptor Modified T Cells. *Biol Blood Marrow Transplant* **26**, 26–33 (2020).
17. Subklewe, M., Benjamin, R., Subklewe, M. & Benjamin, R. Management of Myelotoxicity (Aplasia) and Infectious Complications. *The EBMT/EHA CAR-T Cell Handbook* 151–155 (2022) doi:10.1007/978-3-030-94353-0_29.
18. Strati, P. *et al.* Hematopoietic recovery and immune reconstitution after axicabtagene ciloleucel in patients with large B-cell lymphoma. *Haematologica* **106**, 2667–2672 (2021).
19. Sharma, N., Reagan, P. M. & Liesveld, J. L. Cytopenia after CAR-T Cell Therapy—A Brief Review of a Complex Problem. *Cancers (Basel)* **14**, (2022).
20. Rejeski, K. *et al.* CAR-HEMATOTOX: a model for CAR T-cell–related hematologic toxicity in relapsed/refractory large B-cell lymphoma. *Blood* **138**, 2499 (2021).
21. Read, J. A., Rouce, R. H., Mo, F., Mamonkin, M. & King, K. Y. Apoptosis of Hematopoietic Stem Cells Contributes to Bone Marrow Suppression Following Chimeric Antigen Receptor T Cell Therapy. *Transplant Cell Ther* **29**, 165.e1-165.e7 (2023).